# STanHop: Sparse Tandem Hopfield Model for Memory-Enhanced Time Series Prediction

**Dennis Wu**[*†]    **Jerry Yao-Chieh Hu**[*†]    **Weijian Li**[*†]    **Bo-Yu Chen**[‡]    **Han Liu**[†♮]

[†] Department of Computer Science, Northwestern University, Evanston, IL 60208, USA
[‡] Department of Physics, National Taiwan University, Taipei 10617, Taiwan
[♮] Department of Statistics and Data Science, Northwestern University, Evanston, IL 60208, USA
{hibb, jhu, weijianli}@u.northwestern.edu, b12202023@ntu.edu.tw
hanliu@northwestern.edu

## Abstract

We present **STanHop-Net** (**S**parse **Tan**dem **Hop**field **Net**work) for multivariate time series prediction with memory-enhanced capabilities. At the heart of our approach is **STanHop**, a novel Hopfield-based neural network block, which sparsely learns and stores both temporal and cross-series representations in a data-dependent fashion. In essence, STanHop sequentially learn temporal representation and cross-series representation using two tandem sparse Hopfield layers. In addition, STanHop incorporates two external memory modules: **Plug-and-Play** and **Tune-and-Play** for train-less and task-aware memory-enhancements, respectively. They allow StanHop-Net to swiftly respond to certain sudden events. Methodologically, we construct the STanHop-Net by stacking STanHop blocks in a hierarchical fashion, enabling multi-resolution feature extraction with resolution-specific sparsity. Theoretically, we introduce a unified construction (**Generalized Sparse Modern Hopfield Model**) for both dense and sparse modern Hopfield models and show that it endows a tighter memory retrieval error compared to the dense counterpart without sacrificing memory capacity. Empirically, we validate the efficacy of STanHop-Net on many settings: time series prediction, fast test-time adaptation, and strongly correlated time series prediction.

## 1 Introduction

In this work, we aim to enhance multivariate time series prediction by incorporating relevant additional information specific to the inference task at hand. This problem holds practical importance due to its wide range of real-world applications. On one hand, multivariate time series prediction itself poses a unique challenge given its multi-dimensional sequential structure and noise-sensitivity (Masini et al., 2023; Reneau et al., 2023; Nie et al., 2022; Fawaz et al., 2019). A proficient model should robustly not only discern the correlations between series within each time step, but also grasp the intricate dynamics of each series over time. On the other hand, in many real-world prediction tasks, one significant challenge with existing time series models is their slow responsiveness to sudden or rare events. For instance, events like the 2008 financial crisis and the pandemic-induced market turmoil in 2021 (Laborda and Olmo, 2021; Bond and Dow, 2021; Sevim et al., 2014; Bussiere and Fratzscher, 2006), or extreme climate changes in weather forecasting (Le et al., 2023; Sheshadri et al., 2021) often lead to compromised model performance. To combat these challenges, we present **STanHop-Net** (**S**parse **Tan**dem **Hop**field **Net**work), a novel Hopfield-based deep learning model, for multivariate time series prediction, equipped with optional memory-enhanced capabilities.

Our motivation comes from the connection between associative memory models of human brain (specifically, the modern Hopfield models) and the attention mechanism (Hu et al., 2023; Ramsauer et al., 2020). Based on this link, we propose to enhance time series models with external information (e.g., real-time or relevant auxiliary data) via the memory retrieval mechanism of Hopfield models. In its core, we utilize and extend the deep-learning-compatible Hopfield layers (Hu et al., 2023; Ramsauer et al., 2020). Differing from typical transformer-based architectures, these layers not only replace the attention mechanisms (Ramsauer et al., 2020; Widrich et al., 2020) but also serve as differentiable memory modules, enabling integration of external stimuli for enhanced predictions.

---

[*]Equal contribution. Code is available at GitHub; full version and future updates are on arXiv.

In this regard, we first introduce a set of generalized sparse Hopfield layers, as an extension of the sparse modern Hopfield model (Hu et al., 2023). Based on these layers, we propose a structure termed the **STanHop** (**S**parse **Tan**dem **Hop**field layers) block. In STanHop, there are two sequentially joined sub-blocks of generalized sparse Hopfield layers, hence tandem. This tandem design sparsely learn and store temporal and cross-series representations in a sequential manner.

Furthermore, we introduce **STanHop-Net** (**S**parse **Tan**dem **Hop**field **Net**work) for time series, consisting of multiple layers of STanHop blocks to cater for multi-resolution representation learning. To be more specific, rather than relying only on the input sequence for predictions, each stacked Stan-Hop block is capable of incorporating additional information through the Hopfield models' memory retrieval mechanism from a pre-specified external memory set. This capability facilitates the injection of external memory at every resolution level when necessary. Consequently, STanHop-Net not only excels at making accurate predictions but also allows users to integrate additional information they consider valuable for their specific downstream inference tasks with minimal effort.

We provide visual overviews of STanHop-Net in Figure 1 and STanHop block in Figure 2.

**Contributions.** We summarize our contributions as follows:

- Theoretically, we introduce a unified sparsity-aware modern Hopfield model, termed the generalized sparse Hopfield model. We show that it not only offer a tighter memory retrieval error bound compared to the dense modern Hopfield model (Ramsauer et al., 2020), but also retains the robust theoretical properties of the dense model, such as fast fixed point convergence and exponential memory capacity. Moreover, it serves as a unified model that encompasses both the sparse (Hu et al., 2023) and dense (Ramsauer et al., 2020) models as its special cases.

- Computationally, we show the one-step approximation of the retrieval dynamics of the generalized sparse Hopfield model is connected to sparse attention mechanisms, akin to (Hu et al., 2023; Ramsauer et al., 2020). This connection allows us to introduce the GSH layers featuring learnable sparsity, for time series representation learning. As a result, these layers achieve faster memory-retrieval convergence and greater noise-robustness compared to the dense model.

- Methodologically, with GSH layer, we present **STanHop** (**S**parse **Tan**dem **Hop**field layers) block, a hierarchical tandem Hopfield model design to capture the intrinsic multi-resolution structure of both temporal and cross-series dimensions of time series with resolution-specific sparsity at each level. In addition, we introduce the idea of pseudo-label retrieval, and debut two external memory plugin schemes — Plug-and-Play and Tune-and-Play memory plugin modules — for memory-enhanced predictions.

- Experimentally, we validate STanHop-Net in multivariate time series predictions, considering both with and without the incorporation of external memory. When external memory isn't utilized, STanHop-Net consistently matches or surpasses many popular baselines, across diverse real-world datasets. When external memory is utilized, STanHop-Net demonstrates further performance boosts in many settings, benefiting from both proposed external memory schemes.

**Notations.** We write $\langle \mathbf{a}, \mathbf{b} \rangle := \mathbf{a}^\mathsf{T}\mathbf{b}$ as the inner product for vectors $\mathbf{a}, \mathbf{b}$. The index set $\{1, \cdots, I\}$ is denoted by $[I]$, where $I \in \mathbb{N}_+$. The spectral norm is denoted by $\lVert \cdot \rVert$, which is equivalent to the $l_2$-norm when applied to a vector. Throughout this paper, we denote the memory patterns (keys) by $\boldsymbol{\xi} \in \mathbb{R}^d$ and the state/configuration/query pattern by $\mathbf{x} \in \mathbb{R}^d$ with $n := \lVert \mathbf{x} \rVert$, and $\boldsymbol{\Xi} := (\boldsymbol{\xi}_1, \cdots, \boldsymbol{\xi}_M) \in \mathbb{R}^{d \times M}$ as shorthand for stored memoery (key) patterns $\{\boldsymbol{\xi}_\mu\}_{\mu \in [M]}$. Moreover, we set $m := \mathrm{Max}_{\mu \in [M]} \lVert \boldsymbol{\xi}_\mu \rVert$ be the largest norm of memory patterns.

## 2 BACKGROUND: MODERN HOPFIELD MODELS

Let $\mathbf{x} \in \mathbb{R}^d$ be the query pattern and $\boldsymbol{\Xi} = (\boldsymbol{\xi}_1, \cdots, \boldsymbol{\xi}_M) \in \mathbb{R}^{d \times M}$ be the $M$ memory patterns.

**Hopfield Models.** Hopfield models are associative models that store a set of memory patterns $\boldsymbol{\Xi}$ in such a way that a stored pattern $\boldsymbol{\xi}_\mu$ can be retrieved based on a partially known or contaminated version, a query $\mathbf{x}$. The models achieve this by embedding the memories $\boldsymbol{\Xi}$ in the *energy landscape* $E(\mathbf{x})$ of a physical system (e.g., the Ising model in (Hopfield, 1982) or its higher-order generalizations (Lee et al., 1986; Peretto and Niez, 1986; Newman, 1988)), where each memory $\boldsymbol{\xi}_\mu$ corresponds to a local minimum. When a query $\mathbf{x}$ is introduced, the model initiates energy-minimizing *retrieval dynamics* $\mathcal{T}$ at the query's location. This process then navigate the energy landscape to locate the nearest local minimum $\boldsymbol{\xi}_\mu$, effectively retrieving the memory most similar to the query $\mathbf{x}$.

Constructing the energy function, $E(\mathbf{x})$, is straightforward. As outlined in (Krotov and Hopfield, 2016), memories get encoded into $E(\mathbf{x})$ using the *overlap-construction*: $E(\mathbf{x}) = F(\mathbf{\Xi}^\mathsf{T}\mathbf{x})$, where $F : \mathbb{R}^M \to \mathbb{R}$ is a smooth function. This ensures that the memories $\{\boldsymbol{\xi}_\mu\}_{\mu\in[M]}$ sit at the stationary points of $E(\mathbf{x})$, given $\nabla_\mathbf{x} F(\mathbf{\Xi}^\mathsf{T}\mathbf{x})|_{\boldsymbol{\xi}_\mu} = 0$ for all $\mu \in [M]$. The choice of $F$ results in different Hopfield model types, as demonstrated in (Krotov and Hopfield, 2016; Demircigil et al., 2017; Ramsauer et al., 2020; Krotov and Hopfield, 2020). However, determining a suitable retrieval dynamics, $\mathcal{T}$, for a given energy $E(\mathbf{x})$ is more challenging. For effective memory retrieval, $\mathcal{T}$ must:

(T1) Monotonically reduce $E(\mathbf{x})$ when applied iteratively.

(T2) Ensure its fixed points coincide with the stationary points of $E(\mathbf{x})$ for precise retrieval.

**Modern Hopfield Models.** Ramsauer et al. (2020) propose the modern Hopfield model with a specific set of $E$ and $\mathcal{T}$ satisfying above requirements, and integrate it into deep learning architectures via its strong connection with attention mechanism, offering enhanced performance, and theoretically guaranteed exponential memory capacity. Specifically, they introduce

$$E(\mathbf{x}) = -\operatorname{lse}(\beta, \mathbf{\Xi}^\mathsf{T}\mathbf{x}) + \frac{1}{2}\langle\mathbf{x}, \mathbf{x}\rangle + \text{Const.}, \quad \text{and} \quad \mathcal{T}_{\text{Dense}}(\mathbf{x}) = \mathbf{\Xi}\operatorname{Softmax}(\beta\mathbf{\Xi}^\mathsf{T}\mathbf{x}) = \mathbf{x}^{\text{new}}, \quad (2.1)$$

where $\mathbf{\Xi}^\mathsf{T}\mathbf{x} = (\langle\boldsymbol{\xi}_1, \mathbf{x}\rangle, \ldots, \langle\boldsymbol{\xi}_M, \mathbf{x}\rangle) \in \mathbb{R}^M$, $\operatorname{lse}(\beta, \mathbf{z}) := \log\left(\sum_{\mu=1}^M \exp\{\beta z_\mu\}\right)/\beta$ is the log-sum-exponential for any given vector $\mathbf{z} \in \mathbb{R}^M$ and $\beta > 0$. Their analysis concludes that:

- $\mathcal{T}_{\text{Dense}}$ converges well (Ramsauer et al., 2020, Theorem 1,2) and can retrieve patterns accurately in just one step (Ramsauer et al., 2020, Theorem 4), i.e. (T1) and (T2) are satisfied.
- The modern Hopfield model (2.1) possesses an exponential memory capacity in pattern size $d$ (Ramsauer et al., 2020, Theorem 3).
- Notably, the one-step approximation of $\mathcal{T}_{\text{Dense}}$ mirrors the attention mechanism in transformers, leading to a novel deep architecture design: the Hopfield layers.

In a related vein, Hu et al. (2023) introduce a principled approach to constructing modern Hopfield models using the convex conjugate of the entropy regularizer. Unlike the original modern Hopfield model (Ramsauer et al., 2020), the key insight of (Hu et al., 2023) is that the convex conjugate of various entropic regularizers can yield distributions with varying degrees of sparsity. Leveraging this understanding, we introduce the generalized sparse Hopfield model in the next section.

## 3 GENERALIZED SPARSE HOPFIELD MODEL

In this section, we extend the entropic regularizer construction of the sparse modern Hopfield model (Hu et al., 2023) by replacing the Gini entropic regularizer with the Tsallis $\alpha$-entropy (Tsallis, 1988),

$$\Psi_\alpha(\mathbf{p}) := \begin{cases} \frac{1}{\alpha(\alpha-1)}\sum_{\mu=1}^M\left(p_\mu - p_\mu^\alpha\right), & \alpha \neq 1, \\ -\sum_{\mu=1}^M p_\mu \ln p_\mu, & \alpha = 1, \end{cases}, \quad \text{for } \alpha \geq 1, \quad (3.1)$$

thereby introducing the generalized sparse Hopfield model. Subsequently, we verify the connection between the memory retrieval dynamics of the generalized sparse Hopfield model and attention mechanism. This leads to the Generalized Sparse Hopfield (GSH) layers for deep learning.

### 3.1 ENERGY FUNCTION, RETRIEVAL DYNAMICS AND FUNDAMENTAL LIMITS

Let $\mathbf{z}, \mathbf{p} \in \mathbb{R}^M$, and $\Delta^M := \{\mathbf{p} \in \mathbb{R}_+^M \mid \sum_\mu^M p_\mu = 1\}$ be the $(M-1)$-dimensional unit simplex.
**Energy Function.** We introduce the generalized sparse Hopfield energy function[1]:

$$\mathcal{H}(\mathbf{x}) = -\Psi_\alpha^\star\left(\beta\mathbf{\Xi}^\mathsf{T}\mathbf{x}\right) + \frac{1}{2}\langle\mathbf{x}, \mathbf{x}\rangle + \text{Const.}, \quad \text{with} \quad \Psi_\alpha^\star(\mathbf{z}) := \int d\mathbf{z}\,\alpha\text{-EntMax}(\mathbf{z}), \quad (3.2)$$

where $\alpha\text{-EntMax}(\cdot) : \mathbb{R}^M \to \Delta^M$ is a finite-domain distribution map defined as follows.

**Definition 3.1.** The variational form of $\alpha$-EntMax is defined by the optimization problem
$$\alpha\text{-EntMax}(\mathbf{z}) := \operatorname*{ArgMax}_{\mathbf{p}\in\Delta^M}[\langle\mathbf{p}, \mathbf{z}\rangle - \Psi_\alpha(\mathbf{p})], \quad (3.3)$$
where $\Psi_\alpha(\cdot)$ is the Tsallis entropic regularizer given by (3.1). See Remark G.1 for a closed form.

$\Psi_\alpha^\star(\mathbf{p})$ is the convex conjugate of the Tsallis entropic regularizer $\Psi_\alpha(\mathbf{p})$ (Definition C.1) and hence

---

[1]This energy function (3.2) is equivalent up to an additive constant.

**Lemma 3.1.** $\nabla \Psi_\alpha^\star(\mathbf{z}) = \text{ArgMax}_{\mathbf{p} \in \Delta^M} [\langle \mathbf{p}, \mathbf{z} \rangle - \Psi_\alpha(\mathbf{p})] = \alpha\text{-EntMax}(\mathbf{z})$.

*Proof.* See Appendix C.1 for a detailed proof. □

**Retrieval Dynamics.** With Lemma 3.1, it is clear to see that the energy function (3.2) aligns with the overlap-function construction of Hopfield models, as in (Hu et al., 2023; Ramsauer et al., 2020). Next, we introduce the corresponding retrieval dynamics satisfying the monotinicity property (T1).

**Lemma 3.2** (Generalized Sparse Hopfield Retrieval Dynamics). *Let $t$ be the iteration number. The retrieval dynamics of the generalized sparse Hopfield model is a 1-step update of the form*

$$\mathcal{T}(\mathbf{x}_t) := \nabla_{\mathbf{x}} \Psi_\alpha^\star \left( \beta \Xi^\mathsf{T} \mathbf{x}_t \right) = \Xi \, \alpha\text{-EntMax} \left( \beta \Xi^\mathsf{T} \mathbf{x}_t \right) = \mathbf{x}_{t+1}, \tag{3.4}$$

*that minimizes the energy function (3.2) monotonically over $t$.*

*Proof.* See Appendix C.2 for a detailed proof. □

To see how this model store and retrieve memory patterns, we first introduce the following definition.

**Definition 3.2** (Stored and Retrieved). *Assuming that every pattern $\boldsymbol{\xi}_\mu$ surrounded by a sphere $S_\mu$ with finite radius $R := \frac{1}{2} \text{Min}_{\mu, \nu \neq \mu \in [M]} \|\boldsymbol{\xi}_\mu - \boldsymbol{\xi}_\nu\|$, we say $\boldsymbol{\xi}_\mu$ is stored if there exists a generalized fixed point of $\mathcal{T}$, $\mathbf{x}_\mu^\star \in S_\mu$, to which all limit points $\mathbf{x} \in S_\mu$ converge to, and $S_\mu \cap S_\nu = \emptyset$ for $\mu \neq \nu$. We say $\boldsymbol{\xi}_\mu$ is $\epsilon$-retrieved by $\mathcal{T}$ with $\mathbf{x}$ for an error $\epsilon$, if $\|\mathcal{T}(\mathbf{x}) - \boldsymbol{\xi}_\mu\| \leq \epsilon$.*

To ensure the convergence property (T2) of retrieval dynamics (3.4), we have the next lemma.

**Lemma 3.3** (Convergence of Retrieval Dynamics $\mathcal{T}$). *Given the energy function (3.2) and retrieval dynamics $\mathcal{T}(\mathbf{x})$ (3.4), respectively. For any sequence $\{\mathbf{x}_t\}_{t=0}^\infty$ generated by the iteration $\mathbf{x}_{t'+1} = \mathcal{T}(\mathbf{x}_{t'})$, all limit points of this sequence are stationary points of $\mathcal{H}$.*

*Proof.* See Appendix C.3 for a detailed proof. □

Intuitively, Lemma 3.3 suggests that for any query $\mathbf{x}$, $\mathcal{T}$ (given by (3.4)) monotonically and iteratively approaches stationary points of $\mathcal{H}$ (given by (3.2)), where the memory patterns $\{\boldsymbol{\xi}_\mu\}_{\mu \in [M]}$ are stored. This completes the construction of a well-defined modern Hopfield model.

**Fundamental Limits.** To highlight the computational benefits of the generalized sparse Hopfield model, we analyze the fundamental limits of the memory retrieval error and memory capacity.

**Theorem 3.1** (Retrieval Error). *Let $\mathcal{T}_{\text{Dense}}$ be the retrieval dynamics of the dense modern Hopfield model (Ramsauer et al., 2020). Let $\mathbf{z} \in \mathbb{R}^M$, $z_{(\nu)}$ be the $\nu$'th element in a sorted descending $z$-sequence $\mathbf{z}_{\text{sorted}} := z_{(1)} \geq \ldots \geq z_{(M)}$, and $\kappa(\mathbf{z}) := \text{Max}\{k \in [M] \mid 1 + kz_{(k)} > \sum_{\nu \leq k} z_{(\nu)}\}$. For all $\mathbf{x} \in S_\mu$, it holds $\|\mathcal{T}(\mathbf{x}) - \boldsymbol{\xi}_\mu\| \leq \|\mathcal{T}_{\text{Dense}}(\mathbf{x}) - \boldsymbol{\xi}_\mu\|$, and*

$$\text{for } 2 \geq \alpha \geq 1, \quad \|\mathcal{T}(\mathbf{x}) - \boldsymbol{\xi}_\mu\| \leq 2m(M-1) \exp\left\{ -\beta \left( \langle \boldsymbol{\xi}_\mu, \mathbf{x} \rangle - \underset{\nu \in [M]}{\text{Max}} \langle \boldsymbol{\xi}_\mu, \boldsymbol{\xi}_\nu \rangle \right) \right\}, \tag{3.5}$$

$$\text{for } \alpha \geq 2, \quad \|\mathcal{T}(\mathbf{x}) - \boldsymbol{\xi}_\mu\| \leq m + d^{1/2} m \beta \left[ \kappa \left( \underset{\nu \in [M]}{\text{Max}} \langle \boldsymbol{\xi}_\nu, \mathbf{x} \rangle - \left[ \Xi^\mathsf{T} \mathbf{x} \right]_{(\kappa)} \right) + \frac{1}{\beta} \right]. \tag{3.6}$$

**Corollary 3.1.1** (Noise-Robustness). *In cases of noisy patterns with noise $\boldsymbol{\eta}$, i.e. $\widetilde{\mathbf{x}} = \mathbf{x} + \boldsymbol{\eta}$ (noise in query) or $\widetilde{\boldsymbol{\xi}}_\mu = \boldsymbol{\xi}_\mu + \boldsymbol{\eta}$ (noise in memory), the impact of noise $\boldsymbol{\eta}$ on the sparse retrieval error $\|\mathcal{T}(\mathbf{x}) - \boldsymbol{\xi}_\mu\|$ is linear for $\alpha \geq 2$, while its effect on the dense retrieval error $\|\mathcal{T}_{\text{Dense}}(\mathbf{x}) - \boldsymbol{\xi}_\mu\|$ (or $\|\mathcal{T}(\mathbf{x}) - \boldsymbol{\xi}_\mu\|$ with $2 \geq \alpha \geq 1$) is exponential.*

*Proof.* See Appendix C.4 for a detailed proof. □

Intuitively, Theorem 3.1 implies the sparse model converge faster to memory patterns than the dense model (Ramsauer et al., 2020), and the larger sparsity leads the lower retrieval error.

**Lemma 3.4** (Memory Capacity Lower Bound). *Suppose the probability of successfully storing and retrieving memory pattern is given by $1 - p$. The number of memory patterns sampled from a sphere of radius $m$ that the sparse Hopfield model can store and retrieve has a lower bound: $M \geq \sqrt{p} C^{\frac{d-1}{4}}$, where $C$ is the solution for $C = b/W_0(\exp\{a + \ln b\})$ with $W_0(\cdot)$ being the principal branch of Lambert $W$ function (Olver et al., 2010), $a := 4/d-1\{ \ln [2m(\sqrt{p}-1)/(R+\delta)] + 1\}$ and $b := 4m^2\beta/5(d-1)$. For*

sufficiently large $\beta$, the sparse Hopfield model has a larger lower bound on the exponential-in-d memory capacity compared to that of dense counterpart (Ramsauer et al., 2020): $M \geq M_{\text{Dense}}$.

*Proof.* See Appendix C.5 for a detailed proof. □

Lemma 3.4 offers a lower bound on the count of patterns effectively stored and retrievable by $\mathcal{T}$ with a minimum precision of $R$, as defined in Definition 3.2. Essentially, the capacity of the generalized sparse Hopfield model to store and retrieve patterns grows exponentially with pattern size $d$. This mirrors findings in (Hu et al., 2023; Ramsauer et al., 2020). Notably, when $\alpha = 2$, the results of Theorem 3.1 and Lemma 3.4 reduce to those of (Hu et al., 2023).

## 3.2 GENERALIZED SPARSE HOPFIELD (GSH) LAYERS FOR DEEP LEARNING

Now we introduce the Generalized Sparse Hopfield (GSH) layers for deep learning, by drawing the connection between the generalized sparse Hopfield model and attention mechanism.

**Generalized Sparse Hopfield (GSH) Layer.** Following (Hu et al., 2023), we extend (3.4) to multiple queries $\mathbf{X} := \{\mathbf{x}_i\}_{i \in [T]}$. From previous section, we say that the Hopfield model, as defined by (3.2) and (3.4), functions within the associative spaces $\mathbf{X}$ and $\boldsymbol{\Xi}$. Given any *raw* query $\mathbf{R}$ and memory $\mathbf{Y}$ that are input into the Hopfield model[2], we compute $\mathbf{X}$ and $\boldsymbol{\Xi}$ as $\mathbf{X}^{\mathsf{T}} = \mathbf{R}\mathbf{W}_Q := \mathbf{Q}$ and $\boldsymbol{\Xi}^{\mathsf{T}} = \mathbf{Y}\mathbf{W}_K := \mathbf{K}$, using matrices $\mathbf{W}_Q$ and $\mathbf{W}_K$. Therefore, we rewrite $\mathcal{T}$ in (3.4) as $(\mathbf{Q}^{\text{new}})^{\mathsf{T}} = \mathbf{K}^{\mathsf{T}} \alpha\text{-EntMax}\left(\beta\mathbf{K}\mathbf{Q}^{\mathsf{T}}\right)$. Taking transpose and projecting $\mathbf{K}$ to $\mathbf{V}$ with $\mathbf{W}_V$, we have

$$\mathbf{Z} := \mathbf{Q}^{\text{new}}\mathbf{W}_V = \alpha\text{-EntMax}\left(\beta\mathbf{Q}\mathbf{K}^{\mathsf{T}}\right)\mathbf{K}\mathbf{W}_V = \alpha\text{-EntMax}\left(\beta\mathbf{Q}\mathbf{K}^{\mathsf{T}}\right)\mathbf{V}, \qquad (3.7)$$

which leads to the attention mechanism with $\alpha$-EntMax activation function. Plugging back the raw patterns $\mathbf{R}$ and $\mathbf{Y}$, we arrive the foundation of the Generalized Sparse Hopfield (GSH) layer,

$$\text{GSH}\left(\mathbf{R}, \mathbf{Y}\right) = \mathbf{Z} = \alpha\text{-EntMax}\left(\beta\mathbf{R}\mathbf{W}_Q\mathbf{W}_K^{\mathsf{T}}\mathbf{Y}^{\mathsf{T}}\right)\mathbf{Y}\mathbf{W}_K\mathbf{W}_V. \qquad (3.8)$$

By (3.5), $\mathcal{T}$ retrieves memory patterns with high accuracy after a single activation. This allows (3.8) to integrate with deep learning architectures just like (Hu et al., 2023; Ramsauer et al., 2020).

**Remark 3.1.** $\alpha$ is a learnable parameter (Correia et al., 2019), enabling GSH to learn input sparsity.

`GSHPooling` **and** `GSHLayer` **Layers.** Following (Hu et al., 2023), we introduce two more variants: the `GSHPooling` and `GSHLayer` layers. They are similar to the `GSH`, and only differ in how to obtain the associative sets $\mathbf{Q}, \mathbf{Y}$. For `GHSPooling`$(\mathbf{Y})$, $\mathbf{K} = \mathbf{Y}\mathbf{W}_K$, $\mathbf{V} = \mathbf{K}\mathbf{W}_V$, and $\mathbf{Q}$ is a learnable variable independent from any input. For `GSHLayer`$(\mathbf{R}, \mathbf{Y})$, we have $\mathbf{K} = \mathbf{V} = \mathbf{Y}$, and $\mathbf{Q} = \mathbf{R}$. Note that `GSHLayer` can have $\mathbf{Q}$ as learnable parameter or as an input. Where if $\mathbf{Q}$ was served as an input, the whole `GSHLayer` has no learnable parameters and can be used as a lookup table. We provide an example of memory retrieval for image completion using `GSHLayer` in Appendix D.3.

## 4 STANHOP-NET: SPARSE TANDEM HOPFIELD NETWORK

In this section, we introduce a Hopfield-based deep architecture (STanHop-Net) tailored for memory-enhanced learning of noisy multivariate time series. These additional memory-enhanced functionalities enable STanHop-Net to effectively handle the problem of slow response to sudden or rare events (e.g, 2021 pandemic meltdown in financial market) by making predictions using both in-context inputs (e.g., historical data) and external stimuli (e.g., real-time or relevant past data). In the following, we consider multivariate time series $\mathbf{X} \in \mathbb{R}^{C \times T \times d}$ comprised of $C$ univariate series. Each univariate series has $T$ time steps and $d$ features.

### 4.1 PATCHED EMBEDDING

Motivated by (Zhang and Yan, 2023), we use a patching technique on model input that groups adjacent time steps into subseries patches. This method extends the input time horizon without altering token length, enabling us to capture local semantics and critical information more effectively, which is often missed at the point-level. We define the multivariate input sequence as $\mathbf{X} \in \mathbb{R}^{C \times T \times d}$, where $C, T, d$ denotes the number of variates, number of time steps and the number of dimensions of each variate. Given a time series sequence $\mathbf{X} = \{\mathbf{x}_1, ..., \mathbf{x}_T\}$ and a patch size $P$, the patching operation divides $X$ into $\mathbf{S} = \{\mathbf{s}_1, ..., \mathbf{s}_{T/P}\}$. For each patched sequence $\mathbf{s}_i \in \mathbb{R}^{C \times P \times d}$ for $i \in [T/P]$, we define the patched embedding as $\text{EMB}\left(\mathbf{s}_i\right) = \mathbf{E}^{\text{emb}}\mathbf{s}_i + \mathbf{E}^{\text{pos}}\left(i\right) \in \mathbb{R}^{D_{\text{emb}}}$, where $D_{\text{emb}}$ is the embedding dimension, $\mathbf{E}^{\text{emb}} \in \mathbb{R}^{D_{\text{emb}} \times P}$, and $\mathbf{E}^{\text{pos}} \in \mathbb{R}^{T/P \times D_{\text{emb}}}$ is the positional encoding. When $T$ is

---

[2]The *raw* query $\mathbf{R}$ and memory $\mathbf{Y}$ may originate from data, external sets, or hidden representations throughout a given deep learning pipeline. They are not necessarily usable as $\mathbf{X}$ and $\boldsymbol{\Xi}$. Therefore, to use (3.4), they must be mapped into $d$-dimensional associative spaces.

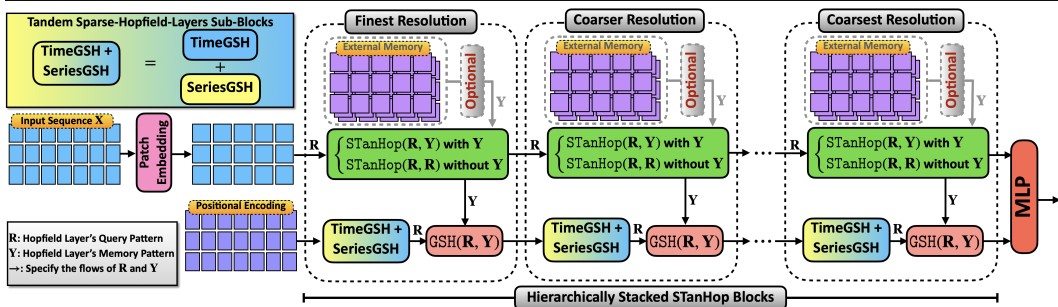

Figure 1: **STanHop-Net Overview. Patch Embedding:** Given an input multivariate time series $\mathbf{X} \in \mathbb{R}^{C \times T \times d}$ consisting $C$ univariate series, $T$ time steps and $d$ features, the patch embedding aggregates temporal information for each univariate series, subsequently reducing temporal dimensionality from $T$ to $P = T/P$ for all $d$ features. **STanHop Block:** The STanHop block leverages the Generalized Sparse Hopfield (GSH) model (Section 3). It captures time series representations from its input through two tandem sparse-Hopfield-layers sub-blocks (i.e. TimeGSH and SeriesGSH, see Figure 2), catering to both temporal and cross-series dimensions. **STanHop-Net:** Using a stacked encoder-decoder structure, STanHop-Net facilitates hierarchical multi-resolution learning. This design allows STanHop-Net to extract distill representations from both temporal and cross-series dimensions across multiple scales (multi-resolution in a hardwired fashion via coarse-graining layers, see Section 4.4). Moreover, each stacked block has optional external memory plugin functionalities for enhanced predictions (Section 4.3). These representations from all resolutions are then merged, providing a holistic representation learning for downstream predictions specially tailored for time series data.

not divisible to $P$, assuming $T = P \times C_n + c$ with $C_n, c \in \mathbb{N}_+$, we pad the sequence by repeating the first $c$ elements in the sequence. Consequently, this patching embedding significantly improves computational efficiency and memory usage.

## 4.2 STanHop: Sparse Tandem Hopfield Block

We introduce STanHop (**S**parse **Tan**dem **Hop**field) block which comprises one GSHLayer-based external memory plugin module, and two tandem sub-blocks of GSH layers to process both time and series dimensions, i.e. TimeGSH and SeriesGSH sub-blocks in Figure 2. In essence, STanHop not only sequentially extracts temporal and cross-series information of multivariate time series with (learnable) data-dependent sparsity, but also utilizes both acquired (in-context) representations and external stimulus through the memory plugin modules for the downstream prediction tasks.

Given a hidden vector, $\mathbf{R} \in \mathbb{R}^{C \times T \times D_{\text{hidden}}}$, and its corresponding external memory set $\mathbf{Y} \in \mathbb{R}^{M \times C \times T \times D_{\text{hidden}}}$, where $C$ denotes the channel number and $T$ denotes the number of time segments (patched time steps), To clarify, the GSH layer only operates on the last two dimensions, i.e., $t \in T$ and $d \in D_{\text{hidden}}$. Thus, the operation $\text{GSH}(\mathbf{Z}, \mathbf{Z})$ extracts information of the temporal dynamics of $\mathbf{Z}$ from the segmented time series. Here we define the dimensional transpose operation $\mathsf{T}$. For a given tensor $\mathbf{X} \in \mathbb{R}^{a \times b \times c}$, we have $\mathsf{T}^{acb}_{abc}(\mathbf{X}) \coloneqq \mathbf{X}' \in \mathbb{R}^{a \times c \times b}$, i.e. this operation rearranges the dimensions of the original tensor $\mathbf{X}$ from $(a, b, c)$ to a new order $(a, c, b)$. Given a set of query pattern $\mathbf{Q} \in \mathbb{R}^{\text{len}_Q \times D_{\text{hidden}}}$, we define a single block of STanHop as

$$\mathbf{Z} = \text{Memory}(\mathbf{R}, \mathbf{Y}), \qquad \text{(Memory Plugin Module, see Section 4.3)}$$

$$\mathbf{Z}^t = \mathsf{T}^{cth}_{tch}(\text{LayerNorm}(\mathbf{Z} + \text{FF}(\text{GSH}(\mathbf{Z}, \mathbf{Z})))) \in \mathbb{R}^{T \times C \times D_{\text{hidden}}}, \qquad \text{(Temporal GSH)}$$

$$\mathbf{Z}^p = \text{GSHPooling}(\mathbf{R}^\star, \mathbf{Z}^t) \in \mathbb{R}^{T \times \text{len}_Q \times D_{\text{hidden}}}, \qquad (\mathbf{R}^\star \text{ is learnable and randomly initialized})$$

$$\mathbf{Z}^c = \text{GSH}(\mathbf{Z}^t, \mathbf{Z}^p) \in \mathbb{R}^{T \times C \times D_{\text{hidden}}}, \qquad \text{(Cross-series GSH)}$$

$$\mathbf{Z}^* = \text{LayerNorm}(\mathbf{Z}^t + \text{FF}(\mathbf{Z}^c)) \in \mathbb{R}^{T \times C \times D_{\text{hidden}}},$$

$$\mathbf{Z}_{\text{out}} = \text{LayerNorm}(\mathbf{Z}^* + \text{FF}(\mathbf{Z}^*)) \in \mathbb{R}^{T \times C \times D_{\text{hidden}}},$$

where $\text{Memory}(\cdot, \cdot)$ is the external memory plugin module introduced in the next section. Note that, if we choose to turn off the external memory functionalities (or external memory is not available) during training, we set $\mathbf{Y} = \mathbf{R}$ such that $\text{Memory}(\mathbf{R}, \mathbf{R}) = \mathbf{R}$ (see Section 4.3 for details). Here $\text{GSHPooling}(\mathbf{R}^\star, \mathbf{Z}^t)$ takes $\mathbf{Z}^t$ and a randomly initialized query $\mathbf{R}^\star$ as input. Importantly, $\mathbf{R}^\star$ not only acts as learnable prototype patterns learned by pooling over $\mathbf{Z}^t$, but also as a knob to control the computational complexity by picking the hidden dimension of $\mathbf{R}^\star$. We summarize the STanHop block as $\mathbf{Z}_{\text{out}} = \text{STanHop}(\mathbf{R}, \mathbf{Y}) \in \mathbb{R}^{T \times C \times D_{\text{hidden}}}$.

## 4.3 External Memory Plugin Module and Pseudo-Label Retrieval

Here we introduce the external memory modules (i.e., $\text{Memory}(\cdot, \cdot)$ in Section 4.2 or Memory Plugin blocks in Figure 2) for external memory functionalities. These modules are tailored for time series modeling by incorporating task-specific supplemental information (such as relevant historical data for sudden or rare events predictions) for subsequent inference. To this end, we introduce two

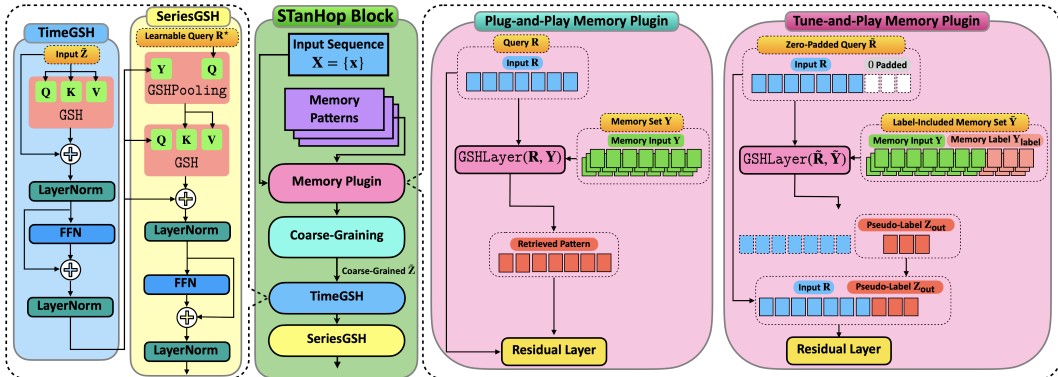

Figure 2: **STanHop Block.** **(Left)** Tandem Hopfield-Layer Blocks: TimeGSH and SeriesGSH. Notably, in the `GSHPooling` block of SeriesGSH, the learnable query $\mathbf{R}^\star$ is initialized randomly and employed to store learned prototype patterns from temporal representations extracted during training. **(Right)** Plug-and-Play and Tune-and-Play Memory Plugins.

memory plugin modules: **Plug-and-Play Memory Plugin** and **Tune-and-Play Memory Plugin**. For query $\mathbf{R}$ and memory $\mathbf{Y}$, we denote them by `PlugMemory`$(\mathbf{R}, \mathbf{Y})$ and `TuneMemory`$(\mathbf{R}, \mathbf{Y})$.

**Plug-and-Play Memory Plugin.** This module enables performance enhancement utilizing external memory without any fine-tuning. Given a trained STanHop-Net (without external memory), we use a parameter fixed `GSHLayer` for memory retrieval. Explicitly, given an input sequence $\mathbf{R} \in \mathbb{R}^{|R| \times D_{\text{hidden}}}$ and a corresponding external memory set $\mathbf{Y} \in \mathbb{R}^{M \times |R| \times d}$, where $|\mathbf{R}|$ and $D_{\text{hidden}}$ are the sequence length and hidden dimension of $\mathbf{R}$ respectively. We define the memory retrieval operation as $\mathbf{Z} = $ `PlugMemory`$(\mathbf{R}, \mathbf{Y}) = $ LayerNorm $(\mathbf{R} + $ `GSHLayer`$(\mathbf{R}, \mathbf{Y}))$ with all parameters fixed.

**Tune-and-Play Memory Plugin.** Here we propose the idea of "pseudo-label retrieval" using `GSHLayer` for time series prediction. Specifically, we use modern Hopfield models' memory retrieval mechanism to generate pseudo-labels for a given $\mathbf{R}$ from a *label-included* memory set $\widetilde{\mathbf{Y}}$, thereby enhancing predictions. Intuitively, this method supplements predictions by learning from demonstrations and we use the retrieved pseudo-labels (i.e., learned *pseudo*-predictions) as additional features. An illustration of this mechanism is shown in Figure 2. Firstly, we prepare the *label-included* external memory as $\widetilde{\mathbf{Y}} = \mathbf{Y} \oplus \mathbf{Y}_{\text{label}}$, where $\widetilde{\mathbf{Y}}$ is the concatenation of memory sequences and their corresponding labels. Next, we denote the padded $\mathbf{R}$ as $\widetilde{\mathbf{R}}$, where $\widetilde{\mathbf{R}} \in \mathbb{R}^{|\widetilde{\mathbf{Y}}| \times d}$. And we utilize the `GSHLayer` to retrieve the pseudo-label from the memory sequences as $\mathbf{Z}_{\text{out}}$. Then we concatenate $\mathbf{R}$ and the pseudo-label $\mathbf{Z}_{\text{out}}$ and send it to a feed forward layer to encode the pseudo-label information: $\mathbf{Z}_{\text{out}} = $ `GSHLayer`$(\widetilde{\mathbf{R}}, \widetilde{\mathbf{Y}})$, $\mathbf{Z}_{\text{pseudo}} = \mathbf{R} \oplus \mathbf{Z}_{\text{out}}$ and then $\widetilde{\mathbf{Z}} = $ LayerNorm $($FF $(\mathbf{Z}_{\text{pseudo}}) + \mathbf{Z}_{\text{pseudo}})$. In other words, we first obtain a weight matrix from the association between $\widetilde{\mathbf{R}}$ and $\widetilde{\mathbf{Y}}$, and then multiply this weight matrix with $\mathbf{Y}_{\text{label}}$ to obtain $\mathbf{Z}_{\text{out}}$. We summarize the Tune-and-Play memory plugin as $\widetilde{\mathbf{Z}} = $ `TuneMemory`$(\mathbf{R}, \mathbf{Y})$.

### 4.4 COARSE-GRAINING

To cope with the intrinsic multi-resolution inductive bias of time series, we introduce a coarse-graining layer in each STanHop block. Given an hidden vector output, $\mathbf{Z} \in \mathbb{R}^{C \times T \times D_{\text{hidden}}}$, grain level $\Delta$, and a weight matrix $\mathbf{W} \in \mathbb{R}^{D_{\text{hidden}} \times 2D_{\text{hidden}}}$, and $\oplus$ denotes the concatenation operation. We denote $\mathbf{Z}_{c,t,d}$ with $c \in [C], t \in [T], d \in [D_{\text{hidden}}]$ as the element representing the $c$-th series, $t$-th time segment, and $d$-th dimension. The coarse-graining layer consists a vector concatenation and a matrix multiplication: $\widehat{\mathbf{Z}}_{c,t,:} = \mathbf{Z}_{c,t,:} \oplus \mathbf{Z}_{c,t+\Delta,:} \in \mathbb{R}^{2D_{\text{hidden}}}$ and then $\widetilde{\mathbf{Z}}_{c,t,:} = \mathbf{W}\widehat{\mathbf{Z}}_{c,t,:} \in \mathbb{R}^{D_{\text{hidden}}}$, such that $\widehat{\mathbf{Z}} \in \mathbb{R}^{C \times T \times 2D_{\text{hidden}}}$ and $\widetilde{\mathbf{Z}} \in \mathbb{R}^{C \times T \times D_{\text{hidden}}}$, similar to (Liu et al., 2021b; Zhang and Yan, 2023). Operationally, it first obtains the representation of smaller resolution, and then distills information via a linear transformation. We express this course-graining layer as CoarseGrain $(\mathbf{Z}, \Delta) = \widetilde{\mathbf{Z}}$.

### 4.5 MULTI-LAYER STANHOP FOR MULTI-RESOLUTION LEARNING

Finally, we construct the STanHop-Net by stacking STanHop blocks in a hierarchical fashion, enabling multi-resolution feature extraction with resolution-specific sparsity. Given a prediction window size $P \in \mathbb{R}$, number of layer $L \in \mathbb{R}$, and a learnable positional embedding for the decoder $\mathbf{E}_{\text{dec}}$, we construct our multi-layer STanHop as an autoencoder structure. The encoder structure consists of a course-graining operation first, following by an `STanHop` layer. The decoder follows the similar structure as the standard transformer decoder (Vaswani et al., 2017), but we replace the cross-attention mechanism to a `GSH` layer, and self-attention layer as `STanHop` layer. We summarize the STanHop-Net network structure in Figure 1, and in Algorithm 2 in appendix.

Table 1: **Accuracy Comparison for Multivariate Time Series Predictions without External Memory.** We implement 3 STanHop variants, **STanHop-Net (D)** with **D**ense Hopfield layer (Ramsauer et al., 2020), **STanHop-Net (S)** with **S**parse SparseHopfield layer (Hu et al., 2023) and **STanHop-Net** with our GSH layer respectively. We report the average Mean Square Error (MSE) and Mean Absolute Error (MAE) metrics of 10 runs, with variance omitted as they are all $\leq 0.44\%$. We benchmark our method against leading transformer-based methods (FEDformer (Zhou et al., 2022), Informer (Zhou et al., 2021) and Autoformer (Wu et al., 2021), Crossformer (Zhang and Yan, 2022)) and a linear model with seasonal-trend decomposition (DLinear (Zeng et al., 2023)). We evaluate each dataset with different prediction horizons (showed in the second column). We have the best results **bolded** and the second best results underlined. In 47 out of 58 settings, STanHop-Nets rank either first or second. Our results indicate that our proposed STanHop-Net delivers consistent top-tier performance compared to all the baselines, even without external memory.

| Models | | FEDFormer | | DLinear | | Informer | | Autoformer | | Crossformer | | STanHop-Net (D) | | STanHop-Net (S) | | STanHop-Net | |
|---|---|---|---|---|---|---|---|---|---|---|---|---|---|---|---|---|---|
| Metric | | MSE | MAE | MSE | MAE | MSE | MAE | MSE | MAE | MSE | MAE | MSE | MAE | MSE | MAE | MSE | MAE |
| ETTh1 | 24 | 0.318 | 0.384 | 0.312 | 0.355 | 0.577 | 0.549 | 0.439 | 0.440 | 0.305 | 0.367 | 0.301 | 0.363 | 0.298 | 0.360 | **0.294** | **0.351** |
| | 48 | 0.342 | 0.396 | 0.352 | **0.383** | 0.685 | 0.625 | 0.429 | 0.442 | 0.352 | 0.394 | 0.356 | 0.406 | 0.355 | 0.399 | **0.340** | 0.387 |
| | 168 | 0.412 | 0.449 | 0.416 | **0.430** | 0.931 | 0.752 | 0.493 | 0.479 | 0.410 | 0.441 | **0.398** | 0.440 | 0.419 | 0.458 | **0.398** | 0.437 |
| | 336 | 0.456 | 0.474 | 0.450 | **0.452** | 1.128 | 0.873 | 0.509 | 0.492 | **0.440** | 0.461 | 0.458 | 0.472 | 0.484 | 0.484 | 0.450 | 0.472 |
| | 720 | 0.521 | 0.515 | **0.484** | **0.501** | 1.215 | 0.896 | 0.539 | 0.537 | 0.519 | 0.524 | 0.516 | 0.522 | 0.541 | 0.533 | 0.512 | 0.511 |
| ETTm1 | 24 | 0.290 | 0.364 | 0.217 | 0.289 | 0.323 | 0.369 | 0.410 | 0.428 | 0.211 | 0.293 | 0.205 | 0.278 | **0.191** | **0.270** | 0.195 | 0.273 |
| | 48 | 0.342 | 0.396 | 0.278 | 0.330 | 0.494 | 0.503 | 0.483 | 0.464 | 0.300 | 0.352 | 0.303 | 0.340 | 0.293 | 0.341 | **0.270** | 0.333 |
| | 96 | 0.366 | 0.412 | 0.310 | 0.354 | 0.678 | 0.614 | 0.502 | 0.476 | 0.320 | 0.373 | 0.325 | 0.377 | 0.322 | 0.362 | **0.286** | **0.352** |
| | 288 | 0.398 | 0.433 | **0.369** | **0.386** | 1.056 | 0.786 | 0.604 | 0.522 | 0.404 | 0.427 | 0.410 | 0.429 | 0.395 | 0.413 | 0.366 | 0.399 |
| | 672 | 0.455 | 0.464 | **0.416** | **0.417** | 1.192 | 0.926 | 0.607 | 0.530 | 0.569 | 0.528 | 0.574 | 0.516 | 0.556 | 0.510 | 0.400 | 0.431 |
| ECL | 48 | 0.229 | 0.338 | 0.155 | 0.258 | 0.344 | 0.393 | 0.241 | 0.351 | 0.156 | 0.255 | 0.159 | 0.264 | 0.170 | 0.273 | **0.152** | **0.252** |
| | 168 | 0.263 | 0.361 | **0.195** | **0.287** | 0.368 | 0.424 | 0.299 | 0.387 | 0.231 | 0.309 | 0.296 | 0.368 | 0.288 | 0.373 | 0.227 | 0.304 |
| | 336 | 0.305 | 0.386 | **0.238** | **0.316** | 0.381 | 0.431 | 0.375 | 0.428 | 0.323 | 0.369 | 0.326 | 0.374 | 0.317 | 0.375 | 0.317 | 0.361 |
| | 720 | 0.372 | 0.434 | **0.272** | **0.346** | 0.406 | 0.443 | 0.377 | 0.434 | 0.404 | 0.423 | 0.412 | 0.428 | 0.440 | 0.450 | 0.405 | 0.416 |
| | 960 | 0.393 | 0.449 | **0.299** | **0.367** | 0.460 | 0.548 | 0.366 | 0.426 | 0.433 | 0.438 | 0.446 | 0.447 | 0.467 | 0.463 | 0.430 | 0.431 |
| WTH | 24 | 0.357 | 0.412 | 0.357 | 0.391 | 0.335 | 0.381 | 0.363 | 0.396 | 0.294 | 0.343 | 0.304 | 0.351 | 0.303 | 0.352 | **0.292** | **0.341** |
| | 48 | 0.428 | 0.458 | 0.425 | 0.444 | 0.395 | 0.459 | 0.456 | 0.462 | 0.370 | 0.411 | 0.374 | 0.411 | 0.372 | 0.411 | **0.363** | **0.402** |
| | 168 | 0.564 | 0.541 | 0.516 | 0.516 | 0.608 | 0.567 | 0.574 | 0.548 | 0.473 | 0.494 | 0.480 | 0.501 | 0.496 | 0.511 | **0.332** | **0.393** |
| | 336 | 0.533 | 0.536 | 0.536 | 0.537 | 0.702 | 0.620 | 0.600 | 0.571 | **0.495** | **0.515** | 0.507 | 0.526 | 0.514 | 0.530 | 0.499 | 0.515 |
| | 720 | 0.562 | 0.557 | 0.582 | 0.571 | 0.831 | 0.731 | 0.587 | 0.570 | **0.526** | **0.542** | 0.545 | 0.557 | 0.548 | 0.556 | 0.533 | 0.546 |
| ILI | 24 | 2.687 | 1.147 | 2.940 | 1.205 | 4.588 | 1.462 | 3.101 | 1.238 | 3.041 | 1.186 | 3.305 | 1.241 | 3.194 | 1.176 | 3.121 | **1.139** |
| | 36 | 2.887 | **1.160** | 2.826 | 1.184 | 4.845 | 1.496 | 3.397 | 1.270 | 3.406 | 1.232 | 3.542 | 1.314 | 3.193 | 1.169 | 3.288 | 1.182 |
| | 48 | 2.797 | 1.155 | **2.677** | 1.155 | 4.865 | 1.516 | 2.947 | 1.203 | 3.459 | 1.221 | 3.409 | 1.208 | 3.15 | 1.142 | 3.122 | **1.120** |
| | 60 | **2.809** | 1.163 | 3.011 | 1.245 | 5.212 | 1.576 | 3.019 | 1.202 | 3.640 | 1.305 | 3.668 | 1.269 | 3.43 | 1.196 | 3.416 | **1.180** |
| Traffic | 24 | 0.562 | 0.375 | **0.351** | 0.261 | 0.608 | 0.334 | 0.550 | 0.363 | 0.491 | 0.271 | 0.484 | 0.266 | 0.499 | 0.277 | 0.452 | **0.247** |
| | 48 | 0.567 | 0.374 | 0.370 | 0.270 | 0.644 | 0.359 | 0.595 | 0.376 | 0.519 | 0.295 | 0.516 | 0.293 | 0.516 | 0.290 | **0.315** | **0.261** |
| | 168 | 0.607 | 0.385 | **0.395** | 0.277 | 0.660 | 0.391 | 0.649 | 0.407 | 0.513 | 0.289 | 0.511 | 0.301 | 0.517 | 0.289 | 0.501 | **0.276** |
| | 336 | 0.624 | 0.389 | **0.415** | 0.289 | 0.747 | 0.405 | 0.624 | 0.388 | 0.530 | 0.300 | 0.531 | 0.316 | 0.544 | 0.303 | 0.506 | **0.288** |
| | 720 | 0.623 | 0.378 | **0.455** | 0.313 | 0.792 | 0.430 | 0.674 | 0.417 | 0.573 | 0.313 | 0.569 | 0.303 | 0.563 | 0.311 | 0.539 | **0.300** |

# 5 EXPERIMENTAL STUDIES

We demonstrate the validity of STanHop-Net and external memory modules by testing them on various experimental settings with both synthetic and real-world datasets.

## 5.1 MULTIVARIATE TIME SERIES PREDICTION WITHOUT EXTERNAL MEMORY

Table 1 includes the experiment results of the multivariate time series predictions using STanHop-Net without external memory. We implement three variants of STanHop-Net: **StanHop-Net**, **StanHop-Net (D)** and **StanHop-Net (S)**, with GSH, Hopfield (Ramsauer et al., 2020) and SparseHopfield (Hu et al., 2023) layers respectively. Our results show that in 47 out of 58 cases, STanHop-Nets rank in the top two, delivering top-tier performance compared to all baselines.

**Data.** Following (Zhang and Yan, 2023; Zhou et al., 2022; Wu et al., 2021), we use 6 realistic datasets: ETTh1 (Electricity Transformer Temperature-hourly), ETTm1 (Electricity Transformer Temperature-minutely), WTH (Weather), ECL (Electricity Consuming Load), ILI (Influenza-Like Illness), Traffic. The first four datasets are split into train/val/test ratios of 14/5/5, and the last two are split into 7/1/2. **Metrics.** We use Mean Square Error (MSE) and Mean Absolute Error (MAE) as accuracy metrics. **Setup.** Here we use the same setting as in (Zhang and Yan, 2022): multivariate time series predictions tasks on 6 real-world datasets. For each dataset, we evaluate our models with several different prediction horizons. For all experiments, we report the mean MSE, MAE over 10 runs. **Baselines.** We benchmark our method against 5 leading methods listed in Table 1. Baseline results are quoted from competing papers when possible and reproduced otherwise. **Hyperparameters.** For each experiment, we optimize the hyperparameters using the "sweep" function from Weights and Biases (Biewald et al., 2020). We conduct 100 random search iterations for each setting, selecting the best set based on the validation performance.

For datasets, hyperparameter tuning, implementations and training details, please see Appendix F.

## 5.2 MEMORY-ENHANCED PREDICTION: MEMORY PLUGIN VIA HOPFIELD LAYER

In Table 2 and Figure 3, we showcase STanHop-Net with external memory enhancements delivers performance boosts in many scenarios. The external memory enhancements support two plugin schemes, **Plug-and-Play** and **Tune-and-Play**. They focus on different benefits. TuneMemory is es-

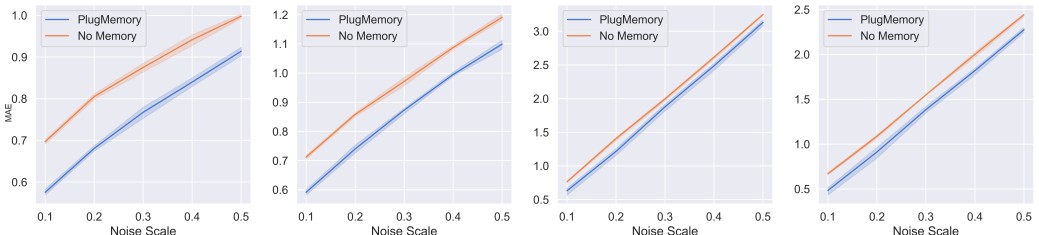

Figure 3: **Visualization of Memory Plugin Scenarios Case 3 & 4. From Left to Right:** MAE against different noise levels with (1) ETTh1 + prediction horizon 336; (2) ETTh1 + prediction horizon 168; (3) ETTm1 + prediction horizon 288; and (4) ETTm1 + prediction horizon 96. The results show the robustness of `PlugMemory` against different level of noise.

pecially useful for task-relevant knowledge incorporation by fine-tuning on an external task-relevant memory set[3]. On the other hand, `PlugMemory` provides a more robust representation of inputs with high uncertainty by doing a retrieval (Figure 2) on an external task-relevant memory set, without the work of any training or fine-tuning. Below we provide 4 practical scenarios to showcase the aforementioned benefits of `TuneMemory` and `PlugMemory` external memory modules. The detailed setups of each case can be found in the appendix.

**Case 1** (`TuneMemory`). We take the single variate, *Number of Influenza incidence in a week* (denoted as ILI OT), from the **ILI** dataset as a straightforward example. In this dataset, we are aware of the existence of recurring annual patterns, which can be readily identified through visualizations in Figure 18. Notably, the signal patterns around the spring of 2014 closely resemble past springs. Thus, in predictions tasks with input located in the yearly recurring period, we collect similar patterns from the past to form a task-relevant external memory set.

**Case 2** (`TuneMemory`). In many sociological studies (Wang et al., 2021a;b), electricity usage exhibits consistent patterns across different regions, influenced by the daily and weekly routines of residents and local businesses. Thus, we collect sequences that match the length of the input sequence but are from 1 to 20 weeks prior, obtaining a task-relevant external memory set of size 20.

In addition, we also include analysis of **"bad" external memory sets**, to verify the effectiveness of incorporating informative external memory sets. We construct the "bad" external memory sets by randomly selecting from dataset without any task-relevant preference, see Appendix F.2 for more details about such selection. The results indicate that, by properly selecting external memory sets, we further improve the models' performance. On the contrary, randomly chosen external memory sets can negatively impact performance. We report the results of Case 1 and Case 2 in Table 2.

**Case 3** (`PlugMemory`). Through `PlugMemory`, informative patterns can be extracted from a memory set for the given noisy input. To verify this ability, we construct the external memory sets based on the weekly pattern spotted in ETTh1 and ETTm1, and add noise of different scales into the input sequence. We add the noise following $x \leftarrow x + \text{scale} \cdot \text{std}(x)$. For Case 3, we use the ETTh1 dataset. **Case 4** (`PlugMemory`). For Case 4, we evaluate `PlugMemory` on the ETTm1 dataset.

Table 2: **Performance Comparison of the StanHop Model with `TuneMemory` and Ablation Using Bad External Memory Sets** (`TuneMemory`-(b)). We report the mean MSE and MAE over 10 runs with variances omitted as they are $\leq 0.79\%$. For ILI OT, we consider prediction horizons of 12, 24, and 60. For ETTh1, we choose prediction horizons of 24, 48, and 720, covering both short and long durations. The results indicate that using dataset insights and `TuneMemory` enhances our model's performance.

| | Case 1 (ILI OT) | | | | | | | Case 2 (ETTh1) | | | | | |
| | Default | | TuneMemory | | TuneMemory-(b) | | | Default | | TuneMemory | | TuneMemory-(b) | |
| | MSE | MAE | MSE | MAE | MSE | MAE | | MSE | MAE | MSE | MAE | MSE | MAE |
|---|---|---|---|---|---|---|---|---|---|---|---|---|---|
| 12 | 4.011 | 1.701 | 3.975 (-0.9%) | 1.693 (-0.5%) | 4.340 (+8.2%) | 1.789 (+5.1%) | 24 | 0.294 | 0.351 | 0.284 (-3.4%) | 0.351 (±0%) | 0.300 (+2%) | 0.361 (+2.8%) |
| 24 | 4.254 | 1.771 | 3.960 (-6.9%) | 1.690 (-4.6%) | 4.271 (+0.4%) | 1.776 (+0.3%) | 48 | 0.340 | 0.387 | 0.328 (-3.5%) | 0.379 (-2.1%) | 0.342 (+0.6%) | 0.388 (+0.3%) |
| 60 | 3.613 | 1.685 | 3.572 (-1.1%) | 1.528 (-9.3%) | 3.821 (+5.8%) | 1.725 (+2.4%) | 720 | 0.512 | 0.511 | 0.504 (-1.6%) | 0.512 (-0.2%) | 0.514 (+0.4%) | 0.521 (+2.0%) |

# 6 CONCLUSION

We propose the generalized sparse modern Hopfield model and present STanHop-Net, a Hopfield-based time series prediction model with external memory functionalities. Our design improves time series forecasting performance, quickly reacts to unexpected or rare events, and offers both strong theoretical guarantees and empirical results. Empirically, STanHop-Nets rank in the top two in 47 out of our 58 experiment settings compared to the baselines. Furthermore, with `PlugMemory` and `TuneMemory` modules, it showcases average performance boosts of $\sim$12% and $\sim$3% for each.

---

[3]Task-relevant means the relevance to the inputs of the time series forecasting. A task-relevant memory set could be a set of some history time series segments that are relevant to the inputs of the prediction.

**Note Added [December 27, 2023].**    During ICLR'24's rebuttal period, the authors learn of an upcoming work by Martins et al. (2023), addressing similar topics from the perspective of the Fenchel-Young loss (Blondel et al., 2020). The authors would like to thank Martins et al. (2023) for reading our manuscript and testing our code.

## ACKNOWLEDGMENTS

JH would like to thank Feng Ruan, Dino Feng and Andrew Chen for enlightening discussions, Pei-Hsuan Chang for pointing out typos in proofs, the Red Maple Family for support, and Jiayi Wang for facilitating experimental deployments. The authors would also like to thank the anonymous reviewers and program chairs for their constructive comments.

JH is partially supported by the Walter P. Murphy Fellowship. BY is supported by the National Taiwan University Fu Bell Scholarship. HL is partially supported by NIH R01LM1372201, NSF CAREER1841569, DOE DE-AC02-07CH11359, DOE LAB 20-2261 and a NSF TRIPODS1740735. This research was supported in part through the computational resources and staff contributions provided for the Quest high performance computing facility at Northwestern University which is jointly supported by the Office of the Provost, the Office for Research, and Northwestern University Information Technology. The content is solely the responsibility of the authors and does not necessarily represent the official views of the funding agencies.

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

# Supplementary Material

## A  BROADER IMPACTS

We envision this approach as a means to refine large foundation models for time series, through a perspective shaped by neuroscience insights. Such memory-enhanced time series foundation models are vital in applications like eco- and climatic-modeling. For example, with a multi-modal time series foundation model, we can effectively predict, detect, and mitigate emerging biological threats associated with the rapid changes in global climate. To this end, the differentiable external memory modules become handy, as they allow users to integrate real-time data into pre-trained foundation models and thus enhance the model's responsiveness in real-time scenarios. Specifically, one can use this memory-enhanced technique to embed historical, sudden, or rare events into any given time series foundation model, thereby boosting its overall performance.

## B  RELATED WORKS AND LIMITATIONS

**Transformers for Time Series Prediction.**  As suggested in Section 3 and (Hu et al., 2023; Ramsauer et al., 2020), besides the additional memory functionalities, the Hopfield layers act as promising alternatives for the attention mechanism. Therefore, we discuss and compare STanHop-Net with existing transformer-based time series prediction methods here.

Transformers have gained prominence in time series prediction, inspired by their success in Natural Language Processing and Computer Vision. One challenge in time series prediction is managing transformers' quadratic complexity due to the typically long sequences. To address this, many researchers have not only optimized for prediction performance but also sought to reduce memory and computational complexity. LogTrans (Li et al., 2019) proposes a transformer-based neural network for time series prediction. They propose a convolution layer over the vanilla transformer to better capture local context information and a sparse attention mechanism to reduce memory complexity. Similarly, Informer (Zhou et al., 2021) proposes convolutional layers in between attention blocks to distill the dominating attention and a sparse attention mechanism where the keys only attend to a subset of queries. Reformer (Kitaev et al., 2020) replaces the dot-product self-attention in the vanilla transformer with a hashing-based attention mechanism to reduce the complexity. Besides directly feeding the raw time series inputs to the model, many works focus on transformer-based time series prediction by modeling the decomposed time series. Autoformer (Wu et al., 2021) introduces a series decomposition module to its transformer-based model to separately model the seasonal component and the trend-cyclical of the time series. FEDformer (Zhou et al., 2022) also models the decomposed time series and they introduce a block to extract signals by transforming the time series to the frequency domain.

Compared to STanHop, the above methods do not model multi-resolution information. Besides, Reformer's attention mechanism sacrifices the global receptive field compared to the vanilla self-attention mechanism and our method, which harms the prediction performance.

Some works intend to model the multi-resolution or multi-scale signals in the time series with a dedicated network design. Pyraformer (Liu et al., 2021a) designs a pyramidal attention module to extract the multi-scale signals from the raw time series. Crossformer (Zhang and Yan, 2022) proposes a multi-scale encoder-decoder architecture to hierarchically extract signals of different resolutions from the time series. Compared to these methods. STanHop adopts a more fine-grained multi-resolution modeling mechanism that is capable of learning different sparsity levels for signals in the data of different resolutions.

Furthermore, all of the above works on time series prediction lack the external memory retrieval module as ours. Thus, our STanHop method and its variations have a unique advantage in that we have a fast response to real-time unexpected events.

**Hopfield Models and Deep Learning.**  Hopfield Models (Hopfield, 1984; 1982; Krotov and Hopfield, 2016) have garnered renewed interest in the machine learning community due to the connection between their memory retrieval dynamics and attention mechanisms in transformers via the Modern Hopfield Models (Wu et al., 2024; Hu et al., 2024b; 2023; Ramsauer et al., 2020). Furthermore, these modern Hopfield models enjoy superior empirical performance and possess several appealing theoretical properties, such as rapid convergence and guaranteed exponential memory capacity. By viewing modern Hopfield models as generalized attentions with enhanced memory functionali-

ties, these advancements pave the way for innovative Hopfield-centric architectural designs in deep learning (Hu et al., 2024a; Hoover et al., 2023; Seidl et al., 2022; Fürst et al., 2022; Ramsauer et al., 2020). Consequently, their applicability spans diverse areas like physics (Krotov, 2023), biology (Schimunek et al., 2023; Kozachkov et al., 2023; Widrich et al., 2020), reinforcement learning (Paischer et al., 2022), tabular learning (Xu et al., 2024), and large language models (Hu et al., 2024a; Fürst et al., 2022).

This work pushes this line of research forward by presenting a Hopfield-based deep architecture (StanHop-Net) tailored for memory-enhanced learning in noisy multivariate time series. In particular, our model emphasizes in-context memorization during training and bolsters retrieval capabilities with an integrated external memory component.

**Sparse Modern Hopfield Model.** Our work extends the theoretical framework proposed in (Hu et al., 2023) for modern Hopfield models. Their primary insight is that using different entropic regularizers can lead to distributions with varying sparsity. Using the Gibbs entropic regularizer, they reproduce the results of the standard dense Hopfield model (Ramsauer et al., 2020) and further propose a sparse variant with the Gini entropic regularizer, providing improved theoretical guarantees. However, their sparse model primarily thrives with data of high intrinsic sparsity. To combat this, we enrich the link between Hopfield models and attention mechanisms by introducing *learnable sparsity* and showing that the sparse model from (Hu et al., 2023) is a specific case of our model when setting $\alpha = 2$. Unlike (Hu et al., 2023), our generalized sparse Hopfield model ensures adaptable sparsity across various data types without sacrificing theoretical integrity. By making this sparsity learnable, we introduce the GSH layers. These new Hopfield layers adeptly learn and store sparse representations in any deep learning pipeline, proving invaluable for inherently noisy time series data.

**Memory Augmented Neural Networks.** The integration of external memory mechanisms with neural networks has emerged as a pivotal technique for machine learning, particularly for tasks requiring complex data manipulation and retention over long sequences, such as open question answering, and few-shot learning.

Neural Turing Machines (NTMs) (Graves et al., 2014) combine the capabilities of neural networks with the external memory access of a Turing machine. NTMs use a differentiable controller (typically an RNN) to interact with an external memory matrix through read and write heads. This design allows NTMs to perform complex data manipulations, akin to a computer with a read-write memory. Building upon this, Graves et al. (2016) further improve the concept through Differentiable Neural Computers (DNCs), which enhance the memory access mechanism using a differentiable attention process. This includes a dynamic memory allocation and linkage system that tracks the relationships between different pieces of data in memory. This feature makes DNCs particularly adept at tasks that require complex data relationships and temporal linkages.

Concurrently, Memory Networks (Weston et al., 2014) showcase the significance of external memory in tasks requiring complex reasoning and inference. Unlike traditional neural networks that rely on their inherent weights to store information, Memory Networks incorporate a separate memory matrix that stores and retrieves information across different processing steps. This capability allows the network to maintain and manipulate a "memory" of past inputs and computations, which is particularly crucial for tasks requiring persistent memory, such as question-answering systems where the network needs to remember context or facts from previous parts of a conversation or text to accurately respond to queries. This concept is further developed into the End-to-End Memory Networks (Sukhbaatar et al., 2015), which extend the utility of Memory Networks (Weston et al., 2014) beyond the limitations of traditional recurrent neural network architectures, transitioning them into a fully end-to-end trainable framework, thereby making them more adaptable and easier to integrate into various learning paradigms.

A notable application of memory-augmented neural networks is in the domain of one-shot learning. The concept of meta-learning with memory-augmented neural networks, as explored by Santoro et al. (2016), has demonstrated the potential of these networks to rapidly adapt to new tasks by leveraging their external memory, highlighting their versatility and efficiency in learning. They showcase the potential of these networks to adapt rapidly to new tasks, a crucial capability in scenarios where data availability is limited. Complementing this, Kaiser et al. (2017) focus on enhancing the recall of

rare events and is particularly notable for its exploration of memory-augmented neural networks designed to improve the retention and recall of infrequent but significant occurrences, highlighting the potential of external memory modules in handling rare data challenges. This is achieved through a unique soft attention mechanism, which dynamically assigns relevance weights to different memory entries, enabling the model to draw on a broad spectrum of stored experiences. This approach not only facilitates the effective recall of rare events but also adapts to new data, ensuring the memory's relevance and utility over time.

In all above methods, (Kaiser et al., 2017) is closest to this work. However, our approach diverges in two key aspects: **(Enhanced Generalization):** Firstly, our external memory enhancements are external plugins with an option for fine-tuning. This design choice avoids over-specialization on rare events, thereby broadening our method's applicability and enhancing its generalization capabilities across various tasks where the frequency and recency of data are less pivotal. **(Adaptive Response over Rare Event Memorization):** Secondly, our approach excels in real-time adaptability. By integrating relevant external memory sets tailored to specific inference tasks, our method can rapidly respond and improve performance, even without the necessity of prior learning. This flexibility contrasts with the primary focus on *memorizing* rare events in (Kaiser et al., 2017).

### B.1 LIMITATIONS

The proposed generalized sparse modern Hopfield model shares the similar inefficiency due to the $\mathcal{O}(d^2)$ complexity[4]. In addition, the effectiveness of our memory enhancement methods is contingent on the relevance of the external memory set to the specific inference task. Achieving a high degree of relevance in the external memory set often necessitates considerable human effort and domain expertise, just like our selection process detailed in Appendix F.2. This requirement could potentially limit the model's applicability in scenarios where such resources are scarce or unavailable.

---

[4]As a side note, Hu et al. (2024c) provides a characterization of the computational efficiency of all possible efficient variants of the modern Hopfield model from the fine-grained complexity theory.

## C  Proofs of Main Text

### C.1  Lemma 3.1

Our proof relies on verifying that $\Psi^\star$ meets the criteria of Danskin's theorem.

*Proof of Lemma 3.1.* Firstly, we introduce the notion of convex conjugate.

**Definition C.1.** Let $F(\mathbf{p}, \mathbf{z}) \coloneqq \langle \mathbf{p}, \mathbf{z} \rangle - \Psi^\alpha(\mathbf{p})$. The convex conjugate of $\Psi^\alpha$, $\Psi^\star$ takes the form:

$$\Psi^\star(\mathbf{z}) = \underset{\mathbf{p} \in \Delta^M}{\operatorname{Max}} \langle \mathbf{p}, \mathbf{z} \rangle - \Psi^\alpha(\mathbf{p}) = \underset{\mathbf{p} \in \Delta^M}{\operatorname{Max}} F(\mathbf{p}, \mathbf{z}). \tag{C.1}$$

By Danskin's theorem (Danskin, 2012; Bertsekas et al., 1999), the function $\Psi^\star$ is convex and its partial derivative with respect to $\mathbf{z}$ is equal to that of $F$, i.e. $\partial \Psi^\star / \partial \mathbf{z} = \partial F / \partial \mathbf{z}$, if the following three conditions are satisfied for $\Psi^\star$ and $F$:

(i) $F(\mathbf{p}, \mathbf{z}) : \mathcal{P} \times \mathbb{R}^M \to \mathbb{R}$ is a continuous function, where $\mathcal{P} \subset \mathbb{R}^M$ is a compact set.

(ii) $F$ is convex in $\mathbf{z}$, i.e. for each given $\mathbf{p} \in \mathcal{P}$, the mapping $\mathbf{z} \to F(\mathbf{p}, \mathbf{z})$ is convex.

(iii) There exists an unique maximizing point $\widehat{\mathbf{p}}$ such that $F(\widehat{\mathbf{p}}, \mathbf{z}) = \operatorname{Max}_{\mathbf{p} \in \mathcal{P}} F(\mathbf{p}, \mathbf{z})$.

Since both $\langle \mathbf{p}, \mathbf{z} \rangle$ and $\Psi^\alpha$ are continuous functions and every component of $\mathbf{p}$ is ranging from 0 to 1, the function $F$ is continuous and the domain $\mathcal{P}$ is a compact set. Therefore, condition (i) is satisfied.

Since we require $\mathbf{p} \in \Delta^M$ (i.e. $\mathcal{P} = \Delta^M$) to be probability distributions, for any fixed $\mathbf{p}$, $F(\mathbf{p}, \mathbf{z}) = \langle \mathbf{p}, \mathbf{z} \rangle - \Psi^\alpha(\mathbf{p})$ reduces to an affine function depending only on input $\mathbf{z}$. Due to the inner product form, this affine function is convex in $\mathbf{z}$, and hence condition (ii) holds for all given $\mathbf{p} \in \mathcal{P} = \Delta^M$.

Since, for any given $\mathbf{z}$, $\alpha$-EntMax only produces one unique probability distribution $\mathbf{p}^\star$, condition (iii) is satisfied. Therefore, from Danskin's theorem, it holds

$$\boldsymbol{\nabla}_z \Psi^\star(\mathbf{z}) = \frac{\partial F}{\partial \mathbf{z}} = \frac{\partial}{\partial \mathbf{z}}(\langle \mathbf{p}, \mathbf{z} \rangle - \Psi^\alpha(\mathbf{p})) = \mathbf{p} = \alpha\text{-EntMax}(\mathbf{z}). \tag{C.2}$$

$\square$

### C.2  Lemma 3.2

Our proof is built on (Hu et al., 2023, Lemma 2.1). We first derive $\mathcal{T}$ by utilizing Lemma 3.1 and Remark G.1, along with the convex-concave procedure (Yuille and Rangarajan, 2003; 2001). Then, we show the monotonicity of minimizing $\mathcal{H}$ with $\mathcal{T}$ by constructing an iterative upper bound of $\mathcal{H}$ which is convex in $\mathbf{x}_{t+1}$ and thus, can be lowered iteratively by the convex-concave procedure.

*Proof.* From Lemma 3.1, the conjugate convex of $\Psi$, $\Psi^\star$, is always convex, and, therefore, $-\Psi^\star$ is a concave function. Then, the energy function $\mathcal{H}$ defined in (3.2) is the sum of the convex function $\mathcal{H}_1(\mathbf{x}) \coloneqq \frac{1}{2} \langle \mathbf{x}, \mathbf{x} \rangle$ and the concave function $\mathcal{H}_2(\mathbf{x}) \coloneqq -\Psi^\star \left( \boldsymbol{\Xi}^\mathsf{T} \mathbf{x} \right)$.

Furthermore, by definition, the energy function $\mathcal{H}$ is differentiable.

Every iteration step of convex-concave procedure applied on $\mathcal{H}$ gives

$$\boldsymbol{\nabla}_\mathbf{x} \mathcal{H}_1 \left( \mathbf{x}_{t+1} \right) = -\boldsymbol{\nabla}_\mathbf{x} \mathcal{H}_2 \left( \mathbf{x}_t \right), \tag{C.3}$$

which implies that

$$\mathbf{x}_{t+1} = \boldsymbol{\nabla}_\mathbf{x} \Psi \left( \boldsymbol{\Xi} \mathbf{x}_t \right) = \boldsymbol{\Xi} \, \alpha\text{-EntMax} \left( \boldsymbol{\Xi}^\mathsf{T} \mathbf{x}_t \right). \tag{C.4}$$

On the basis of (Yuille and Rangarajan, 2003; 2001), we show the decreasing property of (3.2) over $t$ via solving the minimization problem of energy function:

$$\underset{\mathbf{x}}{\text{Min}}\left[\mathcal{H}(\mathbf{x})\right] = \underset{\mathbf{x}}{\text{Min}}\left[\mathcal{H}_1(\mathbf{x}) + \mathcal{H}_2(\mathbf{x})\right], \tag{C.5}$$

which, in convex-concave procedure, is equivalent to solve the iterative programming

$$\mathbf{x}_{t+1} \in \underset{\mathbf{x}}{\text{ArgMin}}\left[\mathcal{H}_1(\mathbf{x}) + \langle\mathbf{x}, \boldsymbol{\nabla}_{\mathbf{x}}\mathcal{H}_2(\mathbf{x}_t)\rangle\right], \tag{C.6}$$

for all $t$. The concept behind this programming is to linearize the concave function $\mathcal{H}_2$ around the solution for current iteration, $\mathbf{x}_t$, which makes $\mathcal{H}_1(\mathbf{x}_{t+1}) + \langle\mathbf{x}_{t+1}, \boldsymbol{\nabla}_{\mathbf{x}}\mathcal{H}_2(\mathbf{x}_t)\rangle$ convex in $\mathbf{x}_{t+1}$.

The convexity of $\mathcal{H}_1$ and concavity of $\mathcal{H}_2$ imply that the inequalities

$$\mathcal{H}_1(\mathbf{x}) \geq \mathcal{H}_1(\mathbf{y}) + \langle(\mathbf{x} - \mathbf{y}), \boldsymbol{\nabla}_{\mathbf{x}}\mathcal{H}_1(\mathbf{y})\rangle, \tag{C.7}$$
$$\mathcal{H}_2(\mathbf{x}) \leq \mathcal{H}_2(\mathbf{y}) + \langle(\mathbf{x} - \mathbf{y}), \boldsymbol{\nabla}_{\mathbf{x}}\mathcal{H}_2(\mathbf{y})\rangle, \tag{C.8}$$

hold for all $\mathbf{x}, \mathbf{y}$, which leads to

$$\mathcal{H}(\mathbf{x}) = \mathcal{H}_1(\mathbf{x}) + \mathcal{H}_2(\mathbf{x}) \tag{C.9}$$
$$\leq \mathcal{H}_1(\mathbf{x}) + \mathcal{H}_2(\mathbf{y}) + \langle(\mathbf{x} - \mathbf{y}), \boldsymbol{\nabla}_{\mathbf{x}}\mathcal{H}_2(\mathbf{y})\rangle \coloneqq \mathcal{H}_U(\mathbf{x}, \mathbf{y}), \tag{C.10}$$

where the upper bound of $\mathcal{H}$ is defined as $\mathcal{H}_U$. Then, the iteration (C.6)

$$\mathbf{x}_{t+1} \in \underset{\mathbf{x}}{\text{ArgMin}}\left[\mathcal{H}_U(\mathbf{x}, \mathbf{x}_t)\right] = \underset{\mathbf{x}}{\text{ArgMin}}\left[\mathcal{H}_1(\mathbf{x}) + \langle\mathbf{x}, \boldsymbol{\nabla}_{\mathbf{x}}\mathcal{H}_2(\mathbf{x}_t)\rangle\right], \tag{C.11}$$

can make $\mathcal{H}_U$ decrease iteratively and thus decreases the value of energy function $\mathcal{H}$ monotonically, i.e.

$$\mathcal{H}(\mathbf{x}_{t+1}) \leq \mathcal{H}_U(\mathbf{x}_{t+1}, \mathbf{x}_t) \leq \mathcal{H}_U(\mathbf{x}_t, \mathbf{x}_t) = \mathcal{H}(\mathbf{x}_t), \tag{C.12}$$

for all $t$. Equation (C.10) shows that the retrieval dynamics defined in (3.2) can lead the energy function $\mathcal{H}$ to decrease with respect to the increasing $t$. $\qquad\square$

## C.3   LEMMA 3.3

To prove the convergence property of retrieval dynamics $\mathcal{T}$, first we introduce an auxiliary lemma from (Sriperumbudur and Lanckriet, 2009).

**Lemma C.1** ((Sriperumbudur and Lanckriet, 2009), Lemma 5). Following Lemma 3.3, $\mathbf{x}$ is called the fixed point of iteration $\mathcal{T}$ with respect to $\mathcal{H}$ if $\mathbf{x} = \mathcal{T}(\mathbf{x})$ and is considered as a generalized fixed point of $\mathcal{T}$ if $\mathbf{x} \in \mathcal{T}(\mathbf{x})$. If $\mathbf{x}^\star$ is a generalized fixed point of $\mathcal{T}$, then, $\mathbf{x}^\star$ is a stationary point of the energy minimization problem (C.5).

*Proof.* Since the energy function $\mathcal{H}$ monotonically decreases with respect to increasing $t$ in Lemma 3.2, we can follow (Hu et al., 2023, Lemma 2.2) to guarantee the convergence property of $\mathcal{T}$ by checking the necessary conditions of Zangwill's global convergence. After satisfying these conditions, Zangwill global convergence theory ensures that all the limit points of $\{\mathbf{x}_t\}_{t=0}^\infty$ are generalized fixed points of the mapping $\mathcal{T}$ and it holds $\lim_{t\to\infty}\mathcal{H}(\mathbf{x}_t) = \mathcal{H}(\mathbf{x}^\star)$, where $\mathbf{x}^\star$ are some generalized fixed points of $\mathcal{T}$. Furthermore, auxiliary Lemma C.1 implies that $\mathbf{x}^\star$ are also the stationary points of energy function $\mathcal{H}$. Therefore, we guarantee that $\mathcal{T}$ can iteratively lead the query $\mathbf{x}$ to converge to the local optimum of $\mathcal{H}$. $\qquad\square$

## C.4  THEOREM 3.1

*Proof.* Compared with $\mathcal{T}_{\text{Dense}}$, $\mathcal{T}$ only sample partial memory patterns near $\boldsymbol{\xi}_\mu$, which gives the tighter bound:

$$\|\mathcal{T}(\mathbf{x}) - \boldsymbol{\xi}_\mu\| \leq \|\mathcal{T}_{\text{Dense}}(\mathbf{x}) - \boldsymbol{\xi}_\mu\|. \tag{C.13}$$

**For $2 \geq \alpha \geq 1$:**  Then, we derive the upper bound on $\|\mathcal{T}_{\text{dense}}(\mathbf{x}) - \boldsymbol{\xi}_\mu\|$ based on (Hu et al., 2023, Theorem 2.2):

$$\|\mathcal{T}_{\text{dense}}(\mathbf{x}) - \boldsymbol{\xi}_\mu\| = \left\| \sum_{\nu=1}^{M} [\text{Softmax}(\beta\boldsymbol{\Xi}^\mathsf{T}\mathbf{x})]_\nu \boldsymbol{\xi}_\nu - \boldsymbol{\xi}_\mu \right\| \tag{C.14}$$

$$= \left\| \sum_{\nu=1,\nu\neq\mu}^{M} [\text{Softmax}(\beta\boldsymbol{\Xi}^\mathsf{T}\mathbf{x})]_\nu \boldsymbol{\xi}_\nu - (1 - \text{Softmax}(\beta\boldsymbol{\Xi}^\mathsf{T}\mathbf{x}))\boldsymbol{\xi}_\mu \right\| \tag{C.15}$$

$$\leq 2\widetilde{\epsilon}m, \tag{C.16}$$

where $\widetilde{\epsilon} := (M-1)\exp\left\{-\beta\widetilde{\Delta}_\mu(\mathbf{x})\right\} = (M-1)\exp\left\{-\beta\left(\langle\boldsymbol{\xi}_\mu,\mathbf{x}\rangle - \text{Max}_{\nu\in[M]}\langle\mathbf{x},\boldsymbol{\xi}_\nu\rangle\right)\right\}$. Consequently, (3.5) results from above and (Ramsauer et al., 2020, Theorem 4,5).

**For $\alpha \geq 2$.**  Following the setting of $\alpha$-EntMax in (Peters et al., 2019), the equation

$$2\text{-EntMax}(\beta\boldsymbol{\Xi}^\mathsf{T}\mathbf{x}) = \text{Sparsemax}(\beta\boldsymbol{\Xi}^\mathsf{T}\mathbf{x}) \tag{C.17}$$

holds. According to the closed form solution of Sparsemax in (Martins and Astudillo, 2016), it holds

$$[\text{Sparsemax}(\beta\boldsymbol{\Xi}^\mathsf{T}\mathbf{x})]_\mu \leq [\beta\boldsymbol{\Xi}^\mathsf{T}\mathbf{x}]_\mu - [\beta\boldsymbol{\Xi}^\mathsf{T}\mathbf{x}]_{(\kappa)} + \frac{1}{\kappa}, \tag{C.18}$$

for all $\mu \in [M]$. Then, the sparsemax retrieval error is

$$\|\mathcal{T}_{\text{Sparsemax}}(\mathbf{x}) - \boldsymbol{\xi}^\mu\| = \left\|\boldsymbol{\Xi}\,\text{Sparsemax}(\beta\boldsymbol{\Xi}^\mathsf{T}\mathbf{x}) - \boldsymbol{\xi}^\mu\right\| = \left\|\sum_{\nu=1}^{\kappa}\boldsymbol{\xi}_{(\nu)}\left[\text{Sparsemax}(\beta\boldsymbol{\Xi}^\mathsf{T}\mathbf{x})\right]_{(\nu)} - \boldsymbol{\xi}^\mu\right\|$$

$$\leq m + m\beta\left\|\sum_{\nu=1}^{\kappa}\left([\boldsymbol{\Xi}^\mathsf{T}\mathbf{x}]_{(\nu)} - [\boldsymbol{\Xi}^\mathsf{T}\mathbf{x}]_{(\kappa)} + \frac{1}{\beta\kappa}\right)\frac{\boldsymbol{\xi}_{(\nu)}}{m}\right\| \qquad \text{(By (C.18))}$$

$$\leq m + d^{1/2}m\beta\left[\kappa\left(\text{Max}_{\nu\in[M]}\langle\boldsymbol{\xi}_\nu,\mathbf{x}\rangle - [\boldsymbol{\Xi}^\mathsf{T}\mathbf{x}]_{(\kappa)}\right) + \frac{1}{\beta}\right]. \tag{C.19}$$

By the first inequality of Theorem 3.1, for $\alpha \geq 2$, we have

$$\|\mathcal{T}(\mathbf{x}) - \boldsymbol{\xi}_\mu\| \leq \|\mathcal{T}_{\text{Sparsemax}}(\mathbf{x}) - \boldsymbol{\xi}_\mu\| \leq m + d^{1/2}m\beta\left[\kappa\left(\text{Max}_{\nu\in[M]}\langle\boldsymbol{\xi}_\nu,\mathbf{x}\rangle - [\boldsymbol{\Xi}^\mathsf{T}\mathbf{x}]_{(\kappa)}\right) + \frac{1}{\beta}\right],$$

which completes the proof of (3.6). $\qquad\square$

## C.5  LEMMA 3.4

Our proof, built on (Hu et al., 2023, Lemma 2.1), proceeds in 3 steps:

- **(Step 1.)** We establish a more refined well-separation condition, ensuring that patterns $\{\boldsymbol{\xi}_\mu\}_{\mu\in[M]}$ are well-stored in $\mathcal{H}$ and can be retrieved by $\mathcal{T}$ with an error $\epsilon$ at most $R$.

- **(Step 2.)** This condition is then related to the cosine similarity of memory patterns, from which we deduce an inequality governing the probability of successful pattern storage and retrieval.
- **(Step 3.)** We pinpoint the conditions for exponential memory capacity and confirm their satisfaction.

Since the generalized sparse Hopfield shares the same well-separation condition (shown in below Lemma C.2), it has the same exponential memory capacity as the sparse Hopfield model (Hu et al., 2023, Lemma 3.1). For completeness, we restate the proof of (Hu et al., 2023, Lemma 3.1) below.

**Step 1.** To analyze the memory capacity of the proposed model, we first present the following two auxiliary lemmas.

**Lemma C.2.** [Corollary 3.1.1 of (Hu et al., 2023)] Let $\delta := \|\mathcal{T}_{\text{Dense}} - \boldsymbol{\xi}_\mu\| - \|\mathcal{T} - \boldsymbol{\xi}_\mu\|$. Then, the well-separation condition can be formulated as:

$$\Delta_\mu \geq \frac{1}{\beta} \ln\left(\frac{2(M-1)m}{R+\delta}\right) + 2mR. \tag{C.20}$$

Furthermore, if $\delta = 0$, this bound reduces to well-separation condition of Softmax-based Hopfield model.

*Proof of Lemma C.2.* Let $\mathcal{T}_{\text{Dense}}$ be the retrieval dynamics given by the dense modern Hopfield model (Ramsauer et al., 2020), and $\|\mathcal{T}(\mathbf{x}) - \boldsymbol{\xi}_\mu\|$ and $\|\mathcal{T}_{\text{Dense}}(\mathbf{x}) - \boldsymbol{\xi}_\mu\|$ be the retrieval error of generalized sparse and dense modern Hopfield model, respectively. By Theorem 3.1, we have

$$\|\mathcal{T}(\mathbf{x}) - \boldsymbol{\xi}_\mu\| \leq \|\mathcal{T}_{\text{Dense}}(\mathbf{x}) - \boldsymbol{\xi}_\mu\|. \tag{C.21}$$

By (Ramsauer et al., 2020, Lemma A.4), we have

$$\|\mathcal{T}_{\text{Dense}}(\mathbf{x}) - \boldsymbol{\xi}_\mu\| \leq 2\widetilde{\epsilon}m, \tag{C.22}$$

where $\widetilde{\epsilon} := (M-1)\exp\left\{-\beta\widetilde{\Delta}_\mu\right\} = (M-1)\exp\left\{-\beta\left(\langle\boldsymbol{\xi}_\mu, \mathbf{x}\rangle - \text{Max}_{\nu\in[M]}\langle\boldsymbol{\xi}_\mu, \boldsymbol{\xi}_\nu\rangle\right)\right\}$. Then, by the Cauchy-Schwartz inequality

$$|\langle\boldsymbol{\xi}_\mu, \boldsymbol{\xi}_\mu\rangle - \langle\mathbf{x}, \boldsymbol{\xi}_\mu\rangle| \leq \|\boldsymbol{\xi}_\mu - \mathbf{x}\| \cdot \|\boldsymbol{\xi}_\mu\| \leq \|\boldsymbol{\xi}_\mu - \mathbf{x}\|m, \quad \forall\mu\in[M], \tag{C.23}$$

we observe that $\widetilde{\Delta}_\mu$ can be expressed in terms of $\Delta_\mu$:

$$\widetilde{\Delta}_\mu \leq \Delta_\mu - 2\|\boldsymbol{\xi}_\mu - \mathbf{x}\|m = \Delta_\mu - 2mR, \tag{C.24}$$

where $R$ is radius of the sphere $S_\mu$. Thus, inserting the upper bound given by (C.22) into (3.5), we obtain

$$\begin{aligned}
\|\mathcal{T}(\mathbf{x}) - \boldsymbol{\xi}_\mu\| &\leq \|\mathcal{T}_{\text{Dense}}(\mathbf{x}) - \boldsymbol{\xi}_\mu\| \leq 2\widetilde{\epsilon}m \tag{C.25}\\
&\leq 2(M-1)\exp\{-\beta(\Delta_\mu - 2mR)\}m. \tag{C.26}
\end{aligned}$$

Then, for any given $\delta := \|\mathcal{T}_{\text{Dense}}(\mathbf{x}) - \boldsymbol{\xi}_\mu\| - \|\mathcal{T}(\mathbf{x}) - \boldsymbol{\xi}_\mu\| \leq 0$, the retrieval error $\|\mathcal{T}(\mathbf{x}) - \boldsymbol{\xi}_\mu\|$ has an upper bound:

$$\|\mathcal{T}(\mathbf{x}) - \boldsymbol{\xi}_\mu\| \leq 2(M-1)\exp\{-\beta(\Delta_\mu - 2mR + \delta)\}m - \delta \leq \|\mathcal{T}_{\text{Dense}}(\mathbf{x}) - \boldsymbol{\xi}_\mu\|. \tag{C.27}$$

Therefore, for $\mathcal{T}$ to be a mapping $\mathcal{T} : S_\mu \to S_\mu$, we need the well-separation condition

$$\Delta_\mu \geq \frac{1}{\beta} \ln\left(\frac{2(M-1)m}{R+\delta}\right) + 2mR. \tag{C.28}$$

$\square$

**Lemma C.3** ((Hu et al., 2023; Ramsauer et al., 2020)). If the identity

$$ac + c \ln c - b = 0, \tag{C.29}$$

holds for all real numbers $a, b \in \mathbb{R}$, then $c$ takes a solution:

$$c = \frac{b}{W_0(\exp(a + \ln b))}. \tag{C.30}$$

*Proof of Lemma C.3.* We restate the proof of (Hu et al., 2023, Lemam 3.1) here for completeness.

With the given equation $ac + c \ln c - b = 0$, we solve for $c$ by following steps:

$$
\begin{aligned}
ac + c \ln c - b &= 0, \\
a + \ln c &= \frac{b}{c}, \\
\frac{b}{c} + \ln\left(\frac{b}{c}\right) &= a + \ln b, \\
\frac{b}{c} \exp\left(\frac{b}{c}\right) &= \exp(a + \ln b), \\
\frac{b}{c} &= W_0(\exp(a + \ln b)), \\
c &= \frac{b}{W_0(\exp(a + \ln b))}.
\end{aligned}
$$

$\square$

Then, we present the main proof of Lemma 3.4.

*Proof of Lemma 3.4.* Since the generalized Hopfield model shares the same well-separation condition as the sparse Hopfield model (Hu et al., 2023), the proof of the exponential memory capacity automatically follows that of (Hu et al., 2023). We restate the proof of (Hu et al., 2023, Corollary 3.1.1) here for completeness.

**(Step 2.) & (Step 3.)** Here we define $\Delta_{\min}$ and $\theta_{\mu\nu}$ as $\Delta_{\min} := \text{Min}_{\mu \in [M]} \Delta_\mu$ and the angle between two patterns $\boldsymbol{\xi}^\mu$ and $\boldsymbol{\xi}^\nu$, respectively. Intuitively, $\theta_{\mu\nu} \in [0, \pi]$ represent the pairwise correlation of two patterns the two patterns and hence

$$\Delta_{\min} = \text{Min}_{1 \le \mu \le \nu \le M} \left[ m^2 \left(1 - \cos(\theta_{\mu\nu})\right) \right] = m^2 \left[1 - \cos(\theta_{\min})\right], \tag{C.31}$$

where $\theta_{\min} := \text{Min}_{1 \le \mu \le \nu \le M} \theta_{\mu\nu} \in [0, \pi]$.

From the well-separation condition (C.2), we have

$$\Delta_\mu \ge \Delta_{\min} \ge \frac{1}{\beta} \ln\left(\frac{2(M-1)m}{R + \delta}\right) + 2mR. \tag{C.32}$$

Hence, we have

$$m^2 \left[1 - \cos(\theta_{\min})\right] \ge \frac{1}{\beta} \ln\left(\frac{2(M-1)m}{R + \delta}\right) + 2mR. \tag{C.33}$$

Therefore, we are able to write down the probability of successful storage and retrieval, i.e. minimal separation $\Delta_{\min}$ satisfies Lemma C.2:

$$P\left(m^2\left[1-\cos(\theta_{\min})\right] \geq \frac{1}{\beta}\ln\left(\frac{2(M-1)m}{R+\delta}\right) + 2mR\right) = 1-p. \tag{C.34}$$

By (Olver et al., 2010, (4.22.2)), it holds

$$\cos(\theta_{\min}) \leq 1 - \frac{\theta_{\min}^2}{5} \quad \text{for} \quad 0 \leq \cos(\theta_{\min}) \leq 1, \tag{C.35}$$

and hence

$$P\left(M^{\frac{2}{d-1}}\theta_{\min} \geq \frac{\sqrt{5}M^{\frac{2}{d-1}}}{m}\left[\frac{1}{\beta}\ln\left(\frac{2(M-1)m}{R+\delta}\right) + 2mR\right]^{\frac{1}{2}}\right) = 1-p. \tag{C.36}$$

Here we introduce $M^{2/d-1}$ on both sides in above for later convenience.

Let $\omega_d := \frac{2\pi^{\frac{d+1/2}{}}}{\Gamma\left(\frac{d+1}{2}\right)}$, be the surface area of a $d$-dimensional unit sphere, where $\Gamma(\cdot)$ represents the gamma function. By (Brauchart et al., 2018, Lemma 3.5), it holds

$$1-p \geq 1 - \frac{1}{2}\gamma_{d-1}5^{\frac{d-1}{2}}M^2m^{-(d-1)}\left[\frac{1}{\beta}\ln\left(\frac{2(M-1)m}{R+\delta}\right) + 2mR\right]^{\frac{d-1}{2}}, \tag{C.37}$$

where $\gamma_d$ is characterized as the ratio between the surface areas of the unit spheres in $(d-1)$ and $d$ dimensions, respectively: $\gamma_d := \frac{1}{d}\frac{\omega_{d-1}}{\omega_d}$.

Since $M = \sqrt{p}C^{\frac{d-1}{4}}$ is always true for $d, M \in \mathbb{N}_+$, $p \in [0,1]$ and some real values $C \in \mathbb{R}$, we have

$$5^{\frac{d-1}{2}}C^{\frac{d-1}{2}}m^{-(d-1)}\left\{\frac{1}{\beta}\ln\left[\frac{2\left(\sqrt{p}C^{\frac{d-1}{4}}-1\right)m}{R+\delta}\right] + \frac{1}{\beta}\right\}^{\frac{d-1}{2}} \leq 1. \tag{C.38}$$

Then, we rearrange above as

$$\frac{5C}{m^2\beta}\left\{\ln\left[\frac{2\left(\sqrt{p}C^{\frac{d-1}{4}}-1\right)m}{R+\delta}\right] + 1\right\} - 1 \leq 0, \tag{C.39}$$

and identify

$$a := \frac{4}{d-1}\left\{\ln\left[\frac{2m(\sqrt{p}-1)}{R+\delta}\right] + 1\right\}, \quad b := \frac{4m^2\beta}{5(d-1)}. \tag{C.40}$$

By Lemma C.3, we have

$$C = \frac{b}{W_0(\exp\{a+\ln b\})}, \tag{C.41}$$

where $W_0(\cdot)$ is the upper branch of the Lambert $W$ function. Since the domain of the Lambert $W$ function is $x > (-1/e, \infty)$ and the fact $\exp\{a+\ln b\} > 0$, the solution for (C.41) exists. When the inequality (C.38) holds, we arrive the lower bound on the exponential storage capacity $M$:

$$M \geq \sqrt{p}C^{\frac{d-1}{4}}. \tag{C.42}$$

In addition, by the asymptotic expansion of the Lambert $W$ function (Hu et al., 2023, Lemma 3.1), it also holds $M \geq M_{\text{Dense}}$, where $M_{\text{Dense}}$ is the memory capacity of the dense modern Hopfield model (Ramsauer et al., 2020). □

# D  METHODOLOGY DETAILS

## D.1  THE MULTI-STEP GSH UPDATES

GSH inherits the capability of multi-step update for better retrieval accuracy, which is summarized in below Algorithm 1 for a given number of update steps $\kappa$. In practice, we find that a single update suffices, consistent with our theoretical finding (3.5) of Theorem 3.1.

---

**Algorithm 1** Multi-Step Generalized Sparse Hopfield Update for GSH

---

**Require:** $\kappa \in \mathbb{R} \geq 1, \mathbf{Q} \in \mathbb{R}^{\text{len}_Q \times D_Q}, \mathbf{K} \in \mathbb{R}^{\text{len}_K \times D_K}, \mathbf{W}_V \in \mathbf{R}^{D_K \times d}, \mathbf{W}_K \in \mathbf{R}^{D_K \times D_Q}$
  **for** $i \to 1$ to $\kappa$ **do**
    $\mathbf{Q}^{\text{new}} = \alpha\text{-EntMax}\left(\beta \mathbf{Q}(\mathbf{KW}_K)^{\mathsf{T}}\right)\mathbf{K}$                        *Hopfield Update*
    $\mathbf{Q} \leftarrow \mathbf{Q}^{\text{new}}$
  **end for**
  $\mathbf{Z} = \mathbf{QW}_V$
  **return** $\mathbf{Z}$

---

## D.2  GSHPooling AND GSHLayer

Here we provide the operational definitions of the GSHPooling and the GSHLayer.

**Definition D.1** (Generalized Sparse Hopfield Pooling (GSHPooling)). Given inputs $\mathbf{Y} \in \mathbb{R}^{\text{len}_Y \times D_Y}$, and $\text{len}_Q$ query patterns $\mathbf{Q} \in \mathbb{R}^{\text{len}_Q \times D_K}$ the 1-step Sparse Adaptive Hopfield Pooling update is

$$\text{GSHPooling}\left(\mathbf{Y}\right) = \alpha\text{-EntMax}\left(\mathbf{QK}^T/\sqrt{D_k}\right)\mathbf{V}, \tag{D.1}$$

Here we have $\mathbf{K}, \mathbf{V}$ equal to $\mathbf{V} = \mathbf{YW}_K\mathbf{W}_V, \mathbf{K} = \mathbf{YW}_K$, and $\mathbf{W}_V \in \mathbb{R}^{D_K \times D_K}, \mathbf{W}_K \in \mathbb{R}^{D_K \times D_K}$. Where $d$ is the dimension of $K$. And the query pattern $\mathbf{Q}$ is a learnable variable, and is independent from the input, the size of $\text{len}_Q$ controls how many query patterns we want to store.

**Definition D.2** (Generalized Sparse Hopfield Layer (GSHLayer)). Given inputs $\mathbf{Y} \in \mathbb{R}^{\text{len}_Y \times D_Y}$, and $\text{len}_Q$ query patterns $\mathbf{Q} \in \mathbb{R}^{\text{len}_Q \times D_K}$ the 1-step Sparse Adaptive Hopfield Layer update is

$$\text{GSHLayer}\left(\mathbf{R}, \mathbf{Y}\right) = \alpha\text{-EntMax}\left(\mathbf{RY}^T/\sqrt{D_k}\right)\mathbf{Y}, \tag{D.2}$$

Here $\mathbf{R}$ is the input and $\mathbf{Y}$ can be either learnable weights or given as an input.

## D.3  EXAMPLE: MEMORY RETRIEVAL FOR IMAGE COMPLETION

The standard memory retrieval mechanism of Hopfield Models contains two inputs, the query $\mathbf{x}$ and the associative memory set $\mathbf{\Xi}$. The goal is to retrieve an associated memory $\boldsymbol{\xi}$ most similar to the query $\mathbf{x}$ from the stored memory set $\mathbf{\Xi}$. For example, in (Ramsauer et al., 2020), the query $\mathbf{x}$ is a corrupted/noisy image from CIFAR10, and the associative memory set $\mathbf{\Xi}$ is the CIFAR10 image dataset. All images are flattened into vector-valued patterns. This task can be achieved by taking the query as $\mathbf{R} = \mathbf{x}$ and the associative memory set as $\mathbf{Y} = \mathbf{\Xi}$ for GSHLayer with fixed parameters. After steps of updates, we expect the output of the GSHLayer to be the recovered version of $\mathbf{x}$.

## D.4  PSEUDO LABEL RETRIEVAL

Here, we present the use of the memory retrieval mechanism from modern Hopfield models to generate pseudo-labels for queries $\mathbf{R}$, thereby enhancing predictions. Given a set of memory patterns $\mathbf{Y}$ and their corresponding labels $\mathbf{Y}_{\text{label}}$, we concatenate them together to form the *label-included*

memory set $\widetilde{\mathbf{Y}}$. Take CIFAR10 for example, we can concatenate the flatten images along with their one-hot encoded labels together as the memory set. For the query, we use the input with padded zeros concatenated at the end of it. The goal here is to "retrieve" the padding part in the query, which is expected to be the retrieved "pseudo-label". Note that this pseudo-label will be a weighted sum over all other labels in the associative memory set. An illustration of this mechanism can be found in Figure 2. For the retrieved pseudo-label, we can either use it as the final prediction, or use it as pseudo-label to provide extra information for the model.

### D.5 ALGORITHM FOR STANHOP-NET

Here we summarize the STanHop-Net as below algorithm.

---

**Algorithm 2** STanHop-Net

---

**Require:** $L \geq 1, \mathbf{Z} \in \mathbb{R}^{T \times C \times D_{hidden}^0}$
    **for** $\ell \to 1$ to $L$ **do**
        $\mathbf{Z}_{\text{enc}}^{\ell} = \texttt{STanHop} \left( \text{Coarse-Graining} \left( \mathbf{Z}_{\text{enc}}^{\ell-1}, \Delta \right) \right)$         *encoder forward*
    **end for**
    $\mathbf{Z}_{\text{dec}}^0 = \mathbf{E}_{\text{dec}}$         *learnable positional embedding*
    **for** $\ell \to 1$ to $L$ **do**         *decoder forward*
        $\widetilde{\mathbf{Z}}_{\text{dec}}^{\ell} = \texttt{STanHop} \left( \mathbf{Z}_{\text{dec}}^{\ell-1} \right)$
        $\widehat{\mathbf{Z}}_{\text{dec}}^{\ell} = \texttt{GSH} \left( \mathbf{Z}_{\text{dec}}^{\ell}, \mathbf{Z}_{\text{enc}}^{\ell} \right)$
        $\check{\mathbf{Z}}_{\text{dec}}^{\ell} = \text{LayerNorm} \left( \widehat{\mathbf{Z}}_{\text{dec}}^{\ell} + \widetilde{\mathbf{Z}}_{\text{dec}}^{\ell} \right)$
        $\mathbf{Z}_{\text{dec}}^{\ell} = \text{LayerNorm} \left( \check{\mathbf{Z}}_{\text{dec}}^{\ell} + \text{MLP} \left( \check{\mathbf{Z}}_{\text{dec}}^{\ell} \right) \right)$
    **end for**
    **return** $\mathbf{Z}_{\text{dec}}^{L} \in \mathbb{R}^{\frac{P}{T} \times C \times D_{\text{hidden}}}$

---

# E    ADDITIONAL NUMERICAL EXPERIMENTS

Here we provide additional experimental investigations to back up the effectiveness of our method.

## E.1    NUMERICAL VERIFICATION'S OF THEORETICAL RESULTS

**Faster Fixed Point Convergence and Better Generalization.**    In Figure 4, to support our theoretical results in Section 4, we numerically analyze the convergence behavior of the GSH, compared with the dense modern Hopfield layer Hopfield.

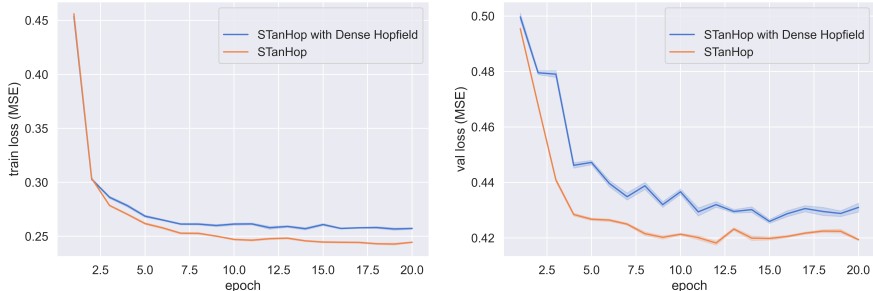

Figure 4: The training and validation loss curves of STanHop (D), i.e. STanHop-Net with dense modern Hopfield Hopfield layer, and STanHop-Net with GSH layer. The results show that the generalized sparse Hopfield model enjoys faster convergence than the dense model and also obtain better generalization.

In Figure 4, we plot the loss curves for STanHop-Net using both generalized sparse and dense modern models on the ETTh1 dataset for the multivariate time series prediction tasks.

The results reveal that the generalized sparse Hopfield model (GSH) converges faster than the dense model (Hopfield) and also achieves better generalization. This empirically supports our theoretical findings presented in Theorem 3.1, which suggest that the generalized sparse Hopfield model provides faster retrieval convergence with enhanced accuracy.

**Memory Capacity and Noise Robustness.**    Following (Hu et al., 2023), we also conduct experiments verifying our memory capacity and noise robustness theoretical results (Lemma 3.4 and Theorem 3.1), and report the results in Figure 5. The plots present average values and standard deviations derived from 10 trials.

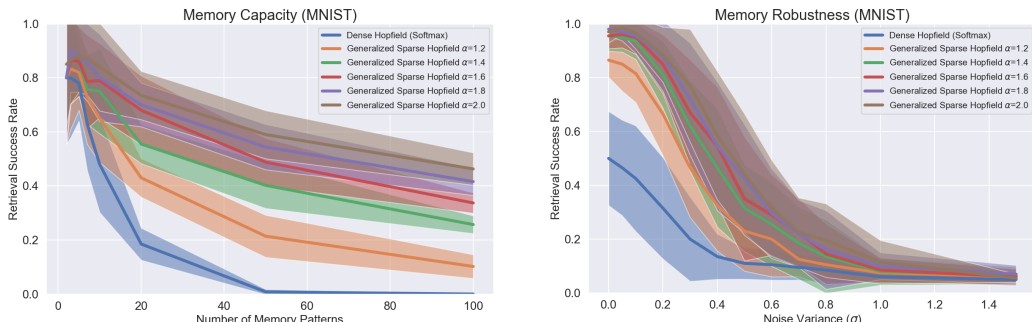

Figure 5: **Left:** Memory Capacity measured by successful half-masked retrieval rates. **Right:** Memory Robustness measured by retrieving patterns with various noise levels. A query pattern is considered accurately retrieved if its cosine similarity error falls below a specified threshold. We set error threshold of 20% and $\beta$=0.01 for better visualization. We plot the average and variance from 10 trials. These findings demonstrate the generalized sparse Hopfield model's ability of capturing data sparsity, improved memory capacity and its noise robustness.

Regarding memory capacity (displayed on the left side of Figure 5), we evaluate the generalized sparse Hopfield model's ability to retrieve half-masked patterns from the MNIST dataset, in comparison to the Dense modern Hopfield model (Ramsauer et al., 2020).

Regarding robustness against noisy queries (displayed on the right side of Figure 5), we introduce Gaussian noises of varying variances ($\sigma$) to the images.

These findings demonstrate the generalized sparse Hopfield model's ability of capturing data sparsity, improved memory capacity and its noise robustness.

## E.2 COMPUTATIONAL COST ANALYSIS OF MEMORY MODULES

Here we analyze the computational cost between the **Plug-and-Play** memory plugin module and the baseline. We evaluate 2 matrices: (i) the number of floating point operations (flops) (ii) number of parameters of the model. Note that for **Plug-and-Play** module, the parameter amount will not be affected by the size of external memory set. The result can be found in Figure 6 and Figure 7.

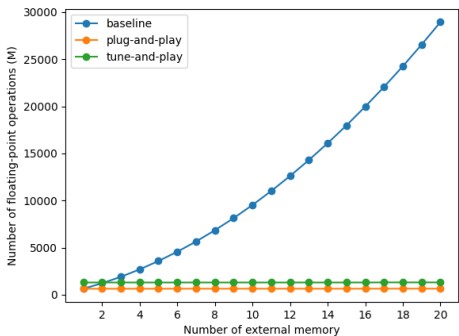 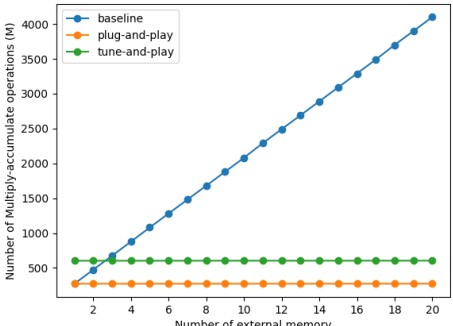

Figure 6: The number of floating-point operations (flops) (in millions) comparison between **Plug-and-Play**, **Tune-and-Play** and the baseline. The result shows that the **Plug-and-Play**, **Tune-and-Play** successfully reduce the required computational cost to process an increased amount of data.

Figure 7: The number of Multiply–accumulate operations (MACs) (in millions) comparison between **Plug-and-Play**, **Tune-and-Play** and the baseline. The result shows that both of our memory plugin modules face little MACs increasement while the baseline model MACs increase almost linearly w.r.t. the input size.

## E.3 ABLATION STUDIES

**Hopfield Model Ablation.** Beside our proposed generalized sparse modern Hopfield model, we also test STanHop-Net with 2 other existing different modern Hopfield models: the dense modern Hopfield model (Ramsauer et al., 2020) and the sparse modern Hopfield model (Hu et al., 2023). We report their results in Table 1.

We terms them as **STanHop-Net (D)** and **STanHop-Net (S)** where (D) and (S) are for "Dense" and "Sparse" respectively.

**Component Ablation.** In order to evaluate the effectiveness of different components in our model, we perform an ablation study by removing one component at a time. In below, we denote Patch Embedding as (PE), StanHop as (SH), Hopfield Pooling as (HP), Multi-Resolution as (MR). We also denote their removals with "w/o" (i.e., without.)

For w/o PE, we set the patch size $P$ equals 1. For w/o MR, we set the coarse level $\Delta$ as 1. For w/o SH and w/o HP, we replace those blocks/layers with an MLP layer with GELU activation and layer normalization.

The results are showed in Table 3. From the ablation study results, we observe that removing the STanHop block gives the biggest negative impact on the performance. Showing that the STanHop block contributes the most to the model performance. Note that patch embedding also provides a notable improvement on the performance. Overall, every component provides a different level of performance boost.

Table 3: **Component Ablation.** We conduct component ablation by separately removing Patch Embedding (PE), STanHop (SH), Hopfield Pooling (HP), and Multi-Resolution (MR). We report the mean MSE and MAE over 10 runs, with variances omitted as they are all $\leq$ 0.15%. The results indicate that while every single component in STanHop-Net provides performance boost, the impact of STanHop block on model performance is the most significant among all other components.

| Models | | STanHop | | w/o PE | | w/o SH | | w/o HP | | w/o MR | |
|---|---|---|---|---|---|---|---|---|---|---|---|
| | Metric | MSE | MAE | MSE | MAE | MSE | MAE | MSE | MAE | MSE | MAE |
| ETTh1 | 24 | 0.294 | 0.360 | 0.318 | 0.368 | 0.305 | 0.365 | 0.306 | 0.363 | 0.307 | 0.363 |
| | 48 | 0.340 | 0.387 | 0.357 | 0.389 | 0.352 | 0.393 | 0.346 | 0.387 | 0.348 | 0.385 |
| | 168 | 0.420 | 0.452 | 0.454 | 0.476 | 0.480 | 0.500 | 0.434 | 0.455 | 0.447 | 0.464 |
| | 336 | 0.450 | 0.472 | 0.501 | 0.524 | 0.530 | 0.535 | 0.462 | 0.473 | 0.482 | 0.486 |
| | 720 | 0.512 | 0.520 | 0.540 | 0.538 | 0.610 | 0.581 | 0.524 | 0.526 | 0.537 | 0.531 |
| WTH | 24 | 0.292 | 0.341 | 0.318 | 0.365 | 0.335 | 0.375 | 0.340 | 0.374 | 0.325 | 0.373 |
| | 48 | 0.363 | 0.402 | 0.386 | 0.421 | 0.414 | 0.439 | 0.385 | 0.420 | 0.391 | 0.427 |
| | 168 | 0.499 | 0.515 | 0.504 | 0.521 | 0.507 | 0.519 | 0.503 | 0.525 | 0.520 | 0.509 |
| | 336 | 0.499 | 0.515 | 0.514 | 0.529 | 0.532 | 0.541 | 0.513 | 0.528 | 0.533 | 0.542 |
| | 720 | 0.548 | 0.556 | 0.570 | 0.565 | 0.569 | 0.565 | 0.539 | 0.548 | 0.555 | 0.557 |

### E.4 THE IMPACT OF VARYING $\alpha$

We examine the impact of increasing the value of $\alpha$ on memory capacity and noise robustness. It is known that as $\alpha$ approaches infinity, the $\alpha$-entmax operation transitions to a hardmax operation (Peters et al., 2019; Correia et al., 2019). Furthermore, it is also known that memory pattern retrieval using hardmax is expected to exhibit perfect retrieval ability, potentially offering a larger memory capacity than the softmax modern Hopfield model in pure retrieval tasks (Millidge et al., 2022). Our empirical investigations confirm that higher values of $\alpha$ frequently lead to higher memory capacity. We report results only up to $\alpha = 5$, as we observed that values of $\alpha$ greater than 5 consistently lead to numerical errors, especially under float32 precision. It is crucial to note, that while the hardmax operation (realized when $\alpha \to \infty$) may maximize memory capacity, its lack of differentiability renders it unsuitable for gradient descent-based optimization.

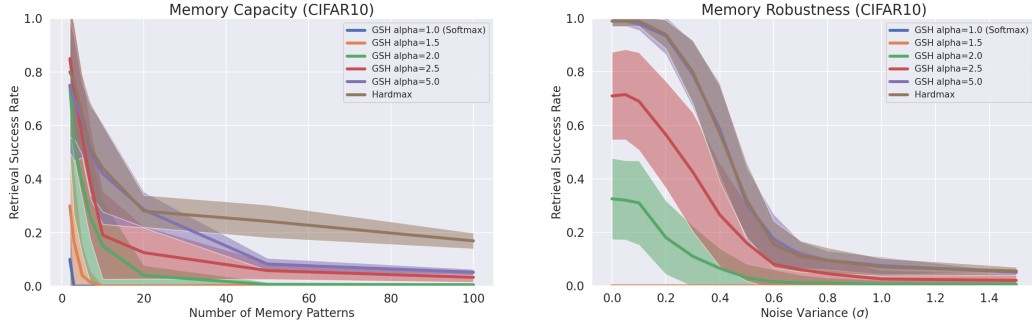

Figure 8: **Left:** Memory Capacity measured by successful half-masked retrieval rates w.r.t. different values of $\alpha$ on CIFAR10. **Right:** Memory Robustness measured by retrieving patterns with various noise levels on CIFAR10. A query pattern is considered accurately retrieved if its cosine similarity error falls below a specified threshold. We set error threshold of 20% and $\beta = 0.01$ for better visualization. We plot the average and variance from 10 trials. We can see that using hardmax (argmax) normally gives the best retrieval result as it retrieves only the most similar pattern w.r.t. dot product distance. And setting $\alpha = 5$ approximately gives the similar result while having $\alpha = 5$ keeps the overall mechanism differentiable.

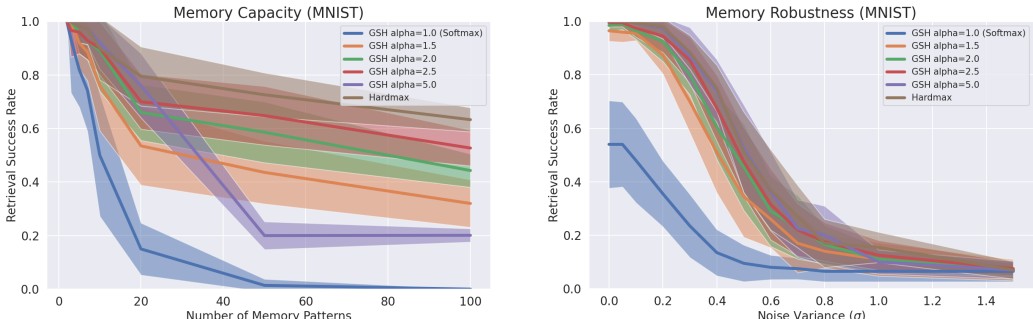

Figure 9: **Left:** Memory Capacity measured by successful half-masked retrieval rates w.r.t. different values of $\alpha$ on MNIST. **Right:** Memory Robustness measured by retrieving patterns with various noise levels on MNIST. A query pattern is considered accurately retrieved if its cosine similarity error falls below a specified threshold. We set error threshold of 20% and $\beta = 0.1$ for better visualization. We plot the average and variance from 10 trials. We can see that using hardmax (argmax) normally gives the best retrieval result as it retrieves only the most similar pattern w.r.t. dot product distance. And setting $\alpha = 5$ approximately gives the similar result while having $\alpha = 5$ keeps the overall mechanism differentiable.

### E.5 MEMORY USAGE OF STANHOP AND LEARNABLE $\alpha$

To compare the memory and GPU usage, we benchmark STanHop with STanHop-(D) (using *dense* modern Hopfield layers). In below Figure 10 and Figure 11, we report the footprints of from the weight and bias (wandb) system (Biewald et al., 2020). The figures clearly demonstrate that the computational cost associated with learning an additional $\alpha$ for adaptive sparsity is negligible.

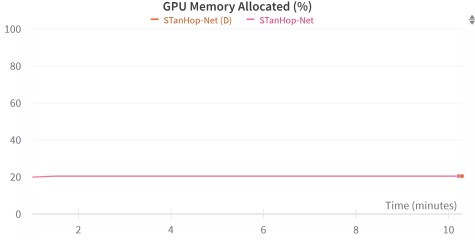 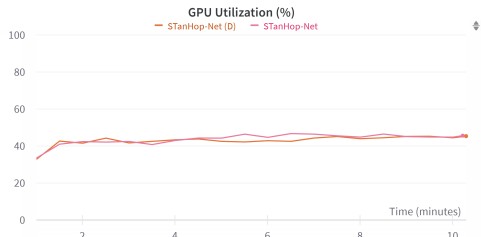

Figure 10: The GPU memory allocation between STanHop-Net with and without learnable alpha (STanHop-Net (D)). We can see that with learnable alpha does not significantly increase reuired GPU memory.

Figure 11: The percentage of gpu utilization between STanHop-Net with and without learnable alpha (STanHop-Net (D)). We can see that with learnable alpha does not significantly increase the GPU utilization.

### E.6 TIME COMPLEXITY ANALYSIS OF STANHOP-NET

Here we use the same notation as introduced in the main paper. Let $T$ be the number of input length on the time dimension, $D_{\text{hidden}}$ be the hidden dimension, $P$ be patch size, $D_{\text{out}}$ be the prediction horizon, $\text{len}_Q$ be the size of query pattern in `GSHPooling`.

- Patch embedding: $\mathcal{O}(D_{\text{hidden}} \times T) = \mathcal{O}(T)$.
- Temporal and Cross series GSH: $\mathcal{O}(D_{\text{hidden}}^2 \times T^2 \times P^{-2}) = \mathcal{O}(T^2 \times P^{-2}) = \mathcal{O}(T^2)$
- Coarse graining: $\mathcal{O}(D_{\text{hidden}}^2 \times T) = \mathcal{O}(T)$
- GSHPooling : $\mathcal{O}(D_{\text{hidden}} \times \text{len}_Q \times T^2 \times P^{-2}) = \mathcal{O}(\text{len}_Q \times T^2)$
- PlugMemory: $\mathcal{O}(T^2)$
- TuneMemory: $\mathcal{O}((T + D_{\text{out}})^2)$

Additionally, the number of parameters of STanHop-Net 0.78 million. As a reference, with batch size of 32, input length of 168, STanHop-Net requires 2 minutes per epoch for the ETTh1 dataset. Meanwhile, STanHop-Net (D) also requires 2 minutes per epoch under same setting.

### E.7 HYPERPARAMETER SENSITIVITY ANALYSIS

We conduct experiments exploring the parameter sensitivity of STanHop-Net on the ILI dataset. Our results (in Table 5 and Table 6) show that STanHop-Net is not sensitive to hyperparameter changes.

The way that we conduct the hyperparameter sensitivity analysis is that to show the model's sensitivity to a hyperparameter $h$, we change the value of $h$ in each run and keep the rest of the hyperparameters' values as default and we record the MAE and MSE on the test set. We train and evaluate the model 3 times for 3 different values for each hyperparameter. We analyze the model's sensitivity to 7 hyperparameters respectively. We conduct all the experiments on the ILI dataset.

Table 4: Default Values of Hyperparameters in Sensitivity Analysis

| Parameter | Default Value |
|---|---|
| seg_len (patch size) | 6 |
| window_size (coarse level) | 2 |
| e_layers (number of encoder layers) | 3 |
| d_model | 32 |
| d_ff (feedforward dimension) | 64 |
| n_heads (number of heads) | 2 |

Table 5: MAEs for each value of the hyperparameter with the rest of the hyperparameters as default. For each hyperparameter, each row's MAE score corresponds to the hyperparameter value inside the parentheses in the same order. For example, when lr is 1e-3, the MAE score is 1.313.

| lr (1e-3, 1e-4, 1e-5) | seg_len (6, 12, 24) | window_size (2,4,8) | e_layers (3,4,5) | d_model (32, 64, 128) | d_ff (64, 128, 256) | n_heads (2,4,8) |
|---|---|---|---|---|---|---|
| 1.313 | 1.313 | 1.313 | 1.313 | 1.313 | 1.313 | 1.313 |
| 1.588 | 1.311 | 1.285 | 1.306 | 1.235 | 1.334 | 1.319 |
| 1.673 | 1.288 | 1.368 | 1.279 | 1.372 | 1.302 | 1.312 |

Table 6: MSEs for each value of the hyperparameter with the rest of the hyperparameters as default. For each hyperparameter, each row's MSE score corresponds to the hyperparameter value inside the parentheses in the same order. For example, when lr is 1e-3, the MSE score is 3.948.

| lr (1e-3, 1e-4, 1e-5) | seg_len (6, 12, 24) | window_size (2,4,8) | e_layers (3,4,5) | d_model (32, 64, 128) | d_ff (64, 128, 256) | n_heads (2,4,8) |
|---|---|---|---|---|---|---|
| 3.948 | 3.948 | 3.948 | 3.948 | 3.948 | 3.948 | 3.948 |
| 5.045 | 3.968 | 3.877 | 3.915 | 3.566 | 3.983 | 3.967 |
| 5.580 | 3.865 | 4.267 | 3.834 | 4.078 | 3.866 | 3.998 |

## E.8 ADDITIONAL MULTIPLE INSTANCE LEARNING EXPERIMENTS

We also evaluate the efficacy of the proposed GSH layer on Multiple Instance Learning (MIL) tasks. In essence, MIL is a type of supervised learning whose training data are divided into bags and labeling individual data points is difficult or impractical, but bag-level labels are available (Ilse et al., 2018). We follow (Ramsauer et al., 2020; Hu et al., 2023) to conduct the multiple instance learning experiments on MNIST.

**Model.** We first flatten each image and use a fully connected layer to project each image to the embedding space. Then, we perform GSHPooling and use linear projection for prediction.

**Hyperparameters.** For hyperparameters, we use hidden dimension of 256, number of head as 4, dropout as 0.3, training epoch as 100, optimizer as AdamW, initial learning rate as $1e - 4$, and we also use the cosine annealing learning rate decay.

**Baselines.** We benchmark the Generalized Sparse modern Hopfield model (GSH) with the Sparse (Hu et al., 2023) and Dense (Ramsauer et al., 2020) modern Hopfield models.

**Dataset.** For the training dataset, we randomly sample 1000 positive and 1000 negative bags for each bag size. For test data, we randomly sample 250 positive and 250 negative bags for each bag size. We set the positive signal to the images of digit 9, and the rest as negative signals. We vary the bag size and report the accuracy and loss curves of 10 runs on both training and test data.

**Results.** Our results (shown in the figures below) demonstrate that GSH converges faster than the baselines in most settings, as it can adapt to varying levels of data sparsity. This is consistent with our theoretical findings in Theorem 3.1, which state that GSH achieves higher accuracy and faster fixed-point convergence compared to the dense model.

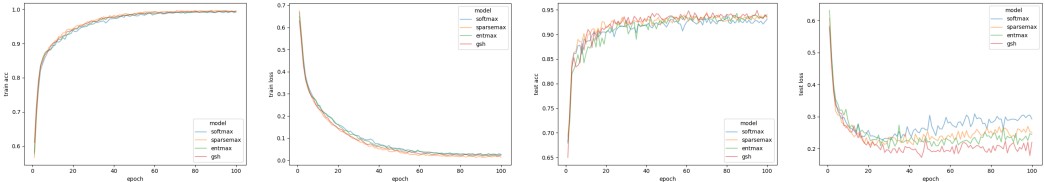

Figure 12: The MIL experiment with bag size 5. From left to right: (1) Training data accuracy curve (2) Training data loss curve (3) Test data accuracy curve (4) Test data accuracy curve

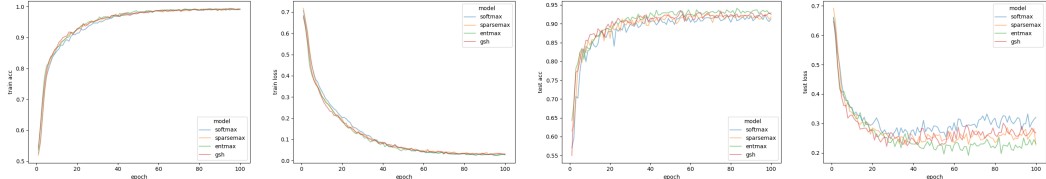

Figure 13: The MIL experiment with bag size 10. From left to right: (1) Training data accuracy curve (2) Training data loss curve (3) Test data accuracy curve (4) Test data accuracy curve

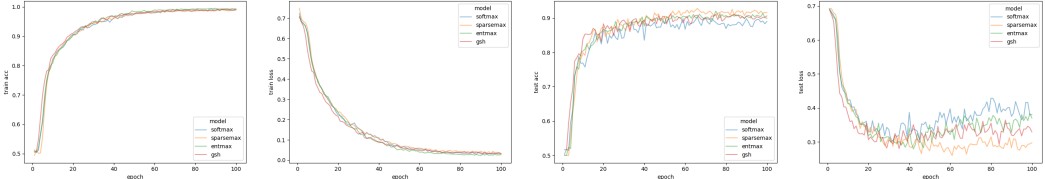

Figure 14: The MIL experiment with bag size 20. From left to right: (1) Training data accuracy curve (2) Training data loss curve (3) Test data accuracy curve (4) Test data accuracy curve

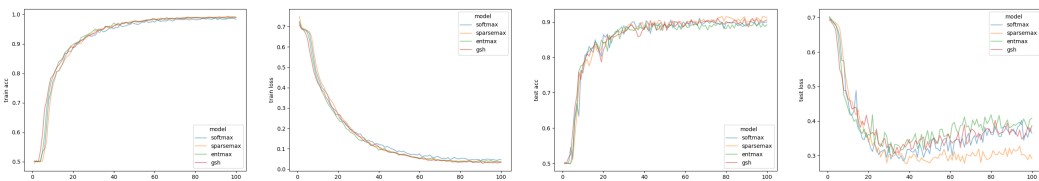

Figure 15: The MIL experiment with bag size 30. From left to right: (1) Training data accuracy curve (2) Training data loss curve (3) Test data accuracy curve (4) Test data accuracy curve

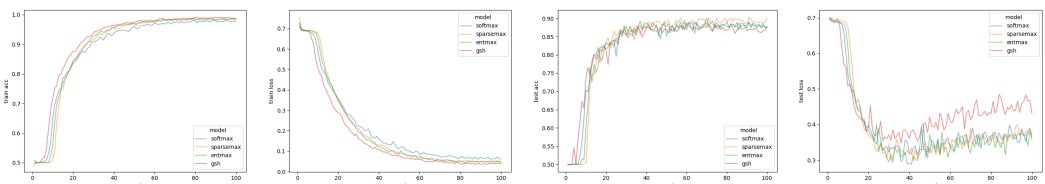

Figure 16: The MIL experiment with bag size 50. From left to right: (1) Training data accuracy curve (2) Training data loss curve (3) Test data accuracy curve (4) Test data accuracy curve

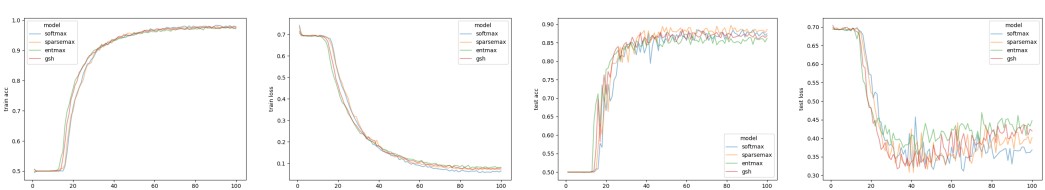

Figure 17: The MIL experiment with bag size 100. From left to right: (1) Training data accuracy curve (2) Training data loss curve (3) Test data accuracy curve (4) Test data accuracy curve

### E.9 STanHop-Net Outperforms DLinear in Settings Dominated by Multivariate Correlations: A Case Study

The performance of STanHop-Net in the main text, as presented in Table 1, does not show a clear superiority over DLinear (Zeng et al., 2023). We attribute this to the nature of the common benchmark datasets in Table 1, which are not dominated by multivariate correlations.

To verify our conjecture, we employ a strongly correlated multivariate time series dataset as a test bed, representing a practical scenario where multivariate correlations are the predominant source of predictive information in input features. In such a scenario, following the same setting in Section 5.1, our experiments show that STanHop-Net consistently outperforms DLinear. We also compare with another top-tier channel-independent method, PatchTST (Nie et al., 2022), and results show that we outperform it.

Specifically, we follow the experimental settings outlined in Section 5.1, but with a specific focus on cases involving *small* lookback windows. This emphasis aims to reduce autoregressive correlation in data with smaller lookback windows, thereby increasing the dominance of multivariate correlations. We refer just a scenario as *Cross-Sectional Regression*[5] (CSR) in time series modelling (Reneau et al., 2023; Fama and French, 2020; Andrews, 2005) and (Wooldridge et al., 2016, Part 1).

**Dataset.** We evaluate our model on the synthetic dataset[6] generated in the ICML2023 Feature Programming paper (Reneau et al., 2023). Feature programming is an automated, programmable method for feature engineering. It produces a large number of predictive features from any given input time series. These generated features, termed *extended features*, are by construction highly correlated. The synthetic dataset, containing 44 extended features derived from the taxi dataset (see (Reneau et al., 2023, Section D.1) for the generation details), is thereby a strongly correlated multivariate time series dataset.

**Baseline.** We mainly compare our performance with DLinear (Zeng et al., 2023) as it showed comparable performance in Table 1. We also include an additional comparison with PatchTST (Nie et al., 2022) to further showcase our method's advantage in capturing multivariate correlations.

**Setting.** For both STaHop-Net and DLinear, we use the same hyperparameter setup as we used in the ETTh1 dataset. For PatchTST, we used the same hyperparameter setup as its original paper. We also conduct two ablation studies with varying lookback windows. We report the mean MAE, MSE and R2 score over 5 runs.

**Results.** Our results (Table 7) demonstrate that STanHop-Net consistently outperforms DLinear and PatchTST when multivariate correlations dominate the predictive information in input features. Importantly, our ablation studies show that increasing the lookback window size, which reduces the dominance of multivariate correlations, results in DLinear's performance becoming comparable to, rather than being consistently outperformed by, STanHop-Net. This explains why DLinear exhibits comparable performance to STanHop-Net in Table 1, when the datasets are not dominated by multivariate correlations.

**Channel-Independent and Channel-Dependent Methods.** Recent studies have compared Channel-Independent (CI) (Zeng et al., 2023; Nie et al., 2022) and Channel-Dependent (CD) (Yu et al., 2024; Liu et al., 2023; Chen et al., 2023; Luo and Wang, 2024) methods for time series prediction. CI methods, like **DLinear** and **PatchTST**, treat multivariate time series as multiple independent uni-variate tasks, often using simpler architectures like linear layers (Zeng et al., 2023). Conversely, CD methods leverage operations like convolution and channel-wise attention to capture inter-channel relationships and enhance performance.

---

[5]From a technical standpoint, this setting doesn't fall under CSR, yet we refer to it as CSR to emphasize its utilization of one time slice for predicting the next. Cross-sectional regression is valuable for understanding the simultaneous impact of variables across different entities (i.e., multivariate correlations), providing insights that are complementary to those obtained from autoregressive time-series analysis.

[6]https://www.dropbox.com/scl/fo/cg49nid4ogmd6f6sdbwqz/h?rlkey=jydf3ay1fzeotl5ws2s0cjnlc&dl=0

Surprisingly, CI methods perform well with minimal model complexity and parameters, and sometimes even outperform CD methods, as noted in (Zeng et al., 2023; Nie et al., 2022). However, (Han et al., 2023) points out that while CD benefits from increased capacity, CI's simple architecture and limited samples in current benchmarks can significantly boost performance. Existing datasets are often too small for CD models to showcase their full potential, leading to potential underestimation of CD model effectiveness on popular benchmarks like ETTh1, ILI etc.

Table 7: Comparison between DLinear, PatchTST and StanHop-Net on the synthetic dataset generated in (Reneau et al., 2023). This dataset is by construction a strongly correlated multivariate time series dataset. We report the mean MAE, MSE and R2 score over 5 runs. The $A \rightarrow B$ denotes the input horizon $A$ and prediction horizon $B$. **CSR (Cross-Sectional Regression):** Focuses on using single time step information from multivariate time series for predictions. **Ablation1:** With a prediction horizon of 1, the lookback window is increasing, thereby the dominance of multivariate correlations is decreasing. **Ablation2:** With a prediction horizon of 2, the lookback window is increasing, thereby the dominance of multivariate correlations is decreasing. Our results aligns with our expectations: STanHop-Net uniformly beats DLinear (Zeng et al., 2023) and PatchTST (Nie et al., 2022) in the Cross-Sectional Regression (CSR) setting. Importantly, our ablation studies show that increasing the lookback window size, which reduces the dominance of multivariate correlations, results in DLinear's performance becoming comparable to, rather than being consistently outperformed by, STanHop-Net. This explains why DLinear exhibits comparable performance to STanHop-Net in Table 1, when the datasets are not dominated by multivariate correlations.

| lookback $\rightarrow$ horizon | | DLinear | | | PatchTST | | | STanHop-Net | | |
|---|---|---|---|---|---|---|---|---|---|---|
| | | MSE | MAE | $R^2$ | MSE | MAE | $R^2$ | MSE | MAE | $R^2$ |
| CSR | $1 \rightarrow 1$ | 0.896 | 0.615 | 0.256 | 0.332 | **0.341** | 0.728 | **0.329** | 0.375 | **0.633** |
| | $1 \rightarrow 2$ | 1.193 | 0.794 | 0.001 | 0.460 | **0.417** | 0.625 | **0.417** | 0.428 | **0.552** |
| | $1 \rightarrow 4$ | 1.211 | 0.806 | -0.002 | 0.734 | 0.546 | 0.401 | **0.592** | **0.522** | **0.383** |
| | $1 \rightarrow 8$ | 1.333 | 0.868 | -0.100 | 1.227 | 0.744 | -0.001 | **0.812** | **0.636** | **0.182** |
| | $1 \rightarrow 16$ | 1.305 | 0.846 | -0.069 | 1.755 | 0.947 | -0.409 | **1.028** | **0.734** | **-0.058** |
| Ablation1 | $2 \rightarrow 1$ | 0.514 | 0.504 | 0.573 | 0.332 | **0.355** | 0.729 | **0.328** | 0.366 | **0.710** |
| | $4 \rightarrow 1$ | 0.373 | 0.417 | 0.690 | 0.332 | **0.357** | 0.729 | **0.328** | 0.364 | **0.712** |
| | $8 \rightarrow 1$ | 0.328 | 0.380 | **0.727** | 0.322 | 0.361 | 0.737 | **0.327** | 0.367 | **0.715** |
| | $16 \rightarrow 1$ | **0.319** | 0.372 | **0.736** | 0.324 | 0.371 | 0.735 | 0.323 | 0.361 | 0.717 |
| Ablation2 | $2 \rightarrow 2$ | 0.771 | 0.632 | 0.359 | 0.458 | 0.426 | **0.626** | 0.424 | **0.425** | 0.630 |
| | $4 \rightarrow 2$ | 0.423 | 0.439 | **0.645** | 0.460 | 0.429 | **0.624** | 0.410 | **0.415** | 0.643 |
| | $8 \rightarrow 2$ | 0.647 | 0.441 | 0.646 | 0.439 | 0.429 | **0.642** | 0.402 | **0.412** | 0.655 |
| | $16 \rightarrow 2$ | **0.419** | 0.435 | **0.652** | 0.447 | 0.446 | 0.635 | **0.435** | **0.433** | **0.626** |

# F    EXPERIMENT DETAILS

Here we present the details of experiments in the main text.

## F.1    EXPERIMENT DETAILS OF MULTIVARIATE TIME SERIES PREDICTIONS WITHOUT MEMORY ENHANCEMENTS

**Datasets.**    These datasets, commonly benchmarked in literature (Zhang and Yan, 2022; Wu et al., 2021; Zhou et al., 2021).

- **ETT (Electricity Transformer Temperature) (Zhou et al., 2021):** ETT records 2 years of data from two counties in China. We use two sub-datasets: ETTh1 (hourly) and ETTm1 (every 15 minutes). Each entry includes the "oil temperature" target and six power load features.
- **ECL (Electricity Consuming Load):** ECL records electricity consumption (Kwh) for 321 customers. Our version, sourced from (Zhou et al., 2021), covers hourly consumption over 2 years, targeting "MT 320".
- **WTH (Weather):** WTH records climatological data from approximately 1,600 U.S. sites between 2010 and 2013, measured hourly. Entries include the "wet bulb" target and 11 climate features.
- **ILI (Influenza-Like Illness):** ILI records weekly data on influenza-like illness (ILI) patients from the U.S. Centers for Disease Control and Prevention between 2002 and 2021. It depicts the ILI patient ratio against total patient count.
- **Traffic:** Traffic records hourly road occupancy rates from the California Department of Transportation, sourced from sensors on San Francisco Bay area freeways.

Table 8: Dataset Sources

| Dataset | URL |
| --- | --- |
| ETTh1 & ETTm1 | https://github.com/zhouhaoyi/ETDataset |
| ECL | https://archive.ics.uci.edu/dataset/321/electricityloaddiagrams20112014 |
| WTH | https://www.ncei.noaa.gov/data/local-climatological-data/ |
| ILI | https://archive.ics.uci.edu/ml/datasets/seismic-bumps |
| Traffic | https://www.kaggle.com/shrutimechlearn/churn-modelling |

**Training.**    We use Adam optimizer to minimize the MSE Loss. The coefficients of Adam optimizer, betas, are set to (0.9, 0.999). We continue training till there are `Patience = 3` consecutive epochs where validation loss doesn't decrease or we reach 20 epochs. Finally, we evaluate our model on test set with the best checkpoint on validation set.

**Hyperparameters.**    For hyperparameter search, for each dataset, we conduct hyperparameter optimization using the "Sweep" feature of Weights and Biases (Biewald et al., 2020), with 200 iterations of random search for each setting to identify the optimal model configuration. The search space for all hyperparameters are reported in Table 9.

Table 9: STanHop-Net hyperparameter space.

| Parameter | Distribution |
|---|---|
| Patch size $P$ | [6, 12, 24] |
| FeedForward dimension | [64, 128, 256] |
| Number of encoder layer | [1, 2, 3] |
| Number of pooling vectors | [10] |
| Number of heads | [4, 8] |
| Number of stacked STanHop blocks | [1] |
| Dropout | [0.1, 0.2, 0.3] |
| Learning rate | [5e-4, 1e-4, 1e-5, 1e-3] |
| Input length on ILI | [24, 36, 48, 60] |
| Input length on ETTm1 | [24, 48, 96, 192, 288, 672] |
| Input length on other dataset | [24, 48, 96, 168, 336, 720] |
| Course level | [2, 4] |
| Weight decay | [0.0, 0.0005, 0.001] |

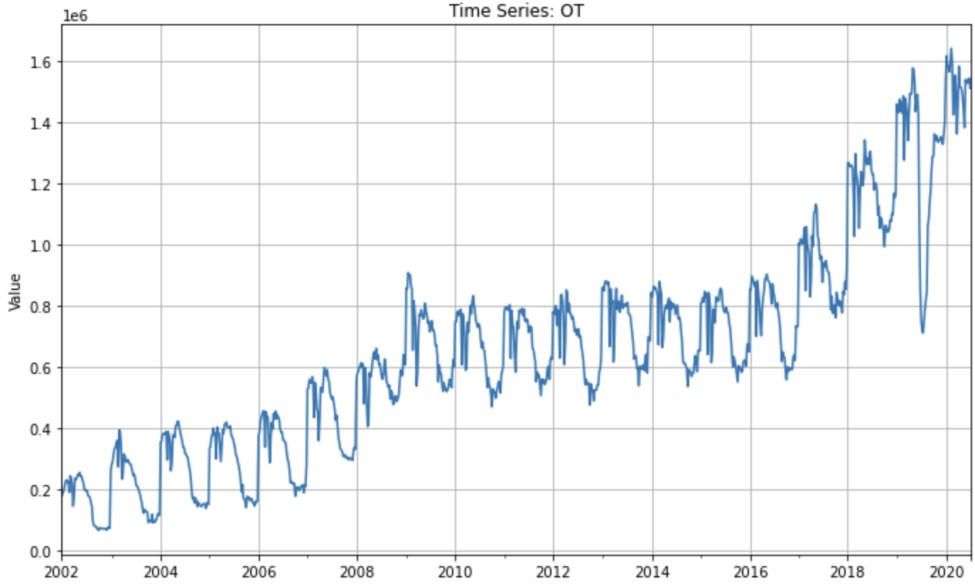

Figure 18: The visualization of ILI dataset "OT" variate.

## F.2 EXTERNAL MEMORY PLUGIN EXPERIMENT DETAILS

The hyperparameter of the external memory plugin experiment can be found in Table 9. For ILI_OT, we set the input length as 24, feed forward dimension as 32 and hidden dimension as 64. For prediction horizon 60, we set the input length as 48, feed-forward dimension as 128 and hidden dimension as 256. For ETTh1, we use the same hyperparameter set found via random search in Table 1.

For the "bad" external memory set intervals, we pick 40 and 200 for ILI_OT and ETTh1, which represents 40 timesteps (weeks) earlier and 200 timesteps (hours) earlier. For ILI dataset, we set the memory set size as 15 and for ETTh1, we set it as 20.

For **Case 3** (ETTh1), we select construct the external memory pattern with interval 168 timesteps earlier (equivalent to 1 week). For **Case 4** (ETTm1), we select construct the external memory pattern with interval 672 timesteps earlier (equivalent to 1 week).

## G ADDITIONAL THEORETICAL BACKGROUND

**Remark G.1.** Peters et al. (2019) provide a closed-form expression for $\alpha$-EntMax as

$$\alpha\text{-EntMax}(\mathbf{z}) = [(\alpha - 1)\mathbf{z} - \tau(\mathbf{z})]^{\frac{1}{\alpha-1}}, \tag{G.1}$$

where we denote $[t]_+ := \max\{t, 0\}$, and $\tau$ is the threshold function $\mathbb{R}^M \to \mathbb{R}$ such that $\sum_{\mu=1}^{M}[(\alpha - 1)\mathbf{z} - \tau(\mathbf{z})]^{\frac{1}{\alpha-1}} = 1$ satisfies the normalization condition of probability distribution.

