**Remark 3.1.** Peters et al. (2019) provide a close-form expression for $\alpha$-EntMax as
$$\alpha\text{-EntMax}(\mathbf{z}) = [(\alpha-1)\mathbf{z} - \tau(\mathbf{z})]^{\frac{1}{\alpha-1}}, \quad (3.4)$$
where we denote $[t]_+ := \max\{t, 0\}$, and $\tau$ is the threshold function $\mathbb{R}^M \to \mathbb{R}$ such that $\sum_{\mu=1}^M [(\alpha-1)\mathbf{z} - \tau(\mathbf{z})]^{\frac{1}{\alpha-1}} = 1$ satisfies the normalization condition of probability distribution.

---

[1]This energy function (3.2) is equivalent up to an additive constant.

Intuitively, $\Psi_\alpha^\star(\mathbf{p})$ is the convex conjugate of the Tsallis entropic regularizer $\Psi_\alpha(\mathbf{p})$ and hence

**Lemma 3.1.** $\nabla\Psi_\alpha^\star(\mathbf{z}) = \mathrm{ArgMax}_{\mathbf{p}\in\Delta^M}\left[\

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

 $\mathrm{EMB}(\mathbf{s}_i) = \mathbf{E}^{\mathrm{emb}} \mathbf{s}_i + \mathbf{E}^{\mathrm{pos}}(i) \in \mathbb{R}^{D_{\mathrm{emb}}}$, where $D_{\mathrm{emb}}$ is the embedding dimension, $\mathbf{E}^{\mathrm{emb}} \in \mathbb{R}^{D_{\mathrm{emb}} \times P}$, and $\mathbf{E}^{\mathrm{pos}} \in \mathbb{R}^{T/P \times D_{\mathrm{emb}}}$ is the positional encoding. When $T$ is not divisible to $P$, assuming $T = P \times C_n + c$ with $C_n, c \in \mathbb{N}_+$, we pad the sequence by repeating the first $c$ elements in the sequence. Consequently, this patching embedding significantly improves computational efficiency and memory usage.

### 4.2 STANHOP: SPARSE TANDEM HOPFIELD BLOCK

We introduce STanHop (**S**parse **Tan**dem **Hop**field) block which comprises one `GSHLayer`-based external memory plugin module, and two tandem sub-blocks of `GSH` layers to process both time and series dimensions, i.e. TimeGSH and SeriesGSH sub-blocks in Figure 2. In essence, STanHop not only sequentially extracts temporal and cross-series information of multivariate time series with (learnable) data-dependent sparsity, but also utilizes both acquired (in-context) representations and external stimulus through the memory plugin modules for the downstream prediction tasks.

Given a hidden vector, $\mathbf{R} \in \mathbb{R}^{C \times T \times D_{\mathrm{hidden}}}$, and its corresponding external memory set $\mathbf{Y} \in \mathbb{R}^{M \times C \times T \times D_{\mathrm{hidden}}}$, where $C$ denotes the channel number and $T$ denotes the number of time segments (patched time steps), To clarify, the `GSH` layer only operates on the last two dimensions, i.e., $t \in T$ and $d \in D_{\mathrm{hidden}}$. Thus, the operation $\mathrm{GSH}(\mathbf{Z}, \mathbf{Z})$ extracts information of the temporal dynamics of $\mathbf{Z}$ from the segmented time series. Here we define the dimensional transpose operation $\mathsf{T}$. For a given tensor $\mathbf{X} \in \mathbb{R}^{a \times b \times c}$, we have $\mathsf{T}_{abc}^{acb}(\mathbf{X}) := \mathbf{X}' \in \mathbb{R}^{a \times c \times b}$, i.e. this operation rearranges the dimensions of the original tensor $\mathbf{X}$ from $(a, b, c)$ to a new order $(a, c, b)$. Given a set of query pattern $\mathbf{Q} \in \mathbb{R}^{\mathrm{len}_Q \times D_{\mathrm{hidden}}}$, we define a single block of STanHop as

$$\mathbf{Z} = \mathrm{Memory}(\mathbf{R}, \mathbf{Y}), \qquad\qquad\qquad \text{(Memory Plugin Module, see Section 4.3)}$$

$$\mathbf{Z}^t = \mathsf{T}_{tch}^{cth}(\mathrm{LayerNorm}(\mathbf{Z} + \mathrm{FF}(\mathrm{GSH}(\mathbf{Z}, \mathbf{Z})))) \in \mathbb{R}^{T \times C \times D_{\mathrm{hidden}}}, \qquad\qquad \text{(Temporal GSH)}$$

$$\mathbf{Z}^p = \mathrm{GSHPooling}(\mathbf{R}^\star, \mathbf{Z}^t) \in \mathbb{R}^{T \times \mathrm{len}_Q \times D_{\mathrm{hidden}}}, \quad (\mathbf{R}^\star \text{ is learnable and randomly initialized})$$

$$\mathbf{Z}^c = \mathrm{GSH}(\mathbf{Z}^t, \mathbf{Z}^p) \in \mathbb{R}^{T \times C \times D_{\mathrm{hidden}}}, \qquad\qquad\qquad\qquad\qquad \text{(Cross-series GSH)}$$

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

$\square$

## C.4 THEOREM 3.1

*Proof.* We observe

$$
\|\mathcal{T}(\mathbf{x}) - \boldsymbol{\xi}_\mu\| - \|\mathcal{T}_{\text{Dense}}(\mathbf{x}) - \boldsymbol{\xi}_\mu\|
$$

$$
= \left\| \sum_{\nu=1}^{\kappa} \boldsymbol{\xi}_\nu \left[ (\alpha+\delta)\text{-entmax} \left( \beta \boldsymbol{\Xi}^\mathsf{T} \mathbf{x} \right) \right]_\nu - \boldsymbol{\xi}_\mu \right\| - \left\| \sum_{\nu=1}^{\kappa} \boldsymbol{\xi}_\nu \left[ \alpha\text{-entmax} \left( \beta \boldsymbol{\Xi}^\mathsf{T} \mathbf{x} \right) \right]_\nu - \boldsymbol{\xi}_\mu \right\| \quad \text{(C.13)}
$$

$$
\leq \left\| \sum_{\nu=1}^{\kappa} \left[ (\alpha+\delta)\text{-entmax}(\beta \boldsymbol{\Xi}^\mathsf{T} \mathbf{x}) \right]_\nu \boldsymbol{\xi}_\nu \right\| - \left\| \sum_{\nu=1}^{\kappa} \left[ \alpha\text{-entmax} \left( \beta \boldsymbol{\Xi}^\mathsf{T} \mathbf{x} \right) \right]_\nu \boldsymbol{\xi}_\nu \right\| \leq 0, \quad \text{(C.14)}
$$

which gives

$$