# OpenReview forum: "STanHop: Sparse Tandem Hopfield Model for Memory-Enhanced Time Series Prediction"
_ICLR.cc/2024/Conference — ICLR 2024 poster_

### Official Review · Reviewer_xuHy · 2023-10-27

**Soundness:** 3 good
**Presentation:** 2 fair
**Contribution:** 2 fair
**Rating:** 5
**Confidence:** 3

**Summary:**

The paper presents a new transformer-like architecture (STANHOP) that uses a new form of sparse Hopfield layers. The layers use a form of Tsallis α-entropy regularization so as to induce sparse encoding. The authors provide several theoretical results on the capacity and convergence speed of new Hopfield model. Besides the use of this layer, the STANHOP architecture adopts several new solutions, such as the use of Plug-and-Play and Tune-and-Play memory plugin modules.

The experiments are solely focused on timeseries prediction tasks and compare several versions of the STANHOP architecture with several existing Transformers baselines.

**Strengths:**

- The introduction and analysis of alpha-entropy regularized Hopfield models and of the associated Transformer layers is interesting and potentially very useful.
- The design of the proposed architecture has several interesting components.
- The paper is well written.
- The experiments compare the results with a large number of relevant baseline models.

**Weaknesses:**

The main issue with this work is that it tries to introduce too many innovations packed together in a single specialized architecture. This results in a paper that lacks a cohesive narrative, as it is unclear why these different novel parts should fit together. As a consequence, it is difficult for the reader to properly evaluate the merits of the different contributions. In my opinion, the main contribution is the introduction of the alpha-entropy regularized sparse Hopfield layers and their analysis. However, it is unclear to me why these layers should only be validated in multivariate timeseries prediction problems.  All in all, the specialized nature of the application does not match well with the general nature of the analysis in the first half of the paper.

While the experimental analysis on timeseries data is rigorous, the results are rather disappointing since the main focus of the paper was to solve this specialized problem. The proposed architecture performs worse than at least one baseline model (DLinear) and in general it performs very similarly to the other methods.

**Questions:**

What are the advantages of using the alpha-entropy regularization instead of the Gini entropy used in the original sparse Hopfield network paper?

---

> ### Author Response · Authors · 2023-11-13
> **Why Time Series Prediction?**
>
> >The main issue with this work is that it tries to introduce too many innovations packed together in a single specialized architecture. This results in a paper that lacks a cohesive narrative, as it is unclear why these different novel parts should fit together. As a consequence, it is difficult for the reader to properly evaluate the merits of the different contributions. In my opinion, the main contribution is the introduction of the alpha-entropy regularized sparse Hopfield layers and their analysis. However, it is unclear to me why these layers should only be validated in multivariate timeseries prediction problems. All in all, the specialized nature of the application does not match well with the general nature of the analysis in the first half of the paper.
>
> Thank you for your insightful feedback. We acknowledge the apparent lack of a cohesive narrative in our paper, particularly in the context of applying our innovations to multivariate time series (MTS) prediction.
>
> To address this, we emphasize that our choice of MTS as an application domain is grounded in its inherent characteristics. These characteristics align well with the capabilities of our proposed methods. Time series data are inherently noisy and sparse, as widely recognized in the literature [Masini, 2023; Fawaz, 2019; Wang, 2013; Ozaki, 2012]. Additionally, they exhibit a multi-resolution structure [Shabani, 2023; Cui, 2016], leading to an inductive bias towards multi-level sparsity and noise.
>
> Our work aims to leverage these characteristics. To do so, we
> * **(GSH:)** introduce the alpha-entropy regularized sparse Hopfield model (GSH). This model is specifically designed to handle varying degrees of sparsity and noise, which are prevalent in time series data.
> * **(STanHop:)** Furthermore, we propose the STanHop framework, utilizing the GSH model to effectively manage the multi-resolution structure of time series, particularly addressing the intrinsic multi-level sparsity and noise.
> * **(Memory Enhancement:)** Additionally, we introduce two novel memory enhancement methods: the Plug-and-Play module and the Tune-and-Play module. These are designed for train-less and task-aware memory enhancements, respectively, offering robust performance improvements, especially in scenarios where noise is dominant at certain resolutions.
>
> Therefore, we contend that the MTS domain is not merely a specialized application for our GSH model. Instead, it represents an ideal testbed that leverages the core strengths of GSH—its adaptable, learnable sparsity, and noise robustness—to address the unique challenges posed by different levels of sparsity and noise inherent in multi-resolution time series data. Our numerical experiments demonstrate the efficacy of our methodology in fulfilling this purpose.
>
> We hope this clarification better articulates the rationale behind our choice of MTS as the application domain for our proposed method.
>
> ---
> - [Masini, 2023] Masini, Ricardo P., Marcelo C. Medeiros, and Eduardo F. Mendes. "Machine learning advances for time series forecasting." Journal of economic surveys 37, no. 1 (2023): 76-111.
> - [Shabani, 2023] Shabani, Amin, Amir Abdi, Lili Meng, and Tristan Sylvain. "Scaleformer: iterative multi-scale refining transformers for time series forecasting." ICLR (2023)
> - [Fawaz, 2019] Ismail Fawaz, Hassan, Germain Forestier, Jonathan Weber, Lhassane Idoumghar, and Pierre-Alain Muller. "Deep learning for time series classification: a review." Data mining and knowledge discovery 33, no. 4 (2019): 917-963.
> - [Kaiser, 2017] Kaiser, Łukasz, Ofir Nachum, Aurko Roy, and Samy Bengio. "Learning to remember rare events." arXiv preprint arXiv:1703.03129 (2017).
> - [Cui, 2016] Cui, Zhicheng, Wenlin Chen, and Yixin Chen. "Multi-scale convolutional neural networks for time series classification." arXiv preprint arXiv:1603.06995 (2016).
> - [Wang, 2013] Wang, Zhaoran, Fang Han, and Han Liu. "Sparse principal component analysis for high dimensional multivariate time series." In Artificial Intelligence and Statistics, pp. 48-56. PMLR, 2013.
> - [Ozaki, 2012] Ozaki, Tohru. Time series modeling of neuroscience data. CRC press, 2012.

---

> ### Author Response · Authors · 2023-11-13
> **What's Better? Comparing with [Hu, 2023]**
>
> >What are the advantages of using the alpha-entropy regularization instead of the Gini entropy used in the original sparse Hopfield network paper?
>
> Thank you for the question. We acknowledge the lack of emphasis on the comparison with previous sparse Hopfield models.
> Our response highlights several key benefits:
>
> 1. **(More General:)** The proposed Generalized Sparse modern Hopfield Model (GSHM) is more general. GSHM is a generalization of [Hu, 2023] and includes [Hu, 2023] and [Ramsauer, 2020] as special cases when $\alpha=2$ and $\alpha=1$, respectively.
> 2. **(Adjustable Sparsity:)** Unlike the fixed sparsity mechanism in [Hu, 2023], GSHM allows for a continuously variable degree of sparsity. This is achieved by tuning the alpha parameter, offering greater flexibility to adapt to different data characteristics.
> 3.  **(Flexibility without Compromise:)**  Theoretically, we demonstrate that this added flexibility does not compromise the desirable properties of Sparse and Dense Modern Hopfield Models (MHM). The GSH model matches the performance of Sparse MHM, exhibiting tighter error bounds (as per Theorem 3.1), enhanced noise robustness (Corollary 3.1.1) than the  dense model, and a memory capacity that scales exponentially with $d$, similar to existing MHMs.
> 4. **(Practical Advantages of Tunable Sparsity:)** The ability to adjust sparsity offers two significant benefits in practical applications:
> **(a.)** It allows for addressing varying levels of sparsity across different stages of a deep learning pipeline, which is particularly vital in time series modeling.
> **(b.)** Sparsity can be learned from data, enabling the model to adapt to the intrinsic sparsity of the dataset and thereby improve performance. This advantage is evidenced in our comparative analysis of STanHop-D, STanHop-S, and STanHop.
> 5. **(Safe Default Choice in Uncertain Scenarios:)** Our experiments demonstrate that when there is limited insight into the dataset's characteristics, such as the level of sparsity and noise, using GSHM is a reliable default choice over Sparse or Dense MHM.
>
> In summary, the alpha-entropy regularization in GSHM offers a blend of theoretical robustness, practical flexibility, and general applicability. We hope that this contributes meaningfully to the ongoing development of modern Hopfield networks and attention/transformer-based methods.

---

> ### Author Response · Authors · 2023-11-16
> **Applying GSH to another application: Additional Experiments on Multiple Instance Learning using GSH (Sec. E.8)**
>
> >All in all, the specialized nature of the application does not match well with the general nature of the analysis in the first half of the paper.
>
> Thank you for pointing this out. We agree that, even though this work focuses on time series prediction methodology, part of the proposed model has a broader applicability. And it would be more beneficial if we can test that part (Generalized Sparse Modern Hopfield Model) with other applications.
>
> In response, we also evaluate the efficacy of the proposed GSH layer on **Multiple Instance Learning (MIL) tasks** following [Ramsauer et al., 2020; Hu et al., 2023]. **Our results are positive** (see the **newly added Sec. E.8** for more details). GSH converges faster than Sparse and Dense modern Hopfield models in most settings. This again verifies our theoretical results in Sec. 3.
>
> In essence, MIL is a type of supervised learning whose training data are divided into bags and labeling individual data points is difficult or impractical, but bag-level labels are available [Ilse et al., 2018]. We follow [Ramsauer et al., 2020; Hu et al., 2023] to conduct the multiple instance learning experiments on MNIST.
>
> **Results:** (please see the **newly added E.8, Fig 12-17**)
>
> Our results (shown in the **newly added E.8**) demonstrate that **GSH converges faster than the baselines in most settings**, as it can adapt to varying levels of data sparsity. **This is consistent with our theoretical findings in Theorem 3.1**, which state that GSH achieves higher accuracy and faster fixed-point convergence compared to the dense model.
>
> We hope these revisions address the reviewers' concerns and improve the overall quality of our paper.
>
> ---
> ### Experimental Details
>
> * **Model:** We first flatten each image and use a fully connected layer to project each image to the embedding space. Then, we perform GSHPooling and use linear projection for prediction.
>
> * **Baselines** We benchmark the Generalized Sparse modern Hopfield model (GSH) with the Sparse [Hu et al., 2023] and Dense [Ramsauer et al., 2020] modern Hopfield models.
>
> * **Dataset:** For the training dataset, we randomly sample 1000 positive and 1000 negative bags for each bag size. For test data, we randomly sample 250 positive and 250 negative bags for each bag size. We set the positive signal to the images of digit 9, and the rest as negative signals. We vary the bag size and report the accuracy and loss curves of `10` runs on both training and test data.
> * **Hyperparameters:** For hyperparameters, we use a hidden dimension of `256`, number of heads as `4`, dropout as `0.3`, training epochs as `100`, optimizer as AdamW, initial learning rate as `1e-4`, and we also use the cosine annealing learning rate decay.
>
>
> ---
> - [Ilse et al., 2018] Maximilian Ilse, Jakub Tomczak, and Max Welling. Attention-based deep multiple instance learning. In International conference on machine learning, pages 2127–2136. PMLR, 2018.
> - [Ramsauer et al., 2020] Hubert Ramsauer, Bernhard Schafl, Johannes Lehner, Philipp Seidl, Michael Widrich, Thomas Adler, Lukas Gruber, Markus Holzleitner, Milena Pavlovic, Geir Kjetil Sandve, et al. Hopfield networks is all you need. arXiv preprint arXiv:2008.02217, 2020.

---

> > ### Comment · Reviewer_xuHy · 2023-11-21
> >
> > Dear authors,
> >
> > I appreciate your work during this rebuttal, which I think increased the quality of the paper.
> >
> > After serious consideration, I decided to not increase my score as I believe that most of my original points still apply. However, I will not oppose acceptance if the other reviewers reach a consensus in that direction.

---

> > > ### Author Response · Authors · 2023-11-22
> > > **Thanks for your time and effort, new results (STanHop-Net uniformly beats DLinear) and some clarification questions**
> > >
> > > Dear Reviewer xuHy,
> > >
> > > Thank you for dedicating your time to reviewing our draft and going through the rebuttal responses.
> > >
> > > We would like to kindly remind you that [additional empirical results](https://openreview.net/forum?id=6iwg437CZs&noteId=p27WvzmvyD) have been provided, demonstrating that **STanHop-Net consistently outperforms DLinear in scenarios with strong correlations between variates**. Therefore, our method achieves comparable performance when correlations between variates are weak and demonstrates clear-cut superiority when these correlations are strong. We believe this have addressed your concern regarding outperforming the baselines.
> > >
> > > On the other hand, is that possible you can elaborate more on why you still feel the proposed method is disconnected from the time series prediction problem, especially conditioned on our response [Why Time Series Prediction?](https://openreview.net/forum?id=6iwg437CZs&noteId=3wSnYRGSTu) ?
> > >
> > > As a group of practitioners in time series modeling, we understand that the first half of this paper (GSH model) is generally applicable. Yet, in the 2nd half, the properties of the GSH model and the proposed STanHop-Net are used, in a natural and well-motivated way, to address the fundamental hardness of time series prediction and obtain positive empirical results. If there is a clear logical gap or inconsistency that we overlooked, we would really like to address it, even for the sake of future submissions.
> > >
> > > Once again, thanks for your time and efforts. Your constructive comments are important in improving our work.
> > >
> > > Best,
> > >
> > > Authors

---

> ### Author Response · Authors · 2023-11-22
> **STanHop-Net Outperforms DLinear in Settings Dominated by Multivariate Correlations: A Cross-Sectional Regression Case Study (Part1)**
>
> >While the experimental analysis on timeseries data is rigorous, the results are rather disappointing since the main focus of the paper was to solve this specialized problem. The proposed architecture performs worse than at least one baseline model (DLinear) and in general it performs very similarly to the other methods.
>
> Thank you for raising this point. This issue indeed deserves further explanation and investigation.
>
> We apologize for any confusion and believe there might have been an oversight. While DLinear ranks as top-2 in only 36 settings, STanHop ranks as top-2 in 47. **We believe that our method is, at the very least, on par with DLinear, if not better.**
>
> **In the newly added Section E.9, we showcase a practical and common setting (cross-sectional regression) where STanHop-Net consistently outperforms DLinear.** Moreover, through ablation studies, we pinpoint the reason why DLinear shows comparable performance in our main text: the common benchmark datasets are not dominated by multivariate correlations among time series. Specifically, in these datasets, the autoregressive correlation is the primary source of predictive information in the input features.
>
> Therefore, in settings where multivariate correlations dominate the predictive information in the input features, the superiority of STanHop-Net becomes more pronounced.
>
> ---
> ## Newly Added Sec. E.9
>
> It is true that the performance of STanHop-Net in the main text, as presented in Table 1, does not show a clear superiority over DLinear [Zeng et al., 2023]. We attribute this to the nature of the common benchmark datasets in Table 1, which are not dominated by multivariate correlations.
>
> To verify our conjecture, we employ a strongly correlated multivariate time series dataset as a test bed, representing a practical scenario where multivariate correlations are the predominant source of predictive information in input features. In such a scenario, following the same setting in Section 5.1, **our experiments show that STanHop-Net consistently outperforms DLinear.**
>
> Specifically, we follow the experimental settings outlined in Section 5.1, but with a specific focus on cases involving small lookback windows. This emphasis aims to reduce autoregressive correlation in data with smaller lookback windows, thereby increasing the dominance of multivariate correlations. Such a scenario is referred to as Cross-Sectional Regression 4 (CSR) in time series modelling [Reneau et al., 2023; Fama and French, 2020; Andrews, 2005] and [Wooldridge et al., 2016, Part 1].
>
> **Dataset:**
> We evaluated our model on the synthetic dataset generated in the ICML2023 Feature Programming paper [Reneau et al., 2023]. Feature programming is an automated, programmable method for feature engineering, producing a large number of predictive features from any given input time series. These genereated features, termed *extended features*, are inherently highly correlated. The synthetic dataset, which includes 44 extended features derived from the taxi dataset, is thus a strongly correlated multivariate time series dataset. Special thanks to the authors of [Reneau et al., 2023] for sharing this dataset. [Dataset Link](https://www.dropbox.com/scl/fo/cg49nid4ogmd6f6sdbwqz/h?rlkey=jydf3ay1fzeotl5ws2s0cjnlc&dl=0)
>
> **Baseline:**
> Our primary comparison is with DLinear [Zeng et al., 2023], which demonstrated comparable performance in Table 1 of our main text.
>
> **Setting:**
> For both STaHop-Net and DLinear, we employed the same hyperparameter setup as in the ETTh1 dataset. Additionally, we conducted two ablation studies with varying lookback windows. We report the mean MAE, MSE and R2 score over 5 runs.
>
> **Results:**
> Our results (below table or (Table 1)) show that STanHop-Net consistently outperforms DLinear when multivariate correlations dominate the predictive information in input features. Notably, our ablation studies reveal that increasing the lookback window size, thus reducing the dominance of multivariate correlations, brings DLinear's performance closer to that of STanHop-Net. This explains why DLinear exhibits comparable performance to STanHop-Net in Table 1, when the datasets are not dominated by multivariate correlations.
>
> ---
> -  [Zeng et al., 2023] Ailing Zeng, Muxi Chen, Lei Zhang, and Qiang Xu. Are transformers effective for time series forecasting? In Proceedings of the AAAI conference on artificial intelligence, volume 37, pages 11121–11128, 2023
> - [Reneau et al., 2023] Alex Reneau, Jerry Yao-Chieh Hu, Chenwei Xu, Weijian Li, Ammar Gilani, and Han Liu. Feature programming for multivariate time series prediction. In Proceedings of the 40th International Conference on Machine Learning, volume 202 of Proceedings of Machine Learning Research, pages 29009–29029. PMLR, 23–29 Jul 2023. URL https://arxiv.org/abs/2306.06252.

---

> ### Author Response · Authors · 2023-11-22
> **STanHop-Net Outperforms DLinear in Settings Dominated by Multivariate Correlations: A Cross-Sectional Regression Case Study (Part2)**
>
> ## Performance Comparison: DLinear vs. STanHop-Net (Newly Added Table 7 in Sec. E.9)
>
> **Caption.** This dataset is by construction a strongly correlated multivariate time series
> dataset (generated in [Reneau et al., 2023]). The A → B denotes the input horizon A and prediction horizon B. **CSR (Cross-Sectional Regression):** Focuses on using single time step information from multivariate time series for predictions. **Ablation1:** With a prediction horizon of 1, the lookback window is increasing, thereby
> the dominance of multivariate correlations is decreasing. **Ablation2:** With a prediction horizon of 2, the lookback
> window is increasing, thereby the dominance of multivariate correlations is decreasing.
>
> | Lookback Window → Prediction Horizon | MSE (DLinear) | MAE (DLinear) | R^2 (DLinear) | MSE (STanHop-Net) | MAE (STanHop-Net) | R^2 (STanHop-Net) |
> |---------------------------------------|---------------|---------------|---------------|-------------------|-------------------|-------------------|
> | CSR 1 → 1                              | 0.896         | 0.615         | 0.256         | **0.329**         | **0.375**         | **0.633**         |
> | CSR 1 → 2                              | 1.193         | 0.794         | 0.001         | **0.417**         | **0.428**         | **0.552**         |
> | CSR 1 → 4                              | 1.211         | 0.806         | -0.002        | **0.592**         | **0.522**         | **0.383**         |
> | CSR 1 → 8                              | 1.333         | 0.868         | -0.100        | **0.812**         | **0.636**         | **0.182**         |
> | CSR 1 → 16                             | 1.305         | 0.846         | -0.069        | **1.028**         | **0.734**         | **-0.058**        |
> | Ablation1 2 → 1                        | 0.514         | 0.504         | 0.573         | **0.328**         | **0.366**         | **0.710**         |
> | Ablation1 4 → 1                        | 0.373         | 0.417         | 0.690         | **0.328**         | **0.364**         | **0.712**         |
> | Ablation1 8 → 1                        | 0.328         | 0.380         | **0.727**     | **0.327**         | **0.367**         | 0.715             |
> | Ablation1 16 → 1                       | **0.319**     | 0.372         | **0.736**     | 0.323             | **0.361**         | 0.717             |
> | Ablation2 2 → 2                        | 0.771         | 0.632         | 0.359         | **0.424**         | **0.425**         | **0.630**         |
> | Ablation2 4 → 2                        | 0.423         | 0.439         | **0.645**     | **0.410**         | **0.415**         | 0.643             |
> | Ablation2 8 → 2                        | 0.647         | 0.441         | 0.646         | **0.402**         | **0.412**         | **0.655**         |
> | Ablation2 16 → 2                       | **0.419**     | 0.435         | **0.652**     | 0.435         | **0.433**             | 0.626             |
>
> **Note:** Bold values indicate superior performance.
>
> Our results aligns with our expectations: STanHop-Net uniformly beats DLinear [Zeng et al., 2023] in the Cross-Sectional Regression (CSR) setting. Importantly, our ablation studies show that increasing the lookback window
> size, which reduces the dominance of multivariate correlations, results in DLinear’s performance
> becoming comparable to, rather than being consistently outperformed by, STanHop-Net. This explains why DLinear exhibits comparable performance to STanHop-Net in Table 1, when the datasets
> are not dominated by multivariate correlations.
>
> This experiment has successfully demonstrated our clear-cut superiority over DLinear. We hope this addresses the reviewer's concerns regarding outperforming the baselines. Thank you for your time and effort in reviewing our paper.

---

### Official Review · Reviewer_u61L · 2023-11-03

**Soundness:** 3 good
**Presentation:** 3 good
**Contribution:** 2 fair
**Rating:** 8
**Confidence:** 4

**Summary:**

In this work, authors augmented hopefield network with external memory. They introduce the Sparse Tandem Hopfield Network which was tested on  multivariate time series prediction task and the proposed model exhibits improved memory capabilities. To be specific the memory module has Plug-and-Play module and a Tune-and-Play module for train-less and task-aware memory improvements. The model is theoretically motivated and series of simulation studies show proposed model achieves consistent gain compared to other transformer-based models.

**Strengths:**

1. Well-written paper
2. Proofs are incremental, mainly based on Hu's et al work in 2023, however, the presentation is neat.
3. good set of experiments
4. Hypothesis is backed by ablation study

**Weaknesses:**

1. Comparison against stateful models such as RNNs is missing, also, the proposed work focuses on external memory, a comparison against memory-augmented is needed.
2. Explanations for Lemma 3.1, 3.2, and other lemmas in the main paper, can be written in a better way. Rather than stating to refer to proof, you should try and provide the simplest explanation of what each proof talks about.

**Questions:**

Adding memory to NNs is not a new concept, it has been out since early 90’s [1], and are even extended to modern NNs [2-4]. Similar to Hopfield networks these networks are shown to reach stable point [5-6]. Thus it is important to mention these relevant work.

Second, improving the memory capability of Hopefield network is widely studied these days [7-9], thus comparison should be done with these relevant approaches, especially 8 and 9.

I would like to see a comparison against RNNs and memory-augmented RNNs, given that the proposed model is focused on time-series which is a stateful problem.

Finally, what is the memory footprint? How much time model takes per epoch? Model size?
Do you observe faster convergence? How stable is the model? All these questions should be addressed.

I would like to see the variance of the model, including baseline models.



1.	Das, S., Giles, C. and Sun, G.Z., 1992. Using prior knowledge in a NNPDA to learn context-free languages. Advances in neural information processing systems, 5.
2.	Joulin, A. and Mikolov, T., 2015. Inferring algorithmic patterns with stack-augmented recurrent nets. Advances in neural information processing systems, 28.
3.	Graves, A., Wayne, G. and Danihelka, I., 2014. Neural turing machines. arXiv preprint arXiv:1410.5401.
4.	Weston, J., Chopra, S. and Bordes, A., 2014. Memory networks. arXiv preprint arXiv:1410.3916.
5.	Stogin, J., Mali, A. and Giles, C.L., 2020. A provably stable neural network Turing Machine. arXiv preprint arXiv:2006.03651.
6.	Mali, A.A., Ororbia II, A.G. and Giles, C.L., 2020. A neural state pushdown automata. IEEE Transactions on Artificial Intelligence, 1(3), pp.193-205.
7.	Millidge, B., Salvatori, T., Song, Y., Lukasiewicz, T. and Bogacz, R., 2022, June. Universal hopfield networks: A general framework for single-shot associative memory models. In International Conference on Machine Learning (pp. 15561-15583). PMLR.
8.	Hillar, C.J. and Tran, N.M., 2018. Robust exponential memory in Hopfield networks. The Journal of Mathematical Neuroscience, 8(1), pp.1-20.
9.	Ota, T., Sato, I., Kawakami, R., Tanaka, M. and Inoue, N., 2023, April. Learning with Partial Forgetting in Modern Hopfield Networks. In International Conference on Artificial Intelligence and Statistics (pp. 6661-6673). PMLR.

---

> ### Author Response · Authors · 2023-11-14
> **Discussion of Memory-Augmented NNs in Related Work Section (Sec. B)**
>
> Thanks for the references. We have added a discussion of memory-augmented neural networks into the related work section. Please see **Sec. B** of the latest revision.
>
> We quote the paragraph of relevance to our work here:
>
> ...[Kaiser, 2017] focus on enhancing the recall of rare events and  is particularly notable for its exploration of memory-augmented neural networks designed to improve the retention and recall of infrequent but significant occurrences, highlighting the potential of external memory modules in handling rare data challenges.
> This is achieved through a unique soft attention mechanism, which dynamically assigns relevance weights to different memory entries, enabling the model to draw on a broad spectrum of stored experiences.
> This approach not only facilitates the effective recall of rare events but also adapts to new data, ensuring the memory's relevance and utility over time.
>
> In all above methods, [Kaiser, 2017]  is closest to this work.
> However, our approach diverges in two key aspects:
> - **(Enhanced Generalization):** Firstly, our external memory enhancements are external plugins with an option for fine-tuning.
> This design choice avoids over-specialization on rare events, thereby broadening our method's applicability and enhancing its generalization capabilities across various tasks where the frequency and recency of data are less pivotal.
> - **(Adaptive Response over Rare Event Memorization):** Secondly, our approach excels in real-time adaptability.
> By integrating relevant external memory sets tailored to specific inference tasks, our method can rapidly respond and improve performance, even without the necessity of prior learning.
> This flexibility contrasts with the primary focus on *memorizing* rare events in [Kaiser, 2017].
>
> ---
> -  [Kaiser, 2017] Łukasz Kaiser, Ofir Nachum, Aurko Roy, and Samy Bengio. Learning to remember rare events. arXiv preprint arXiv:1703.03129, 2017

---

> ### Author Response · Authors · 2023-11-15
> **Memory footprint, time per epoch, model size, faster convergence, variance of results and proof sketchs**
>
> Thanks for asking the following clarification questions. In response, we have revised the draft to enhance clarity and address the points raised as follows.
>
> >Finally, what is the memory footprint?
>
> In the **newly added Sec. E.5**, we report the footprint of memory and GPU usage. The results show that the difference in memory and GPU usage between the proposed GSH and the vanila **dense** modern Hopfield layers is **negligible**. Please see Fig. 10 and Fig. 11 for more details.
>
> >How much time model takes per epoch?
>
> The time per epoch varies depending on the specific setting.
>
> As a reference, when using a batch size of 32 and an input length of 168, STanHop-Net requires approximately 2 minutes per epoch for the ETTh1 dataset. This duration is roughly equivalent to that of using dense modern Hopfield layers, such as STanHop-Net (D). Together with our results from Thm. 3.1 and Sec. E.1 — the tighter fixed point convergence of the Generalized Sparse modern Hopfield (GSH) model leads to fewer epochs required for convergence  — **STanHop-Net using the GSH model generally requires less time to converge** (than those of using dense Hopfield layers.)
>
> >Model size?
>
> The total number of parameters of the STanHop-Net is ~0.78 million. We have added this in the **new Sec. E.6** of the latest revision.
>
> >Do you observe faster convergence?
>
> The tighter fixed point convergence of the Generalized Sparse modern Hopfield (GSH) model leads to fewer epochs required for convergence. Thus, yes, our empirical results (Fig. 4 in Sec. E.1) verifies our theoretical results (Thm. 3.1) --- the GSH model converges faster than the dense modern Hopfield model.
>
> >I would like to see the variance of the model, including baseline models.
>
> In Multivariate Time Series (MTS) prediction research, it is common to exclude the variance due to compactness of papers.
>
> This convention is observed in all our baseline models — including Crossformer, Informer, FEDformer, DLinear, and Autoformer — none of which report variance in their respective papers.  As most of our baseline values are directly quoted from these original papers, we adhere to the same convention for consistency.
>
> However, **we did report the maximum variance across all settings of our model in our table caption, stating “with variance omitted as they are all <=0.44%”.**
>
> >Explanations for Lemma 3.1, 3.2, and other lemmas in the main paper, can be written in a better way. Rather than stating to refer to proof, you should try and provide the simplest explanation of what each proof talks about.
>
> Thanks for pointing this out. We apologize for any confusion caused. In earlier drafts, proof sketches were included in the main text, but they were mostly removed or relocated to the appendix due to page constraints. We have now added the removed ones back in Sec.C. Please see revised version for details.
>
> To clarify:
>
> - **Lemma 3.1**: The high-level proof sketch begins in Section C.1, followed by detailed elaboration.
> The explanation/intuition/purpose are outlined directly below Lemma 3.1:
> >Lemma 3.1 demonstrates how the energy function (3.2) aligns with the overlap-function construction of Hopfield models, referenced in Hu et al. (2023) and Ramsauer et al. (2020). The corresponding retrieval dynamics satisfying the monotonicity property (T1) are then introduced.
>
> - **Lemma 3.2**: The proof sketch for Lemma 3.2 is presented at the start of Section C.1, distinctly marked by a gray block for emphasis and ease of reference.
> >Our proof is built on (Hu et al., 2023, Lemma 2.1). We first derive T by utilizing Lemma 3.1 and Remark G.1, along with the convex-concave procedure (Yuille and Rangarajan, 2003; 2001). Then, we show the monotonicity of minimizing H with T by constructing an iterative upper bound of H which is convex in x t+1 and thus, can be lowered iteratively by the convex-concave procedure.
> - **Lemma 3.3**: he proof sketch for Lemma 3.2 is presented as the first sentence of the proof.
> >Since the energy function H monotonically decreases with respect to increasing t in Lemma 3.2, we can follow [(Hu et al., 2023), Lemma 2.2] to guarantee the convergence property of T by checking the necessary conditions of Zangwill’s global convergence.
> - **Lemma 3.4**: the proof sketch is inside and highlighted by the gray blcok:
> >Our proof, built on (Hu et al., 2023, Lemma 2.1), proceeds in 3 steps:...
>
> We hope these changes and clarifications enhance the clarity of explanations for our theoretical results.
>
> We hope that our responses adequately address the reviewer's concerns, and we look forward to any further feedback. Thank you for your time and consideration.

---

> ### Author Response · Authors · 2023-11-15
> **Additional Comparison to Some Recent Works**
>
> We appreciate the reviewer for directing our attention to these relevant references. Please see below added discussions.
>
> >Second, improving the memory capability of Hopefield network is widely studied these days [7-9], thus comparison should be done with these relevant approaches, especially 8 and 9.
>
> **The major difference between our work and that presented in [8] lies in the scope of theoretical assumptions.** Our analysis avoids overly strong assumptions on the model and data, contrasting with the focus on combinatorial problems with strong assumptions in [8]. While both studies demonstrate exponential memory capacity relative to pattern size, the practical relevance of these findings differs significantly. **Our results hold greater practical applicability** in areas we prioritize, such as developing new methodologies and theoretical insights for attention/transformer models and large foundation models. However, this approach comes with a trade-off: our analysis may not provide as detailed characterizations as those found in [8], such as the derivation of Shannon’s bound for low-density error-correcting codes.
>
> In contrast to [7], which offers a unified framework for associative memory models by dissecting retrieval dynamics into three components: similarity, separation (distinct from our paper's definition), and projection, our work takes a different approach. [7] introduces the Manhattan distance as an alternative similarity metric and highlights its empirical relevance. However, while they connect their framework to [Kortov and Hopfield, 2020], their theoretical exploration, particularly in terms of retrieval error, robustness, and memory capacity, remains limited. **Our work addresses this gap by providing a comprehensive theoretical understanding of a range of separation functions, employing the Euclidean distance as the similarity metric.**
>
> In [9], the authors introduce the Partial Forgetting Function as a novel projection function within the frameworks of [8] and [Kortov and Hopfield, 2020]. Their approach, which includes the development of the Partially Forgetting Attention mechanism, effectively addresses the phenomenon of partial forgetting in the human brain. Similar to the above discussion, our research complements [9] by investigating theoretical properties. **Although our study does not explore changes in the projection map — a modification that might intuitively lead to efficient sparse modern Hopfield models, as suggested in [Beltagy, 2020] — we recognize this as a compelling and significant direction for future research.**
>
> We hope these discussions have addressed your concerns. Thank you!
>
> ---
> - [Kortov and Hopfield, 2020] Krotov, Dmitry, and John Hopfield. "Large associative memory problem in neurobiology and machine learning." arXiv preprint arXiv:2008.06996 (2020).
> - [Beltagy, 2020] Beltagy, Iz, Matthew E. Peters, and Arman Cohan. "Longformer: The long-document transformer." arXiv preprint arXiv:2004.05150 (2020).

---

> ### Author Response · Authors · 2023-11-15
> **Stability Analysis (Sec. E.7)**
>
> Thanks for pointing out the lack of stability analysis. We have conducted it in the **newly added Sec. E.7**. We quote the results as follows:
>
> We conduct experiments exploring the parameter sensitivity of  STanHop-Net on the ILI dataset.
>
> Our results (in Table 5 and Table 6 of Sec.  E.7) show that **STanHop-Net is quite stable.** It is **not** sensitive to hyperparameter changes.
>
> The way that we conduct the hyperparameter sensitivity analysis is that to show the model's sensitivity to a hyperparameter $h$, we change the value of $h$ in each run and keep the rest of the hyperparameters' values as default and we record the MAE and MSE on the test set. We train and evaluate the model 3 times for 3 different values for each hyperparameter. We analyzed the model's sensitivity to 7 hyperparameters respectively.
>
> * **MAE Results:** MAEs for each value of the hyperparameter with the rest of the hyperparameters as default
> | lr           | seg_len      | window_size | e_layers | d_model     | d_ff        | n_heads |
> |--------------|--------------|-------------|----------|-------------|-------------|---------|
> | (1e-3, 1e-4, 1e-5) | (6, 12, 24) | (2,4,8)     | (3,4,5)  | (32, 64, 128) | (64, 128, 256) | (2,4,8) |
> | 1.313        | 1.313        | 1.313       | 1.313    | 1.313       | 1.313       | 1.313   |
> | 1.588        | 1.311        | 1.285       | 1.306    | 1.235       | 1.334       | 1.319   |
> | 1.673        | 1.288        | 1.368       | 1.279    | 1.372       | 1.302       | 1.312   |
>
> * **MSE Results:** MSEs for each value of the hyperparameter with the rest of the hyperparameters as default.
> | lr           | seg_len      | window_size | e_layers | d_model     | d_ff        | n_heads |
> |--------------|--------------|-------------|----------|-------------|-------------|---------|
> | (1e-3, 1e-4, 1e-5) | (6, 12, 24) | (2,4,8)     | (3,4,5)  | (32, 64, 128) | (64, 128, 256) | (2,4,8) |
> | 3.948        | 3.948        | 3.948       | 3.948    | 3.948       | 3.948       | 3.948   |
> | 5.045        | 3.968        | 3.877       | 3.915    | 3.566       | 3.983       | 3.967   |
> | 5.580        | 3.865        | 4.267       | 3.834    | 4.078       | 3.866       | 3.998   |
>
> * Default Values of Hyperparameters in Sensitivity Analysis:
> | Parameter                          | Default Value |
> |------------------------------------|---------------|
> | seg_len (patch size)               | 6             |
> | window_size (coarse level)         | 2             |
> | e_layers (number of encoder layers)| 3             |
> | d_model                            | 32            |
> | d_ff (feedforward dimension)       | 64            |
> | n_heads (number of heads)          | 2             |
>
> We hope that the analysis (and modifications) provided in this response address the reviewer's concerns.
> We're open to any further questions or clarifications you might have about our work.

---

> ### Author Response · Authors · 2023-11-17
> **Response to "comparison against stateful and memory-augmented models"**
>
> >I would like to see a comparison against RNNs and memory-augmented RNNs, given that the proposed model is focused on time-series which is a stateful problem.
>
> Our paper proposes a multivariate time series prediction model. To show the **prediction performance** of our model, we believe it is more reasonable and sufficient to compare our model’s prediction performance with the latest and best multivariate time series prediction models. Namely, **a newly proposed time series prediction model should always benchmark with SOTA baselines, regardless of whether the proposed model is stateful or stateless**. In light of this, we have conducted the comparison experiments and included the results in Table 1 in the original submission. The stateful/stateless models the reviewer mentioned refer to recurrent neural networks because they have the RNN-specific intern state. After a thorough literature review, we are not aware of any published stateful-RNN time series prediction model with top-tier prediction performance in major conferences or journals. Thus, we kindly ask the reviewer for specific suggestions on what additional literature/baseline should be considered.
>
> On the other hand, the external memory module is only one component of our proposed method. Because there are other variables of the model affecting the performance, **directly comparing our model with other memory augmented models is not an apple-to-apple comparison** (as detailed below) and cannot provide a clear demonstration on the effectiveness of the external memory module. To more effectively showcase its impact, we believe an ablation study is both reasonable and sufficient. Thus, we have conducted this study and included the results in Table 2 of our original submission.
>
> Here, we highlight the differences and advantages of our external memory design compared to previous similar approaches. To the best of our understanding, in previous works, their main focus is to learn an accessible/informative memory set from the training data. However, in our paper, we focused on allowing models to accept external stimuli during inference. Besides, the main novelty of our external memory design lies in utilizing the memory retrieval mechanism of Hopfield models. This design enables us to incorporate both feature-level and label-level auxiliary information, corresponding to the Plug-and-Play and Tune-and-Play modules, respectively. In contrast, previous works involving memory-augmented RNNs typically utilize an additional learnable weight matrix and incorporate auxiliary information only at the feature level.
>
> ---
>
> ### **TL;DR**
>
> Directly comparing our model with other memory augmented models is not an apple-to-apple comparison.
>
> **Different Focus:**
> - Previous memory-augmented RNN methods emphasize learning to memorize specific data from the training set  [Kaiser, 2017;Santoro, 2016;Sukhbaatar, 2015;Graves, 2014].
> - Our approach focuses on enabling models to incorporate external stimuli during inference for a performance boost via enhanced memory.
>
> **Richer and Different Axillary Information from Memory:**
> - Previous methods leverage auxiliary information within the feature space  [Kaiser, 2017;Santoro, 2016;Sukhbaatar, 2015;Graves, 2014; Weston, 2014].
> - STanHop-Net's Tune-and-Play memory module retrieves pseudo-labels, incorporating both label-specific and feature space information to enhance predictions.
>
> **Different Training Requirements:**
> - Previous works require extensive training of the memory module to access auxiliary information, making it challenging to modify for different tasks without retraining the entire system  [Kaiser, 2017;Santoro, 2016].
> - STanHop-Net doesn't demand complete memory module retraining and aims to accept external stimuli, allowing for the inclusion of human-curated patterns.
>
> **Module Flexibility:**
> - Previous works are not compatible with plug-and-play memory integration, and the memory pattern is fixed after training.
> - Our Plug-Memory module doesn't require training, facilitating easy integration of external memory. This indicates that our method tackles different challenges and possesses broader applicability compared to prior approaches.
>
> ---
>
> Thank you for your time and valuable feedback. We hope that above clarifications address the reviewer's concerns.
>
> ---
> - [Kaiser  2017]  Łukasz Kaiser, Ofir Nachum, Aurko Roy, Samy Bengio. “Learning to Remember Rare Events”
> - [Santoro 2016] Adam Santoro, Sergey Bartunov, Matthew Botvinick, Daan Wierstra, Timothy Lillicrap, “One-shot Learning with Memory-Augmented Neural Networks.”
> - [Sukhbaatar 2015] Sainbayar Sukhbaatar, Arthur Szlam, Jason Weston, Rob Fergus, “End-To-End Memory Networks”
> - [Graves 2014] Alex Graves, Greg Wayne, Ivo Danihelka, “Neural Turing Machines”
> -  [Joulin 2015] Armand Joulin, Tomas Mikolov, “Inferring Algorithmic Patterns with Stack-Augmented Recurrent Nets”
> - [Weston 2014] Jason Weston, Sumit Chopra, Antoine Bordes, “Memory Networks”

---

> ### Author Response · Authors · 2023-11-21
> **A Gentle Reminder**
>
> Dear Reviewer u61L,
>
> Thank you for your review and valuable insights on our work. We have addressed all your comments in our detailed response and believe we have resolved all concerns. Should any issues remain, we are ready to provide additional clarifications.
>
> As the rebuttal phase is nearing its deadline, we're looking forward to engaging in a timely discussion.
>
> Thank you again for your time and effort!
>
> Best regards,
>
> Authors

---

> > ### Comment · Reviewer_u61L · 2023-11-22
> > **Thank you for your response**
> >
> > I apologize for the delay. I have read the rebuttal and the authors have done a good job in answering them. Thus I increase my rating.

---

> > > ### Author Response · Authors · 2023-11-22
> > > **Thank you!**
> > >
> > > Dear Reviewer u61L,
> > >
> > > We are glad to hear that our revisions have addressed your concerns.
> > > Thank you again for your constructive comments. They are important in improving our work. We truly appreciate your thorough review.
> > >
> > > Best,
> > >
> > > Authors

---

### Official Review · Reviewer_ntR2 · 2023-11-04

**Soundness:** 2 fair
**Presentation:** 2 fair
**Contribution:** 2 fair
**Rating:** 5
**Confidence:** 2

**Summary:**

The work introduces a novel neural network model called STanHop-Net, which is based on the Hopfield model and offers memory-enhanced capabilities for multivariate time series prediction. The model incorporates sparsity and external memory modules, which enable it to respond quickly to sudden events and achieve good results in both synthetic and real-world settings.

**Strengths:**

1. The paper is is well-written and comprehensive.
2. The authors present case studies demonstrating the effectiveness of the model in practical applications.

**Weaknesses:**

1. The paper does not provide available code source to reproduce the experiments.
2. The paper's contributions are limited in scope.
3. The model is too complex, it's may difficult to optimize.

**Questions:**

1. How does the sparsity of STanHop-Net affect its performance and memory usage?
2. Can the author provide the time complexity and number of the parameters of the model?
3. Can the author provide the limitation section?

---

> ### Author Response · Authors · 2023-11-13
> **Q2 & Q3 & Optimizability**
>
> >Can the author provide the time complexity and number of the parameters of the model?
>
> Thanks for pointing this out. We report the numbers here and update the draft accordingly. Please see the **new Sec. E.6** of the latest revision.
>
> **Number of Parameters:** 0.78 million
>
> **Time Complexity:**
> Let $T$ be the number of input length on the time dimension, $D_{\text{hidden}}$ be the hidden dimension, $P$ be patch size, $D_{\text{out}}$ be the prediction horizon, $\text{len}_Q$ be the size of query pattern in $\mathtt{GSHPooling}$.
>
> - **Patch Embedding:** $O(D_{\text{hidden}} \times T) = O(T)$
> - **Temporal and Cross Series GSH:** $O(D_{\text{hidden}}^2 \times T^2 \times P^{-2}) = O(T^2 \times P^{-2}) = O(T^2)$
> - **Coarse Graining:** $O(D_{\text{hidden}}^2 \times T) = O(T)$
> - **GSHPooling:** $O(D_{\text{hidden}} \times \text{len}_Q \times T^2 \times P^{-2}) = O(\text{len}_Q \times T^2)$
> - **PlugMemory:** $O(T^2)$
> - **TuneMemory:** $O((T + D_{\text{out}})^2)$
> - **One STanHop Encoder Layer with PlugMemory:** $O(T + T^2 + T + (\text{len}_Q \times T^2) + T^2) = O(T + T^2 + \text{len}_Q)$
> - **One STanHop Encoder Layer with TuneMemory:**
>   $O(T + T^2 + T + (\text{len}_Q \times T^2) + (T + D_{\text{out}})^2)$  $=O(T + T^2 + \text{len}_Q + (T + D_{\text{out}})^2)$
> - **One STanHop Decoder Layer with PlugMemory:**
>   $O(T + T^2 + T + (\text{len}_Q \times T^2) + T^2) = O(T + T^2 + \text{len}_Q)$
> - **One STanHop Decoder Layer with TuneMemory:**
>   $O(T + T^2 + T + (\text{len}_Q \times T^2) + (T + D_{\text{out}})^2) = O(T + T^2 + \text{len}_Q + (T + D_{\text{out}})^2)$
>
> >Can the author provide the limitation section?
>
> Thank you for pointing this out. We have included a discussion on the limitations of this work, the **new Sec. B1** in the latest revision, which we quote as follows:
>
> The proposed generalized sparse modern Hopfield model shares similar inefficiencies due to the $\mathcal{O}(d^2)$ complexity. In addition, the effectiveness of our memory enhancement methods is contingent on the relevance of the external memory set to the specific inference task. Achieving a high degree of relevance in the external memory set often necessitates considerable human effort and domain expertise, just like our selection process detailed in Sec. F.2. This requirement could potentially limit the model's applicability in scenarios where such resources are scarce or unavailable.
>
> >The model is too complex, it may be difficult to optimize.
>
> We appreciate the reviewer's concern regarding the complexity and optimizability of our model.
>
> To address this, we evaluate the model's optimization difficulty qualitatively by considering two factors: the size of the training dataset and the model's prediction performance.
>
> Our results, as detailed in the submitted paper, demonstrate that our model achieves top-tier prediction performance, even with relatively small training datasets. This indicates that despite its complexity, the model is not challenging to optimize. For instance, on the Weather (WTH) dataset, which is only 1.5MB in size, our model attains state-of-the-art (SOTA) and near-SOTA performance across various prediction horizons.
>
> Here are the sizes of the training datasets used and the corresponding performance achievements:
>
> - **Electricity (ECL):** 35.7MB
> - **ETTh1:** 1.8MB
> - **ETTm1:** 6.9MB
> - **Influenza-Like Illness (ILI):** 56KB
> - **Traffic:** 92MB
> - **Weather (WTH):** 1.5MB
>
> These results clearly demonstrate that our model efficiently handles training on datasets of varying sizes while maintaining high prediction accuracy. This evidence strongly suggests that our model, despite its complexity, is not difficult to optimize.
>
> We hope this clarify your concerns. Thank you!

---

> ### Author Response · Authors · 2023-11-13
> **Reproducibility and Contributions**
>
> >The paper does not provide available code source to reproduce the experiments.
>
> We promise that we will make our code publicly available for reproducibility once the paper is accepted.
>
> >The paper's contributions are limited in scope.
>
> We appreciate the opportunity to clarify the scope of our contributions.
> In the initial draft, we may not have described the breadth and depth of our work as effectively as we intended.
>
> **On the technical side**, our contributions are threefold:
>
> 1. **Generalized Sparse Modern Hopfield Model (GSHM):** We present the GSHM as a generalization of [Hu, 2023] and include [Hu, 2023] and [Ramsauer, 2020] as special cases with $\alpha=2$ and $\alpha=1$, respectively. Additionally, we provide a theoretical analysis characterizing the fixed-point convergence property, memory retrieval error bound, noise robustness, and memory capacity. We leverage the GSHM's ability to capture varying degrees of sparsity and noise robustness to handle the intrinsic noise and sparsity of multivariate time series [Masini, 2023; Fawaz, 2019; Wang, 2013; Ozaki, 2012].
> 2. **STanHop-Net for Multivariate Time Series (MTS):** Recognizing that MTS inherently possesses a multi-resolution structure [Shabani, 2023; Cui, 2016], we introduce STanHop-Net. This network utilizes hierarchically stacked GSH layers to manage the multi-level sparsity and noise of an MTS, thereby enhancing performance. Our method effectively captures the intrinsic multi-resolution structure of both temporal and cross-variate correlations of time series, with resolution-specific sparsity at each level (as demonstrated through the component ablation study in Sec. E.3 of the submitted draft).
> 3. **External Memory Plugin Schemes:** We introduce two schemes for memory-enhanced prediction: a Plug-and-Play module and a Tune-and-Play module for train-less and task-aware memory enhancements, respectively. Unlike existing works [Kaiser, 2017], our approach avoids over-specialization on rare events, thus broadening its applicability and enhancing generalization capabilities across various tasks where the frequency and recency of data are less critical. Furthermore, our method demonstrates superior real-time adaptability.
>
> **On the strategic side**, in the era of Large Foundation Models (such as ChatGPT), a theoretical understanding of transformer/attention-based models without overly strong assumptions is increasingly important. MHM offers a low-assumption framework for analyzing attention mechanisms through the lens of associative memory models. This unique perspective opens up new opportunities to study large generative foundation models within a rigorous, brain science-informed scientific framework. We believe our work contributes timely insights by proposing and analyzing the generalized sparse modern Hopfield model.
>
> In summary, our contributions extend beyond the conventional scope, offering both theoretical advancements and practical applications in the realm of neural networks and time series analysis. We hope this clarification better highlights the novelties and impact of our work.
>
> ---
> - [Masini, 2023] Masini, Ricardo P., Marcelo C. Medeiros, and Eduardo F. Mendes. "Machine learning advances for time series forecasting." Journal of economic surveys 37, no. 1 (2023): 76-111.
> - [Shabani, 2023] Shabani, Amin, Amir Abdi, Lili Meng, and Tristan Sylvain. "Scaleformer: iterative multi-scale refining transformers for time series forecasting." ICLR (2023)
> - [Fawaz, 2019] Ismail Fawaz, Hassan, Germain Forestier, Jonathan Weber, Lhassane Idoumghar, and Pierre-Alain Muller. "Deep learning for time series classification: a review." Data mining and knowledge discovery 33, no. 4 (2019): 917-963.
> - [Kaiser, 2017] Kaiser, Łukasz, Ofir Nachum, Aurko Roy, and Samy Bengio. "Learning to remember rare events." arXiv preprint arXiv:1703.03129 (2017).
> - [Cui, 2016] Cui, Zhicheng, Wenlin Chen, and Yixin Chen. "Multi-scale convolutional neural networks for time series classification." arXiv preprint arXiv:1603.06995 (2016).
> - [Wang, 2013] Wang, Zhaoran, Fang Han, and Han Liu. "Sparse principal component analysis for high dimensional multivariate time series." In Artificial Intelligence and Statistics, pp. 48-56. PMLR, 2013.
> - [Ozaki, 2012] Ozaki, Tohru. Time series modeling of neuroscience data. CRC press, 2012.

---

> ### Author Response · Authors · 2023-11-14
> **Q1**
>
> >How does the sparsity of STanHop-Net affect its performance and memory usage?
>
> Thank you for raising this crucial aspect regarding the performance and memory usage of STanHop-Net in relation to its sparsity.
>
> To address this, we have included a **new Sec. E.5** in our latest revision.
> This section specifically examines the memory and GPU usage footprints of our proposed method.
> We conduct a comparative analysis between the Generalized Sparse Hopfield (GSH) model and the vanila **dense** modern Hopfield layer.
>
> The results clearly indicate that the computational cost and memory usage incurred by incorporating an additional $\alpha$ for adaptive sparsity in STanHop-Net is **negligible**. This finding indicates the capability of adapting to varying sparsity levels does not introduce heavy computational burden.
>
> Furthermore, the affects of sparsity in the hopfield layers are included in the Table 1 of the main text. It’s clear that with our proposed GSH model, the performance of STanhop-Net is better.
>
> We hope that the revisions and clarifications provided in this response address the reviewer's concerns and make the value of our work clear. We look forward to further feedback and discussion.

---

> ### Author Response · Authors · 2023-11-21
> **A Gentle Reminder**
>
> Dear Reviewer ntR2,
>
> Thank you for your review and valuable insights on our work. We have addressed all your comments in our detailed response and believe we have resolved all concerns. Should any issues remain, we are ready to provide additional clarifications.
>
> As the rebuttal phase is nearing its deadline, we're looking forward to engaging in a timely discussion.
>
> Thank you again for your time and effort!
>
> Best regards,
>
> Authors

---

### Official Review · Reviewer_BpQa · 2023-11-08

**Soundness:** 3 good
**Presentation:** 3 good
**Contribution:** 3 good
**Rating:** 8
**Confidence:** 4

**Summary:**

This paper extends the recent Sparse Modern Hopfield Network construction of Hu et al (2023) to use the alpha-entmax family of sparse probability mappings (alpha=1 giving Ramsauer et al softmax MHNs, alpha=2 giving the Hu et al (2023) sparsemax HSN), and prove some nice theoretical advantages of this construction.

Further, the paper constructs some neural network layers based on the proposed Generalized Sparse MHN for time series prediction and demonstrates its empirical performance in some benchmarks, with competitive performance.

**Strengths:**

- The GSH construction is a nice and intuitive extension of Sparse Modern Hopfield Networks
- Quite nice theoretical results about the GSH construction  and the impact of alpha.
- Good empirical performance.

**Weaknesses:**

- Some theoretical inaccuracies (perhaps typos?) casting a bit of doubt.
- Missing a few comparisons and reports that I would be very interested in (details below).

**Questions:**

Some experiments and results that I would have liked to see and found valuable:
 - You treat $\alpha$ as a learnable parameter: how is it parametrized and what do the learned values converge to?
 - The case $\alpha \to \infty$ corresponds to using an argmax instead of softmax, i.e., retrieving the most compatible pattern in one step. How would such a "argmax"-based lookup memory perform like in the experiments? (Some of the gradients will be zero, but the same happens some of the times with high alpha too, even if not always.)

Some theoretical issues in definitions:

- in the definition of the Tsallis entropy (3.1), the bottom branch for alpha=1 seems wrong, as it gives the negative of what would be the top branch for alpha=2. I expected by continuity to define alpha=1 as the Shannon entropy -sum p log p. Am I missing something?

- In equation (3.2), the definition of $\Psi_{\alpha}^\star$ seems surprising: $\alpha$-entmax is a vector-value function, thus so should be its integral, but the energy H(x) should be scalar-value. I expect (as stated also elsewhere in the paper) that $\Psi_\alpha^\star$ should be the Fenchel convex conjugate of $\Psi_\alpha$, i.e. $\Psi_\alpha^\star(z) = \sup_{z^\star \in \operatorname{dom}{\Psi_\alpha}} \langle z^\star, z \rangle - \Psi_\alpha(z^\star)$. Could you please clarify?

Other questions:

 - It was not clear to me how the memories Y are constructed; in section 4.3 it seems like the memories must be the same length as the input sequence R. Is this a strong requirement or could it be avoided? A nice property of attention models is that they should support variable length data.
- The qualitative change at alpha=2 in Theorem 3.1 seems surprising and interesting. Could you discuss the difference a bit, especially the difference between the Max terms that show up, and give some intuition about why we see this change? Is one of the bounds always tighter than the other or does this depend on choices of (m, M, d, beta, etc?) I did not have the time to read the proof.

---

> ### Author Response · Authors · 2023-11-13
> **Some theoretical issues in definitions, other questions**
>
> > in the definition of the Tsallis entropy (3.1).... I expected by continuity to define alpha=1 as the Shannon entropy -sum p log p. Am I missing something?
>
> You are absolutely correct. That is a typo. Thanks for pointing this out. We have fixed it.
>
> > In equation (3.2), the definition seems surprising: alpha-enmax is a vector-value function, thus so should be its integral, but the energy H(x) should be scalar-value. I expect (as stated also elsewhere in the paper) that  should be the **Fenchel convex conjugate** of , i.e. . Could you please clarify?
>
> Yes, your observation is astute. Sorry for the lack of clarity. We stated "$\Psi^\star$ is the convex conjugate of $\Psi_\alpha$" in the 1st line of page 4 (and provided explicit form in Definition C.1). However, we believe it can be more precise to avoid the confusion reviewer raised. We have modified the draft accordingly. The revision will be posted shortly.
>
> >It was not clear to me how the memories Y are constructed; in section 4.3 it seems like the memories must be the same length as the input sequence R. Is this a strong requirement or could it be avoided? A nice property of attention models is that they should support variable length data.
>
> Sorry for the confusion. The sequence length of $\mathbf{Y}$ and $\mathbf{R}$ can be arbitrary.
>
> The idea of Sec. 3.2 is to make the retrieval dynamics of modern Hopfield model into a deep learning layer.
> This new layer (GSH) should store and retrieve (hidden) representations throughout a DL pipeline/architecture.
>
> To do so, we first introduce two hidden representations as input of the GSH layer:
> * $\mathbf{Y} \in \mathbb{R}^{M \times a}$: $M$ **raw** stored memory patterns $\mathbf{y}\in \mathbb{R}^a$
> * $\mathbf{R} \in \mathbb{R}^{T \times b}$: $T$ **raw** query patterns $\mathbf{r}\in \mathbb{R}^b$
>
> As we know MHM operates in the $d$-dimensional *associative space*. We have memory and query patterns as follows:
> * $\boldsymbol{\Xi} \in \mathbb{R}^{M \times d}$: $M$ stored memory patterns $\boldsymbol{\xi} \in \mathbb{R}^d$
> * $\mathbf{X} \in \mathbb{R}^{T \times d}$: $T$ query patterns $\mathbf{r}\in \mathbb{R}^d$
>
> Therefore, to transform $\boldsymbol{\Xi}\cdot \alpha$-$\text{EntMax}(\beta \boldsymbol{\Xi}^T\mathbf{X})$ into $\alpha$-$\text{EntMax}(\beta \mathbf{X}^T\mathbf{K})\mathbf{V}$ in eqn. (3.8), we need some linear transformations (i.e., the $\mathbf{W}$ matrices) to connect $(\mathbf{Y},\mathbf{R}) \leftrightarrow (\boldsymbol{\Xi},\mathbf{X})$.
>
> From the above discussion, it's clear that the sequence length of $\mathbf{Y}$ and $\mathbf{R}$ can be arbitrary, as long as their first dimensions match those of $(\boldsymbol{\Xi}, \mathbf{X})$. **Namely, MHM layers support data of arbitrary length.**
> Similar discussions can be found at [https://ml-jku.github.io/hopfield-layers/](https://ml-jku.github.io/hopfield-layers/) (around Eq. 23).
>
> >The qualitative change at alpha=2 in Theorem 3.1 seems surprising and interesting. Could you discuss the difference a bit, especially the difference between the Max terms that show up, and give some intuition about why we see this change? Is one of the bounds always tighter than the other or does this depend on choices of (m, M, d, beta, etc?) I did not have the time to read the proof.
>
> Let $\mathcal{T}(\mathbf{x})$ be the retrieval dynamics acting on query $\mathbf{x}$, abd $\|| \mathcal{T}(\mathbf{x})- \boldsymbol{\xi}||$ be the retrieval error of the GSH model.
>
> For $2\ge \alpha \ge 1$, the max-terms shows up when we connect $\Delta_\mu$, the separation between $\boldsymbol{\xi}_\mu$ and all other memory patterns in $\boldsymbol{\Xi}$, with $\|| \mathcal{T}(\mathbf{x})- \boldsymbol{\xi}||$. Explicitly, the max term when we define such a separation as the minimal inner product difference from one memory pattern  to all other memory patterns, measured by inner product similarity. Please see Def. 3.1 of [Hu, 2023] for more details.
>
> On the other hand, for $\alpha \ge 2$, it is easy to show that $\|\mathcal{T}(\mathbf{x}) - \boldsymbol{\xi}\|$ is consistently smaller than that of the Sparse Modern Hopfield Model [Hu, 2023]. Specifically, greater sparsity leads to a tighter error bound (thus hardmax can achieve perfect retrieval [Millidge, 2022].) Given that $\alpha = 2$ provides the upper bound on retrieval error within this range of $\alpha$: $\|\mathcal{T}(\mathbf{x}) - \boldsymbol{\xi}\| \le \|\mathcal{T}_{\alpha=2}(\mathbf{x}) - \boldsymbol{\xi}\|$, we can utilize the upper bound at $\alpha = 2$ (sparsemax result from [Hu, 2023]) to establish an upper bound. Here, different from above, the max-term arises from component-wise upper bounding, where the *largest* component is used to upper bound the *average/expected value*.
>
> Yes, **it is possible the two bounds might overlap with some set of (M,beta,d,m,n)**. This is a natural consequence as we do not impose any strong assumption on the data and model.
>
> We hope these points clarify the theoretical analysis of our model.

---

> ### Author Response · Authors · 2023-11-14
> **Additional Experiments on Impacts of $\alpha$**
>
> >You treat $\alpha$ as a learnable parameter: how is it parametrized and what do the learned values converge to?
>
> In each layer of the Generalized Sparse Hopfield Model (GSH), we treat $\alpha$ as a learnable scalar parameter, with its range set between 1 and 2. This parameter is updated concurrently with the $W_K, W_Q, W_V$ matrices through backpropagation. As for initialization, we start with a default value of 1.5. As for convergence, we observe that the learned values of $\alpha$ typically scatter within the 1-2 range. This variation aligns with our expectations, as the optimal value of $\alpha$ is expected to be data-dependent
>
> In addition, we provide additional experimental visualizations demonstrating the relationship between
> * (1) training loss and alpha learning curve,
> * (2) noise-level and learned alpha,
>
> in Sec. E.1 and Sec. E.4.
>
>
> >The case corresponds to using an argmax instead of softmax, i.e., retrieving the most compatible pattern in one step. How would such a "argmax"-based lookup memory perform like in the experiments? (Some of the gradients will be zero, but the same happens some of the times with high alpha too, even if not always.)
>
> Thanks for the question. It is an instructive piece to include into our paper.
>
> In the **newly added Sec. E.4 (Fig. 8 & Fig. 9)**,
> we examine the impact of increasing the value of $\alpha$ on memory capacity and noise robustness.
>
> It is known that as $\alpha$ approaches infinity, the $\alpha$-entmax operation transitions to a hardmax operation [Peters 2019; Correia, 2019]. Furthermore, it is also known that memory pattern retrieval using hardmax is expected to exhibit perfect retrieval ability, potentially offering a larger memory capacity than the softmax modern Hopfield model in pure retrieval tasks [Millidge, 2022].
>
> **Our empirical investigations confirm that higher values of $\alpha$ frequently lead to higher memory capacity.**
>
> We report results only up to $\alpha=5$, as we observed that values of $\alpha$ greater than 5 consistently lead to numerical errors, especially when using float32 precision.
> It is crucial to note, that while the hardmax operation (realized when $\alpha \rightarrow \infty$) may maximize memory capacity, its lack of differentiability renders it unsuitable for gradient descent-based optimization.
>
> We hope that the revisions and clarifications provided in this response address the reviewer's concerns and make the value of our work clear. We look forward to further feedback and discussion.
>
> ---
> - [Peters, 2019] Ben Peters, Vlad Niculae, and Andr´e FT Martins. Sparse sequence-to-sequence models. arXiv preprint arXiv:1905.05702, 2019.
> - [Correia, 2019] Gonc¸alo M Correia, Vlad Niculae, and Andr´e FT Martins. Adaptively sparse transformers. arXiv preprint arXiv:1909.00015, 2019.
> - [Millidge, 2022] Beren Millidge, Tommaso Salvatori, Yuhang Song, Thomas Lukasiewicz, and Rafal Bogacz. Universal hopfield networks: A general framework for single-shot associative memory models. In International Conference on Machine Learning, pages 15561–15583. PMLR, 2022.

---

> ### Author Response · Authors · 2023-11-21
> **A Gentle Reminder**
>
> Dear Reviewer BpQa,
>
> Thank you for your review and valuable insights on our work. We have addressed all your comments in our detailed response and believe we have resolved all concerns. Should any issues remain, we are ready to provide additional clarifications.
>
> As the rebuttal phase is nearing its deadline, we're looking forward to engaging in a timely discussion.
>
> Thank you again for your time and effort!
>
> Best regards,
>
> Authors

---

> > ### Comment · Reviewer_BpQa · 2023-11-22
> > **Thank you for your responses.**
> >
> > Sorry for the delay, I am unfortunately ill.
> >
> > Thank you for the responses. This all makes sense.
> >
> > Indeed, my original rating for the paper was already positive anticipating good clarifications for all my questions. I still believe this is a good paper and maintain my score.

---

> > > ### Author Response · Authors · 2023-11-22
> > > **Thank you!**
> > >
> > > Dear Reviewer BpQa22,
> > >
> > > Thank you for the kind words and confirming that our revisions meet your concerns. We greatly appreciate your constructive feedback, which has been instrumental in improving our work.
> > >
> > > Wishing you a speedy recovery!
> > >
> > > Best regards,
> > >
> > > Authors

---

### Author Response · Authors · 2023-11-14
**General Rebuttal/Revision Response**

Dear Reviewers,

We thank the reviewers for the insightful questions and reviews. Your time and effort dedicated to improving our work are truly appreciated.
We have answered all the questions and addressed all of the raised issues in detail in our rebuttal and the latest revision.

All modifications are marked in **BLUE** color.

These revisions include additional explanations, paragraphs and sections to help readers understand the proposed method, and additional experiments to highlight the advantages of it.
Most importantly, a new section E.9 in the appendix (or **[this response](https://openreview.net/forum?id=6iwg437CZs&noteId=p27WvzmvyD)**), has been added to present settings that STanHop-Net **uniformly** outperforms DLinear (the best baseline in our main text.) Ablation studies are also provided in order to back up our analysis.

For the convenience of reviewers/future readers, **[this response](https://openreview.net/forum?id=6iwg437CZs&noteId=VF39one3md)** provides a high-level overview of our contributions.

---
## Revision Details

In response to the reviewers' suggestions, we have made several key modifications, summarized as follows:

1. **Updated Section B:** A **new discussion on memory-augmented neural networks** in the related works section.   [**Reviewer u61L**]

2. **New Section B.1:** A discussion of **limitations of our proposed method**.   [**Reviewer ntR2**]

3. **New Section E.4:** Additional experimental results of the **impacts of varying $\alpha$**. [**Reviewer BpQa**]
>Our empirical investigations confirm that higher values of $\alpha$ lead to higher memory capacity.
4. **New Section E.5:** Additional experiments focusing on the **Memory Usage of STanHop and learnable sparsity ($\alpha$)**.  [**Reviewer ntR2 & u61L**]
>Our results show that the difference in memory and GPU usage between the proposed GSH and the vanila dense modern Hopfield layers is **negligible**.
5. **New Section E.6:** The **complexity analysis of the STanHop-Net**. [**Reviewer ntR2**]

6. **New Section E.7:** The analysis of **[hyperparameter sensitivity (stability) of STanHop-Net](https://openreview.net/forum?id=6iwg437CZs&noteId=NoiGSd2Ae2)**.  [**Reviewer u61L**]
>Our results  show that **STanHop-Net is quite stable**. It is not sensitive to hyperparameter changes.
7. **New Section E.8:** Additional experiments on **[using GSH for Multiple Instance Learning (MIL) tasks](https://openreview.net/forum?id=6iwg437CZs&noteId=r8h9m5uJ5I)**.  [**Reviewer xuHy**]
>Our results  demonstrate that **GSH converges faster than the baselines (dense & sparse MHM) in most settings.**
8. **New Section E.9:** Additional experiments on **[STanHop-Net Uniformly Outperforms DLinear in Settings Dominated by Multivariate Correlations](https://openreview.net/forum?id=6iwg437CZs&noteId=p27WvzmvyD)**.  [**Reviewer xuHy**]
>**STanHop-Net consistently outperforms DLinear in scenarios with strong correlations between variates.** Importantly, our method achieves comparable performance when correlations between variates are weak and demonstrates clear-cut superiority when these correlations are strong.  This explains why DLinear exhibits comparable performance to STanHop-Net in Table 1, when the datasets are not dominated by multivariate correlations.

We have also improved the overall clarity of our draft and corrected all typos identified by both the reviewers and the authors.

We hope these revisions address the reviewers' concerns and improve the overall quality of our paper.

Thank you again for your review!

Best regards,

Authors

---

### Meta-Review · Area_Chair_DDfN · 2023-12-14

**Metareview:**

This paper extends sparse modern Hopfield networks (which use sparsemax activations) by replacing the Gini entropy term by a alpha-Tsallis entropy (leading to alpha-entmax activations). It applies the resulting HNs to multivariate time series prediction problems, by sequentially learning temporal representation and cross-series representation using two tandem sparse Hopfield layers.

There was some disagreement among reviewers about this paper. The author response clarified some of the concerns. Unfortunately the reviewers with the biggest concerns did not engage in the discussion. It is my understanding that the clarifications and corrections made by the authors are satisfactory. This seems to be a solid paper which generalizes prior work in a non-trivial way and demonstrates that the proposed model is useful in practice. Therefore I recommend acceptance.

**Justification For Why Not Higher Score:**

Even though the paper introduces a new model, the magnitude of the contributions with respect to the prior work by Hu et al. 2023 does not seem to justify distinction with spotlight.

**Justification For Why Not Lower Score:**

Already stated in the metareview.

---

### Decision · Program_Chairs · 2024-01-16

Accept (poster)